# Derivative-Free Policy Optimization for Linear Risk-Sensitive and Robust Control Design: Implicit Regularization and Sample Complexity

**Kaiqing Zhang**[♮]  **Xiangyuan Zhang**[♮]  **Bin Hu**  **Tamer Başar**

## Abstract

Direct policy search serves as one of the workhorses in modern reinforcement learning (RL), and its applications in continuous control tasks have recently attracted increasing attention. In this work, we investigate the convergence theory of policy gradient (PG) methods for learning the linear risk-sensitive and robust controller. In particular, we develop PG methods that can be implemented in a *derivative-free* fashion by sampling system trajectories, and establish both global convergence and sample complexity results in the solutions of two fundamental settings in risk-sensitive and robust control: the finite-horizon linear exponential quadratic Gaussian, and the finite-horizon linear-quadratic disturbance attenuation problems. As a by-product, our results also provide the first sample complexity for the global convergence of PG methods on solving zero-sum linear-quadratic dynamic games, a nonconvex-nonconcave minimax optimization problem that serves as a baseline setting in multi-agent reinforcement learning (MARL) with continuous spaces. One feature of our algorithms is that during the learning phase, a certain level of *robustness/risk-sensitivity* of the controller is preserved, which we termed as the *implicit regularization* property, and is an essential requirement in safety-critical control systems.

## 1 Introduction

Recent years have witnessed the rapid development of reinforcement learning (RL) methods in handling continuous control tasks [1, 2, 3]. Central to the success of RL are policy optimization (PO) methods, including policy gradient (PG) [4, 5, 6], actor-critic [7, 8], and other variants [9, 10]. Progress reported in the literature has clearly shown an increasing interest in understanding theoretical properties of PO methods for relatively simple baselines such as linear control problems [11, 12, 13, 14, 15, 16, 17, 18, 19, 20, 21]. However, the theory of model-free PO methods on *risk-sensitive/robust* control remains underdeveloped in the literature. Since risk-sensitivity and robustness are important issues for designing safety-critical systems, it is natural to bring up the questions of whether and how model-free PO methods would converge for these continuous control tasks.

Our work in this paper is motivated by the above concern, and studies the sample complexity of model-free PG methods on two important baseline problems in risk-sensitive/robust control, namely the linear exponential quadratic Gaussian (LEQG), and the linear quadratic (LQ) disturbance attenuation problems. The former covers a fundamental setting in risk-sensitive control, and the latter is an important baseline for robust control. Based on the well-known equivalence between these problems and LQ dynamic games [22, 23, 24, 25, 26], we develop a *unified* PO perspective

---

[♮]Equal contribution. K. Zhang is with the LIDS and the CSAIL at the Massachusetts Institute of Technology. Email: kaiqing@mit.edu. X. Zhang, B. Hu, and T. Başar are with the Department of ECE and the CSL at the University of Illinois at Urbana-Champaign. Emails: {xz7, binhu7, basar1}@illinois.edu.

35th Conference on Neural Information Processing Systems (NeurIPS 2021).

for both. A common feature for the above two problems is that their optimization landscapes are by nature more challenging than that of the linear quadratic regulator (LQR) problem, and existing proof techniques for model-free PG methods [11, 17, 18, 19] are no longer effective due to lack of coercivity of the objective functions. Specifically, when applying PG methods to LQR, the feasible set for the resultant constrained optimization problem is the set of all linear state-feedback controllers that *stabilize* the closed-loop dynamics. The objective function of LQR is coercive on this feasible set and serves as a barrier function itself [14], guaranteeing for the PG iterates to stay in the feasible set and converge to the globally optimal controller. For the LEQG and LQ disturbance attenuation problems, the risk-sensitivity/robustness conditions have reshaped the feasible set in a way that the objective function becomes non-coercive, i.e. the objective value can remain finite while approaching the boundary. The objective is no longer a barrier function, and new proof techniques are needed to show that model-free PG iterates will stay in the feasible set. This is significant for safety-critical control systems with model uncertainty, as the iterates' feasibility here is equivalent to *risk-sensitivity/robustness* of the controller (cf. Remark A.8), and the failure to preserve robustness during learning can cause catastrophic effects, e.g., destabilizing the system in face of disturbances.

The most relevant result was developed in [16], which proposes implicit regularization (IR) arguments to show the convergence of several PG methods on the mixed $\mathcal{H}_2/\mathcal{H}_\infty$ control design problem (which can be viewed as the infinite-horizon variant of the LQ disturbance attenuation problem studied in this paper). The main finding there is that two specific PG search directions (with perfect model information) are automatically biased towards the interior of the robustness-related feasible set. In [16], it is emphasized that IR is a feature of both the *problem* and the *algorithm*, contrasting to that the stability-preserving nature of PG methods for LQR problems is based on the barrier function property of the objective and hence is *algorithm-agnostic*. Although the idea of IR is relevant to PO problems with non-coercive objective functions, the arguments in [16] only apply to the setting with a *known* model, since they rely on a specialized perturbation technique which may potentially generate arbitrarily small "margins". However, in the model-free setting, a uniform margin is required for provable tolerance of statistical errors. See a detailed comparison of the literature in §A.1.

In this paper, for the LEQG and LQ disturbance attenuation problems, we overcome the above margin issue and obtain the first IR result in the model-free setting. This enables the first model-free PG method that provably solves these control problems with a finite number of samples. We highlight our contributions as follows.

**Contributions.** We provide the first sample complexity results for model-free PG methods for solving linear control problems with risk-sensitivity/robustness concerns (the LEQG and LQ disturbance attenuation problems), which were viewed as important open problems in the seminal work [11]. From the robust control perspective, one feature of our algorithms is that, during the learning process, a certain level of *robustness/risk-sensitivity* of the controller is proved to be preserved. This has generalized the results in [16] with a known model, and has thus enabled the finite-sample convergence guarantees of PG methods for risk-sensitive/robust control design. Our algorithms and sample complexity results also address two-player zero-sum LQ dynamic games in the finite-horizon time-varying setting, which are among the first sample complexity results for the global convergence of policy-based methods for competitive multi-agent RL. Second, in the context of minimax optimization, our results address a class of nonconvex-nonconcave minimax constrained optimization problems, using *zeroth-order* multi-step gradient descent-ascent methods. Finally, part of our results provide the sample complexity analysis for PG methods that solve the finite-horizon time-varying LQR problem with system noises and a possibly *indefinite* state-weighting matrix.

## 2 Background

We first introduce two classic settings in risk-sensitive and robust control, namely LEQG, and LQ disturbance attenuation, and their equivalence to zero-sum LQ dynamic games.

### 2.1 Linear Exponential Quadratic Gaussian

Consider the finite-horizon LEQG problem [22, 27, 28], with time-varying systems dynamics described by $x_{t+1} = A_t x_t + B_t u_t + w_t$, $t \in \{0, \cdots, N-1\}$, where $x_t \in \mathbb{R}^m$ represents the system state; $u_t \in \mathbb{R}^d$ is the control input; $w_t \in \mathbb{R}^m$ is an independent (across time) Gaussian random noise with $w_t \sim \mathcal{N}(\mathbf{0}, W)$ for some $W > 0$; the initial state $x_0 \sim \mathcal{N}(\mathbf{0}, X_0)$ for some $X_0 > 0$, which is

independent of $\{w_t\}$; and $A_t, B_t$ are time-varying system matrices. The objective function is given by $\mathcal{J}(\{u_t\}) := \frac{2}{\beta} \log \mathbb{E} \exp\left[\frac{\beta}{2}\left(\sum_{t=0}^{N-1}(x_t^\top Q_t x_t + u_t^\top R_t u_t) + x_N^\top Q_N x_N\right)\right]$, where $Q_t, Q_N \geq 0$ and $R_t > 0$ are symmetric weighting matrices; and $\beta > 0$ is a parameter capturing the degree of risk-sensitivity, which is upper-bounded by some $\beta^* > 0$ [23, 28, 29].

The goal in the LEQG problem is to find the $\mathcal{J}$-minimizing optimal control policy $\mu_t^* : (\mathbb{R}^m \times \mathbb{R}^d)^t \times \mathbb{R}^m \to \mathbb{R}^d$ that maps, at time $t$, the history of state-control pairs up to time $t$ and the current state $x_t$ to the control $u_t$. It has been shown in [22] that $\mu_t^*$ has a linear state-feedback form $\mu_t^*(x_t) = -K_t^* x_t$, where $K_t^* \in \mathbb{R}^{d \times m}$, for all $t$. Therefore, it suffices to search $K_t^*$ in the matrix space $\mathbb{R}^{d \times m}$ for all $t \in \{0, \cdots, N-1\}$, without losing any optimality. The resulting PO problem as well as its closed-form objective function and PG are provided in §A.3.1.

## 2.2 LQ Disturbance Attenuation

Second, consider the LQ disturbance attenuation problem, with time-varying dynamical systems described by $x_{t+1} = A_t x_t + B_t u_t + D_t w_t$ and $z_t = C_t x_t + E_t u_t$, where $x_t \in \mathbb{R}^m$ is the system state; $u_t \in \mathbb{R}^d$ is the control input; $w_t \in \mathbb{R}^n$ is the (unknown) disturbance input; $z_t \in \mathbb{R}^l$ is the controlled output; $A_t, B_t, C_t, D_t, E_t$ are system matrices with appropriate dimensions; and $x_0 \in \mathbb{R}^m$ is unknown. In addition, we assume that $E_t^\top [C_t \ E_t] = [\mathbf{0} \ R_t]$ for some $R_t > 0$ for normalization, with no loss of generality (see §3.5.1 of [25] for a simple procedure that can transform the general problem into a form that satisfies this "normalization" assumption). Subsequently, we denote the $\ell^2$-norms of the vectors $\omega := \left[x_0^\top C_0^\top, w_0^\top, \cdots, w_{N-1}^\top\right]^\top$ and $z := \left[z_0^\top, \cdots, z_{N-1}^\top, x_N^\top Q_N^{1/2}\right]^\top$ as $\|\omega\|$ and $\|z\| = \left\{\mathcal{C}(\{u_t\}, \{w_t\})\right\}^{1/2}$, where $Q_N \geq 0$. Then, the robustness of a designed controller can be guaranteed by a constraint on the ratio between $\|z\|$ and $\|\omega\|$. Specifically, the goal of disturbance attenuation is the following: Given a $\gamma > \gamma^*$, where $\gamma^* = \sqrt{(\beta^*)^{-1}} > 0$ is the *optimal* (*minimax*) level of disturbance attenuation at the output and recall that $\beta^*$ is the upper bound for the risk-sensitivity parameter in LEQG [23, 25], find a control policy $\mu_t = -K_t x_t$, $t \in \{0, \cdots, N-1\}$, that solves $\min_{\{\mu_t\}} \overline{\mathcal{J}}(\{\mu_t(x_t)\})$ subject to $\sup_{x_0, \{w_t\}} \left\{\mathcal{C}(\{\mu_t(x_t)\}, \{w_t\})\right\}^{1/2}/\|\omega\| < \gamma$, where $\overline{\mathcal{J}}(\{\mu_t(x_t)\})$ is an upper bound of the LQG cost [30]. We defer the precise formulations of the LQ disturbance attenuation problem to §A.3.2. Also, see Remark A.3 for the challenges when addressing LEQG and LQ disturbance attenuation directly using derivative-free PG methods.

## 2.3 An Equivalent Dynamic Game Formulation

To overcome the challenges spelled out in Remark A.3, we now introduce an equivalent *dynamic game* formulation to the LEQG and the LQ disturbance attenuation problems due to that the natural PGs of the objective function of the game can be directly sampled (as we will show in §4). Under certain conditions to be introduced shortly, the saddle-point gain matrix of the minimizing player in the dynamic game (if exists) also addresses these two robust/risk-sensitive control problems. We aim to address these three classes of problems in a unified fashion.

Now, consider a two-player zero-sum LQ stochastic dynamic game (henceforth, game) with closed-loop perfect-state information pattern, characterized by $x_{t+1} = A_t x_t + B_t u_t + D_t w_t + \xi_t$, where $x_t \in \mathbb{R}^m$ is the system state, and $u_t := -K_t x_t \in \mathbb{R}^d$ (resp., $w_t := -L_t x_t \in \mathbb{R}^n$) is the linear state-feedback controller of the minimizing (resp., maximizing) player[1], which suffices to attain the saddle point whenever exists [26]. The initial state $x_0$ and the additive process noises $\xi_t$ are drawn independently from a distribution $\mathcal{D}$ and $A_t, B_t, D_t$ are system matrices. Further, we assume that $\mathcal{D}$ is zero-mean, has a positive-definite covariance, and satisfies almost surely that $\|x_0\|, \|\xi_t\| \leq \vartheta$, for all $t \in \{0, \cdots, N-1\}$ and some constant $\vartheta$. The goal of the minimizing (resp. maximizing) player is to minimize (resp. maximize) a quadratic objective function, namely to solve the game

$$\inf_{\{K_t\}} \sup_{\{L_t\}} \mathbb{E}_{x_0, \xi_0, \cdots, \xi_{N-1}}\left[\sum_{t=0}^{N-1} x_t^\top (Q_t + K_t^\top R_t^u K_t - L_t^\top R_t^w L_t)x_t + x_N^\top Q_N x_N\right], \tag{2.1}$$

---

[1] Hereafter we will use player and agent interchangeably. Also, restrictions to instantaneous linear state-feedback policies do not lead any loss of generality, as the results of §2.3 hold for general square-integrable policies, allowed to use the entire state history.

where $Q_t, Q_N \geq 0$, $R_t^u, R_t^w > 0$ are symmetric weighting matrices. Whenever the solution to (2.1) exists such that the inf and sup can be interchanged, the value (2.1) is the *value* of the game, and the corresponding policies are the saddle-point/Nash equilibrium policies. To characterize the solution to (2.1), we first introduce the following time-varying Riccati difference equation (RDE):

$$P_t^* = Q_t + A_t^\top P_{t+1}^* \Lambda_t^{-1} A_t, \quad t \in \{0, \cdots, N-1\}, \tag{2.2}$$

where $\Lambda_t := I + \big(B_t(R_t^u)^{-1}B_t^\top - D_t(R_t^w)^{-1}D_t^\top\big)P_{t+1}^*$ and $P_N^* = Q_N$. From [26], whenever a saddle point exists, the saddle-point control policies are linear state-feedback (i.e., $\mu_{u,t}^*(h_{u,t}) = -K_t^* x_t$ and $\mu_{w,t}^*(h_{w,t}) = -L_t^* x_t$), and the gain matrices $K_t^* = (R_t^u)^{-1}B_t^\top P_{t+1}^* \Lambda_t^{-1} A_t$ and $L_t^* = -(R_t^w)^{-1}D_t^\top P_{t+1}^* \Lambda_t^{-1} A_t$ are unique, where $P_{t+1}^* \geq 0$ is generated by (2.2). Then, we introduce a standard assumption that ensures the existence of the value of the game [25].

**Assumption 2.1** $R_t^w - D_t^\top P_{t+1}^* D_t > 0$, *for all* $t \in \{0, \cdots, N-1\}$, *where* $P_{t+1}^* \geq 0$ *is generated by* (2.2).

Under Assumption 2.1, the value in (2.1) is attained by the sequence $\big(\{K_t^*\}, \{L_t^*\}\big)$, where $K_t^* \in \mathbb{R}^{d \times m}$ and $L_t^* \in \mathbb{R}^{n \times m}$ are gain matrices at the saddle-point/Nash equilibrium. Thus, we can replace the inf and sup in (2.1) with min and max, respectively. Some further notes on Assumption 2.1 are provided in Remark A.5, showing that it is not restrictive and is in fact "quite tight". Lastly, we state the equivalences between three classes of problems, with its proof provided in §B.3.

**Lemma 2.2** *(Connections) For any fixed $\gamma > \gamma^*$ in the LQ disturbance attenuation problem, we can introduce an equivalent LEQG problem and an equivalent zero-sum LQ dynamic game. Specifically, if we set $\beta^{-1}I, R_t, C_t^\top C_t, W$ in LEQG, $\gamma^2 I, R_t, C_t^\top C_t, D_t D_t^\top$ in the LQ disturbance attenuation problem, and $R_t^w, R_t^u, Q_t, D_t D_t^\top$ in the game to be the same, for all $t \in \{0, \cdots, N-1\}$, then the optimal gain matrices in LEQG, the gain matrix in the LQ disturbance attenuation problem, and the Nash equilibrium gain matrix for the minimizing player in the game are the same.*

By Lemma 2.2 and Remark A.3, we will hereafter focus on solving the game formulation in §2.3 using PG methods. As a result, the minimizing controller we obtain solves three classes of problems introduced above altogether.

# 3  Policy Gradient Methods

For ease of analysis, we define

$$\boldsymbol{x} = \big[x_0^\top, \cdots, x_N^\top\big]^\top, \ \boldsymbol{u} = \big[u_0^\top, \cdots, u_{N-1}^\top\big]^\top, \ \boldsymbol{w} = \big[w_0^\top, \cdots, w_{N-1}^\top\big]^\top, \ \boldsymbol{\xi} = \big[x_0^\top, \xi_0^\top, \cdots, \xi_{N-1}^\top\big]^\top, \ \boldsymbol{Q} = diag(Q_{0-N}),$$

$$\boldsymbol{A} = \begin{bmatrix} \mathbf{0}_{m \times mN} & \mathbf{0}_{m \times m} \\ diag(A_{0-(N-1)}) & \mathbf{0}_{mN \times m} \end{bmatrix}, \ \boldsymbol{B} = \begin{bmatrix} \mathbf{0}_{m \times dN} \\ diag(B_{0-(N-1)}) \end{bmatrix}, \ \boldsymbol{D} = \begin{bmatrix} \mathbf{0}_{m \times nN} \\ diag(D_{0-(N-1)}) \end{bmatrix}, \ \boldsymbol{R}^u = diag(R_{0-(N-1)}^u),$$

$$\boldsymbol{R}^w = diag(R_{0-(N-1)}^w), \quad \boldsymbol{K} = \big[diag(K_{0-(N-1)}) \ \mathbf{0}_{dN \times m}\big], \quad \boldsymbol{L} = \big[diag(L_{0-(N-1)}) \ \mathbf{0}_{nN \times m}\big]. \tag{3.1}$$

where $diag(X_{0-N})$ denotes the block-diagonal matrix with $X_0, \cdots, X_N$ on the diagonal block entries. Some other notations are introduced in §A.2. Now, using the compact notations, we develop PG methods with exact PG accesses that provably converge to the Nash equilibrium of the game, $(\boldsymbol{K}^*, \boldsymbol{L}^*) \in \mathcal{S}(d, m, N) \times \mathcal{S}(n, m, N)$, where $\mathcal{S}(d, m, N)$ and $\mathcal{S}(n, m, N)$ are the subspaces that we confine our searches of $\boldsymbol{K}$ and $\boldsymbol{L}$ to, respectively. In particular, we only search over $\boldsymbol{K} \in \mathbb{R}^{Nd \times (N+1)m}$ and $\boldsymbol{L} \in \mathbb{R}^{Nn \times (N+1)m}$, and $\boldsymbol{K}, \boldsymbol{L}$ satisfy the sparsity patterns in (3.1), without losing any optimality. Then, for any $(\boldsymbol{K}, \boldsymbol{L})$, the objective function $\mathcal{G}(\boldsymbol{K}, \boldsymbol{L})$ is given by

$$\mathcal{G}(\boldsymbol{K}, \boldsymbol{L}) = \mathbb{E}_{\boldsymbol{\xi}}\big[\boldsymbol{\xi}^\top \boldsymbol{P}_{\boldsymbol{K}, \boldsymbol{L}} \boldsymbol{\xi}\big] = \text{Tr}\big(\boldsymbol{P}_{\boldsymbol{K}, \boldsymbol{L}} \Sigma_0\big) = \text{Tr}\big((\boldsymbol{Q} + \boldsymbol{K}^\top \boldsymbol{R}^u \boldsymbol{K} - \boldsymbol{L}^\top \boldsymbol{R}^w \boldsymbol{L})\Sigma_{\boldsymbol{K}, \boldsymbol{L}}\big), \tag{3.2}$$

where $\boldsymbol{P}_{\boldsymbol{K}, \boldsymbol{L}} := diag(P_{K_0, L_0}, \cdots, P_{K_N, L_N})$ and $P_{K_t, L_t}$, $t \in \{0, \cdots, N-1\}$, solves the Lyapunov equation

$$P_{K_t, L_t} = (A_t - B_t K_t - D_t L_t)^\top P_{K_{t+1}, L_{t+1}}(A_t - B_t K_t - D_t L_t) + Q_t + K_t^\top R_t^u K_t - L_t^\top R_t^w L_t, \tag{3.3}$$

with $P_{K_N, L_N} := Q_N$. Moreover, $\boldsymbol{P}_{\boldsymbol{K}, \boldsymbol{L}}, \Sigma_{\boldsymbol{K}, \boldsymbol{L}}$ are the solutions to the Lyapunov equations

$$\boldsymbol{P}_{\boldsymbol{K}, \boldsymbol{L}} = (\boldsymbol{A} - \boldsymbol{B}\boldsymbol{K} - \boldsymbol{D}\boldsymbol{L})^\top \boldsymbol{P}_{\boldsymbol{K}, \boldsymbol{L}}(\boldsymbol{A} - \boldsymbol{B}\boldsymbol{K} - \boldsymbol{D}\boldsymbol{L}) + \boldsymbol{Q} + \boldsymbol{K}^\top \boldsymbol{R}^u \boldsymbol{K} - \boldsymbol{L}^\top \boldsymbol{R}^w \boldsymbol{L}, \tag{3.4}$$

$$\Sigma_{\boldsymbol{K}, \boldsymbol{L}} = (\boldsymbol{A} - \boldsymbol{B}\boldsymbol{K} - \boldsymbol{D}\boldsymbol{L})\Sigma_{\boldsymbol{K}, \boldsymbol{L}}(\boldsymbol{A} - \boldsymbol{B}\boldsymbol{K} - \boldsymbol{D}\boldsymbol{L})^\top + \Sigma_0, \ \Sigma_0 := \mathbb{E}\big[diag(x_0 x_0^\top, \xi_0 \xi_0^\top, \cdots, \xi_{N-1}\xi_{N-1}^\top)\big] > 0, \tag{3.5}$$

where $\Sigma_0 > 0$ is full-rank because $x_0, \xi_0, \cdots, \xi_{N-1}$ are drawn independently from $\mathcal{D}$ which has a positive-definite covariance matrix. The solutions to (3.4) and (3.5) always exist and are unique because (3.4) and (3.5) are recursive formulas in blocks. Since $\|x_0\|, \|\xi_t\| \leq \vartheta$ almost surely, for all $t$, $\left\|diag(x_0 x_0^\top, \xi_0 \xi_0^\top, \cdots, \xi_{N-1} \xi_{N-1}^\top)\right\|_F \leq (N+1)\vartheta^2$ almost surely. Next, define the following notations:

$$c_0 := (N+1)\vartheta^2, \qquad d_\Sigma := m^2(N+1), \qquad \phi := \lambda_{\min}(\Sigma_0) > 0, \qquad H_{K,L} := R^w - D^\top P_{K,L} D,$$

$$E_{K,L} := (-R^w + D^\top P_{K,L} D)L - D^\top P_{K,L}(A - BK), \qquad F_{K,L} := (R^u + B^\top P_{K,L} B)K - B^\top P_{K,L}(A - DL), \quad (3.6)$$

$$G_{K,L(K)} := R^u + B^\top \widetilde{P}_{K,L(K)} B, \qquad \widetilde{P}_{K,L(K)} := P_{K,L(K)} + P_{K,L(K)} D(R^w - D^\top P_{K,L(K)} D)^{-1} D^\top P_{K,L(K)}. \quad (3.7)$$

Our goal is to solve the minimax optimization problem $\min_K \max_L \mathcal{G}(K, L)$ such that $\mathcal{G}(K^*, L) \leq \mathcal{G}(K^*, L^*) \leq \mathcal{G}(K, L^*)$ for any $K$ and $L$. Some properties of $\mathcal{G}(K, L)$ are presented in the following lemmas, with their proofs being deferred to §B.4 and §B.5, respectively.

**Lemma 3.1** *(Nonconvexity-Nonconcavity) There exist zero-sum LQ dynamic games such that the objective function $\mathcal{G}(K, L)$ is nonconcave in $L$ for a fixed $K$, and nonconvex in $K$ for a fixed $L$.*

**Lemma 3.2** *(PG & No Spurious Local Minimum) The PGs of $\mathcal{G}(K, L)$ can be computed as $\nabla_K \mathcal{G}(K, L) = 2F_{K,L} \Sigma_{K,L}$ and $\nabla_L \mathcal{G}(K, L) = 2E_{K,L} \Sigma_{K,L}$. Also, if at some stationary point $(K, L)$ of $\mathcal{G}(K, L)$ (i.e., where $\nabla_K \mathcal{G}(K, L) = 0$ and $\nabla_L \mathcal{G}(K, L) = 0$), it holds that $P_{K,L} \geq 0$ and $R^w - D^\top P_{K,L} D > 0$, then $(K, L)$ is the unique Nash equilibrium.*

## 3.1 Double-Loop Scheme

We introduce a *double-loop* update scheme. Specifically, we first fix $K$, and solve for the optimal $L$, denoted as $L(K)$, whenever exists, by maximizing $\mathcal{G}(K, \cdot)$ over $L \in \mathcal{S}(n, m, N)$. Then, we update $K$ to minimize $\mathcal{G}(K, L(K))$ over $K \in \mathcal{S}(d, m, N)$. For each fixed $K$, the inner-loop is an *indefinite* LQR problem where $-Q - K^\top R^u K$ is not positive semi-definite (p.s.d.). Our goal is to find, via PG methods, the maximizing gain matrix $L(K)$ whenever the objective function $\mathcal{G}(K, \cdot)$ admits a finite upper bound value. We formally state the precise condition for the solution to the inner-loop maximization problem to be *well-defined* as follows, with its proof being deferred to §B.6.

**Lemma 3.3** *(Inner-Loop Well-Definedness Condition) For the Riccati equation*

$$P_{K,L(K)} = Q + K^\top R^u K + (A - BK)^\top \widetilde{P}_{K,L(K)}(A - BK), \quad (3.8)$$

*define the following set of outer-loop gain matrices $K$:*

$$\mathcal{K} := \left\{ K \in \mathcal{S}(d, m, N) \,|\, (3.8) \text{ admits a solution } P_{K,L(K)} \geq 0, \text{ and } R^w - D^\top P_{K,L(K)} D > 0 \right\}. \quad (3.9)$$

*Then, $K \in \mathcal{K}$ is* sufficient *for the inner-loop solution $L(K)$ to be well defined. Also, $L(K)$ is unique, and takes the form of:*

$$L(K) = (-R^w + D^\top P_{K,L(K)} D)^{-1} D^\top P_{K,L(K)}(A - BK). \quad (3.10)$$

*Moreover, $K \in \mathcal{K}$ is also* almost necessary, *in that if $K \notin \overline{\mathcal{K}}$, where*

$$\overline{\mathcal{K}} := \left\{ K \in \mathcal{S}(d, m, N) \,|\, (3.8) \text{ admits a solution } P_{K,L(K)} \geq 0, \text{ and } R^w - D^\top P_{K,L(K)} D \geq 0 \right\}, \quad (3.11)$$

*then the solution to the inner loop is not well defined, i.e., the objective $\mathcal{G}(K, \cdot)$ could be driven to arbitrarily large values. Lastly, the solution to (3.8) satisfies $P_{K,L(K)} \geq P_{K,L}, \forall L$ in the p.s.d. sense.*

Note that $\mathcal{K}$ is nonempty due to Assumption 2.1, but it might be *unbounded*. Also, every $K \in \mathcal{K}$ is equivalent to a control gain matrix that attains a $\gamma$-level of disturbance in the original disturbance attenuation problem (cf. Lemma 2.2). Then, by Lemmas 3.1 and 3.3, the inner-loop maximization problem for a fixed $K \in \mathcal{K}$ is nonconcave in $L$ and always admits a unique maximizing solution. Thus, applying a variant of the Danskin's theorem [31] yields that $\max_L \mathcal{G}(K, L)$ is differentiable with respect to $K$ and $\nabla_K \{\max_L \mathcal{G}(K, L)\} = \nabla_K \mathcal{G}(K, L(K))$. The outer loop is then a nonconvex (as we will show in Lemma 3.5) constrained optimization problem in $K$, expressed as $\min_{K \in \mathcal{K}} \mathcal{G}(K, L(K))$. Then, the PG of the outer loop is computed as $\nabla_K \mathcal{G}(K, L(K)) = 2F_{K,L(K)} \Sigma_{K,L(K)}$. Lastly, we comment on the double-loop scheme we study in Remark A.7.

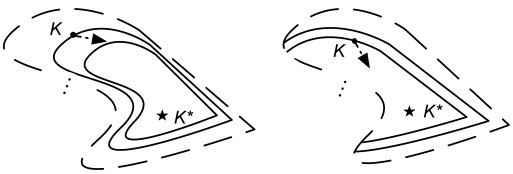

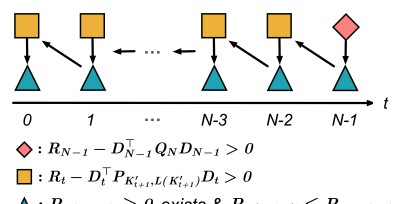

Figure 1: *Left:* Optimization landscape of LQR, where the dashed line represents the boundary of the stabilizing controller set. *Right:* Optimization landscape of the outer loop, with the dashed line representing the boundary of $\mathcal{K}$. The solid lines represent the contour lines of the objective function, $K$ denotes the control gain of one iterate, and ★ is the global minimizer.

Figure 2: Proof idea for Theorem 3.7: For any $K \in \mathcal{K}$, denote the gain matrix after one-step PG update as $K'$. We construct an iterative argument to find a constant stepsize such that $P_{K'_t, L(K'_t)} \geq 0$ exists and satisfies $P_{K'_t, L(K'_t)} \leq P_{K_t, L(K_t)}$ for all $t$. Specifically, any $K'_t$ satisfying ■ also satisfies ▲. Moreover, ♦ is enforced by Assumption 2.1. Combined, $K' \in \mathcal{K}$ is ensured.

## 3.2 Optimization Landscape

For a fixed $K$, we present the optimization landscape of the inner-loop and defer the proof to §B.7.

**Lemma 3.4** *(Inner-Loop Landscape) There exists $K \in \mathcal{K}$ such that $\mathcal{G}(K, L)$ is nonconcave in $L$. For a fixed $K \in \mathcal{K}$, $\mathcal{G}(K, L)$ is coercive (i.e., $\mathcal{G}(K, L) \to -\infty$ as $\|L\|_F \to \infty$), and the superlevel set*

$$\mathcal{L}_K(a) := \left\{ L \in \mathcal{S}(n, m, N) \mid \mathcal{G}(K, L) \geq a \right\} \tag{3.12}$$

*is compact for any $a$ such that $\mathcal{L}_K(a) \neq \varnothing$. Moreover, there exist some $l_{K,L}, \psi_{K,L}, \rho_{K,L} > 0$ such that $\mathcal{G}(K, L)$ is $(l_{K,L}, \rho_{K,L})$ locally Lipschitz and $(\psi_{K,L}, \rho_{K,L})$ locally smooth at $(K, L)$. Further, there exist some $l_{K,a}, \psi_{K,a} > 0$ such that $\mathcal{G}(K, L)$ is $l_{K,a}$-globally Lipschitz and $\psi_{K,a}$-globally smooth over $\mathcal{L}_K(a)$. Lastly, $\mathcal{G}(K, L)$ is $\mu_K$-PL for some $\mu_K > 0$ which depends only on $K$ and the problem parameters.*

The optimization landscape of the inner loop is similar to that of the finite-horizon stochastic LQR [32] (which is an independent work that appears concurrently to our work). However, we allow the state-weighting matrix to be indefinite, while [32] requires $Q$ to be positive-definite. For both our inner-loop problem and the LQR problem in [32], independent process noises (together with the random initial state) with a positive-definite covariance matrix guarantee the non-degeneracy of the state covariance matrix $\Sigma_{K,L}$ at any time, which leads to the stationary point of the inner-loop objective function $\mathcal{G}(K, L)$ also being the unique optimum (cf. Lemma 3.2). Further, the non-degeneracy of $\Sigma_{K,L}$ also ensures that $\mathcal{G}(K, L)$ is PL (cf. Lemma 3.4). These two properties together are essential for establishing the global convergence of PG methods in our nonconvex-nonconcave setting.

Subsequently, we analyze the optimization landscape of the outer loop (proved in §B.8), subject to $\mathcal{K}$ defined by (3.9). Note that $\mathcal{K}$ is critical, as by Lemma 3.3, it is a sufficient and almost necessary condition to ensure that the solution to the associated inner-loop subproblem is well defined. More importantly, from a robust control perspective, such a set $\mathcal{K}$ represents the set of control gains that enjoy a certain level of *robustness*, which share the same vein as the celebrated $\mathcal{H}_\infty$-norm constraint. Indeed, they both enforce the gain matrix to *attenuate* a prescribed level of disturbance. This level of robustness also corresponds to the level of *risk-sensitivity* of the controllers in LEQG problems.

**Lemma 3.5** *(Outer-Loop Landscape) There exist zero-sum LQ dynamic games such that $\mathcal{G}(K, L(K))$ is nonconvex and noncoercive on $\mathcal{K}$. Specifically, as $K$ approaches the boundary of $\mathcal{K}$, $\mathcal{G}(K, L(K))$ does not necessarily approach $+\infty$. Moreover, the stationary point of $\mathcal{G}(K, L(K))$ in $\mathcal{K}$, denoted as $(K^*, L(K^*))$, is unique and constitutes the unique Nash equilibrium of the game.*

Lack of the coercivity brings up challenges for convergence analysis, as a decrease in the value of the objective function cannot ensure feasibility of the updated gain matrix, in sharp contrast to the LQR problem [11, 14]. We illustrate the difficult landscape of the outer loop in Figure 1. To address this challenge, we will show next that the natural PG (NPG) and Gauss-Newton (GN) updates, with some stepsize choices, can automatically preserve the feasibility of the iterates on-the-fly, termed the *implicit regularization* (IR) property.

## 3.3 Update Rules and Global Convergence

We use $l, k \geq 0$ to represent the iteration indices of the inner- and outer-loop updates, respectively, and introduce three PG-based update rules as follows:

PG: $\quad L_{l+1} = L_l + \eta \nabla_L \mathcal{G}(K_k, L_l),$ $\quad$ (3.13) $\qquad K_{k+1} = K_k - \alpha \nabla_K \mathcal{G}(K_k, L(K_k)),$ $\quad$ (3.16)

NPG: $\quad L_{l+1} = L_l + \eta E_{K_k, L_l},$ $\quad$ (3.14) $\qquad K_{k+1} = K_k - \alpha F_{K_k, L(K_k)},$ $\quad$ (3.17)

GN: $\quad L_{l+1} = L_l + \eta H_{K_k, L_l}^{-1} E_{K_k, L_l},$ $\quad$ (3.15) $\qquad K_{k+1} = K_k - \alpha G_{K_k, L(K_k)}^{-1} F_{K_k, L(K_k)},$ $\quad$ (3.18)

where $\eta, \alpha > 0$ are constant stepsizes for the inner loop and the outer loop, respectively. We first present the convergence results for three inner-loop PG updates.

**Theorem 3.6** *(Inner-Loop Global Convergence) For a fixed $K \in \mathcal{K}$ and an arbitrary $L_0$ that induces a finite $\mathcal{G}(K, L_0)$, we define a superlevel set $\mathcal{L}_K(a)$ as in (3.12), where $a < \mathcal{G}(K, L_0)$ is arbitrary. Then, iterates $L_l$ of the updates (3.13), (3.14), and (3.15) with stepsizes satisfying*

$$PG: \eta \leq 1/\psi_{K,a}, \quad NPG: \eta \leq 1/(2\|H_{K,L_0}\|), \quad GN: \eta \leq 1/2,$$

*converge to $L(K)$ with globally linear rates, where $\psi_{K,a}$ is the smoothness constant of $\mathcal{G}(K, L)$ over $\mathcal{L}_K(a)$. Moreover, with $\eta = 1/2$, GN (3.15) converges to $L(K)$ with a locally Q-quadratic rate.*

In contrast to the standard convergence proof for first-order methods in nonconvex optimization problems where the objective function is PL, the proof of Theorem 3.6 is more involved since we do not have the *global* smoothness property in our control setting. A careful analysis that addresses this issue has been carried out in §B.9. For the outer loop, we require the iterates of $K$ to stay within $\mathcal{K}$ in order for the solution to the associated inner-loop subproblem to be well defined. To meet this requirement, we introduce the IR property for the NPG (3.17) and GN (3.18) updates in Theorem 3.7, with its proof being provided in §B.10.

**Theorem 3.7** *(IR) Let $K_0 \in \mathcal{K}$ and let the stepsizes satisfy*

$$NPG: \alpha \leq 1/\|G_{K_0, L(K_0)}\|, \qquad GN: \alpha \leq 1.$$

*Then, the iterates $K_k \in \mathcal{K}$ for all $k \geq 0$. In other words, the sequence of solutions to (3.8), $\{P_{K_k, L(K_k)}\}$, exists, and for all $k \geq 0$, $P_{K_k, L(K_k)}$ always satisfies the conditions in (3.9). Furthermore, the sequence $\{P_{K_k, L(K_k)}\}$ is monotonically non-increasing and bounded below by $P_{K^*, L(K^*)}$, in the p.s.d. sense.*

A key step of the proof for Theorem 3.7 is to ensure the *existence* of a solution to (3.8) along the iterations, by carefully controlling the stepsizes of *certain* descent directions. We provide an illustration of the proof idea in Figure 2. In particular, the IR property holds for NPG and GN directions because they can ensure *matrix-wise* decrease of $P_{K_k, L(K_k)}$, while other descent directions (e.g., vanilla PG) can only decrease $\mathcal{G}(K, L(K))$, which is a *scalar*. We highlight the importance of IR in Remark A.8, and establish the convergence result for the outer loop in the following theorem.

**Theorem 3.8** *(Outer-Loop Global Convergence) Suppose $K_0 \in \mathcal{K}$ and let the stepsizes satisfy*

$$NPG: \alpha \leq 1/(2\|G_{K_0, L(K_0)}\|), \qquad GN: \alpha \leq 1/2.$$

*Then, the sequence of average natural gradient norm squares $\{k^{-1} \sum_{\kappa=0}^{k-1} \|F_{K_\kappa, L(K_\kappa)}\|_F^2\}$, $k \geq 1$, converges to $\mathbf{0}$ with $\mathcal{O}(1/k)$ rate. Moreover, this convergence is towards the unique Nash equilibrium. Lastly, the NPG (3.17) and GN (3.18) updates enjoy locally linear and Q-quadratic rates, respectively, around the Nash equilibrium.*

The proof of Theorem 3.8 is deferred to §B.11. In the derivative-free setting where PGs are estimated through samples of system trajectories, if we can uniformly control the estimation bias using a fixed number of samples per iterate, then Theorems 3.6 and 3.8 together imply that the global convergence to the Nash equilibrium also holds. We will substantiate this in the next section.

## 4 Derivative-Free Policy Gradient Methods

We present the sample complexity of our double-loop algorithm, when the exact PG is not accessible, and can only be estimated through samples of system trajectories. In particular, we propose a zeroth-order NPG (ZO-NPG) algorithm with a (zeroth-order) maximization oracle that approximately solves the inner-loop subproblem (cf. Algorithms 1 and 2). In Remark A.9, we comment on how to construct Algorithms 1 and 2 when explicit knowledge on the system parameters is not available.

---

**Algorithm 1** Inner-Loop Zeroth-Order Maximization Oracle

---

1: Input: gain matrices $(K, L_0)$, iteration $L$, batchsize $M_1$, problem horizon $N$, distribution $\mathcal{D}$, smoothing radius $r_1$, dimension $d_1 = mnN$, stepsize $\eta$.

2: **for** $l = 0, \cdots, L-1$ **do**

3:     **for** $i = 0, \cdots, M_1 - 1$ **do**

4:        Sample $L_l^i = L_l + r_1 U_l^i$, where $U_l^i$ is uniformly drawn from $\mathcal{S}(n, m, N)$ with $\|U_l^i\|_F = 1$.

5:        Simulate $(K, L_l^i)$ and $(K, L_l)$ for horizon $N$ starting from $x_{l,0}^{i,0}, x_{l,0}^{i,1} \sim \mathcal{D}$, and collect the empirical estimates $\overline{\mathcal{G}}(K, L_l^i) = \sum_{t=0}^{N} c_{l,t}^{i,0}$, $\overline{\Sigma}_{K,L_l}^i = diag\left[x_{l,0}^{i,1}(x_{l,0}^{i,1})^\top, \cdots, x_{l,N}^{i,1}(x_{l,N}^{i,1})^\top\right]$, where $\left\{c_{l,t}^{i,0}\right\}$ is the sequence of stage costs following the trajectory generated by $(K, L_l^i)$ and $\left\{x_{l,t}^{i,1}\right\}$ is the sequence of states following the trajectory generated by $(K, L_l)$, for $t \in \{0, \cdots, N\}$, under independently sampled noises $\xi_{l,t}^{i,0}, \xi_{l,t}^{i,1} \sim \mathcal{D}$ for all $t \in \{0, \cdots, N-1\}$.

6:     **end for**

7:     Obtain:   $\overline{\nabla}_L \mathcal{G}(K, L_l) = \frac{1}{M_1} \sum_{i=0}^{M_1-1} \frac{d_1}{r_1} \overline{\mathcal{G}}(K, L_l^i) U_l^i$,     $\overline{\Sigma}_{K,L_l} = \frac{1}{M_1} \sum_{i=0}^{M_1-1} \overline{\Sigma}_{K,L_l}^i$.

8:     Update:   *PG*: $L_{l+1} = L_l + \eta \overline{\nabla}_L \mathcal{G}(K, L_l)$,     *NPG*: $L_{l+1} = L_l + \eta \overline{\nabla}_L \mathcal{G}(K, L_l) \overline{\Sigma}_{K,L_l}^{-1}$.

9: **end for**

10: Return $L_L$.

---

## 4.1 Inner-Loop Maximization Oracle

Sample complexities of zeroth-order PG algorithms for solving standard infinite-horizon LQR have been investigated in both discrete-time [11, 19, 33] and continuous-time [17] settings. Our inner-loop maximization oracle extends the sample complexity result to finite-horizon time-varying LQR with system noises and a possibly *indefinite* state-weighting matrix. In particular, we show that zeroth-order PG and NPG with a one-point minibatch estimation scheme enjoy $\widetilde{\mathcal{O}}(\epsilon_1^{-2})$ sample complexities, where $\epsilon_1$ is the desired accuracy level in terms of inner-loop objective values, i.e., $\mathcal{G}(K, \overline{L}(K)) \geq \mathcal{G}(K, L(K)) - \epsilon_1$ for the $\overline{L}(K)$ returned by the algorithm. The two specific zeroth-order PG updates are introduced as follows:

$$ZO\text{-}PG: L_{l+1} = L_l + \eta \overline{\nabla}_L \mathcal{G}(K, L_l), \qquad (4.1) \qquad ZO\text{-}NPG: L_{l+1} = L_l + \eta \overline{\nabla}_L \mathcal{G}(K, L_l) \overline{\Sigma}_{K,L_l}^{-1}, \qquad (4.2)$$

where $\eta > 0$ is the stepsize to be chosen, $l \geq 0$ is the iteration index, $\overline{\nabla}_L \mathcal{G}(K, L)$ and $\overline{\Sigma}_{K,L}$ are the noisy estimates of $\nabla_L \mathcal{G}(K, L)$ and $\Sigma_{K,L}$, respectively, obtained through zeroth-order oracles. We establish the sample complexity of Algorithm 1 in the following theorem.

**Theorem 4.1** *(Inner-Loop Sample Complexity) Let the desired accuracy level be $\epsilon_1, \delta_1 \in (0, 1)$, and parameters of the minibatch zeroth-order oracle (cf. Algorithm 1) satisfy*

$$\textit{Minibatch Size: } M_1 \sim \widetilde{\mathcal{O}}(\epsilon_1^{-2}), \quad \textit{Stepsize: } \eta \sim \mathcal{O}(1), \quad \textit{Smoothing Radius: } r_1 \sim \Theta(\sqrt{\epsilon_1}).$$

*Then, with probability at least $1 - \delta_1$ and $L \sim \mathcal{O}(\log(\epsilon_1^{-1}))$ steps, updates (4.1) and (4.2) output some $\overline{L}(K) := L_L$ satisfying $\mathcal{G}(K, \overline{L}(K)) \geq \mathcal{G}(K, L(K)) - \epsilon_1$ and $\|L(K) - \overline{L}(K)\|_F \leq \sqrt{\lambda_{\min}^{-1}(H_{K,L(K)}) \cdot \epsilon_1}$.*

Complete versions of Theorem 4.1 for PG and NPG updates (4.1) and (4.2) are given respectively in §B.12 and §B.13, along with their proofs. The total sample complexity of both (4.1) and (4.2) scales as $M_1 \cdot L \sim \widetilde{\mathcal{O}}(\epsilon_1^{-2} \cdot \log(\delta_1^{-1}))$, where the logarithmic dependence on $\epsilon_1$ is suppressed. In contrast to [11], our Algorithm 1 uses an *unperturbed* pair of gain matrices to generate the state sequence for estimating $\Sigma_{K,L}$. This modification avoids the estimation bias induced by the perturbations on the gain matrix, while only adding a constant factor of 2 to the total sample complexity.

## 4.2 Outer-Loop ZO-NPG

With the approximate inner-loop solution $\overline{L}(K)$ obtained from Algorithm 1, the outer-loop ZO-NPG algorithm approximately solves $\min_{K \in \mathcal{K}} \mathcal{G}(K, \overline{L}(K))$, with the following update rule:

$$ZO\text{-}NPG: \quad K_{k+1} = K_k - \alpha \overline{\nabla}_K \mathcal{G}(K_k, \overline{L}(K_k)) \overline{\Sigma}_{K_k, \overline{L}(K_k)}^{-1}, \qquad (4.3)$$

---

**Algorithm 2** Outer-Loop ZO-NPG

---

1: Input: initial gain matrix $K_0 \in \mathcal{K}$, number of iterates $K$, batchsize $M_2$, problem horizon $N$, distribution $\mathcal{D}$, smoothing radius $r_2$, dimension $d_2 = mdN$, stepsize $\alpha$.

2: **for** $k = 0, \cdots, K-1$ **do**

3:     Find $\overline{L}(K_k)$ such that $\mathcal{G}(K_k, \overline{L}(K_k)) \geq \mathcal{G}(K_k, L(K_k)) - \epsilon_1$.

4:     **for** $j = 0, \cdots, M_2 - 1$ **do**

5:         Sample $K_k^j = K_k + r_2 V_k^j$, where $V_k^j$ is uniformly drawn from $\mathcal{S}(d, m, N)$ with $\|V_k^j\|_F = 1$.

6:         Find $\overline{L}(K_k^j)$ such that $\mathcal{G}(K_k^j, \overline{L}(K_k^j)) \geq \mathcal{G}(K_k^j, L(K_k^j)) - \epsilon_1$ using Algorithm 1.

7:         Simulate $(K_k^j, \overline{L}(K_k^j))$ and $(K_k, \overline{L}(K_k))$ for horizon $N$ starting from $x_{k,0}^{j,0}, x_{k,0}^{j,1} \sim \mathcal{D}$, and collect $\overline{\mathcal{G}}(K_k^j, \overline{L}(K_k^j)) = \sum_{t=0}^{N} c_{k,t}^{j,0}, \overline{\Sigma}_{K_k, \overline{L}(K_k)}^{j} = diag\left[x_{k,0}^{j,1}(x_{k,0}^{j,1})^\top, \cdots, x_{k,N}^{j,1}(x_{k,N}^{j,1})^\top\right]$, where $\left\{c_{k,t}^{j,0}\right\}$ is the sequence of stage costs following the trajectory generated by $(K_k^j, \overline{L}(K_k^j))$ and $\left\{x_{k,t}^{j,1}\right\}$ is the sequence of states following the trajectory generated by $(K_k, \overline{L}(K_k))$, for $t \in \{0, \cdots, N\}$, under independently sampled noises $\xi_{k,t}^{j,0}, \xi_{k,t}^{j,1} \sim \mathcal{D}$ for all $t \in \{0, \cdots, N-1\}$.

8:     **end for**

9:     Obtain: $\overline{\nabla}_K \mathcal{G}(K_k, \overline{L}(K_k)) = \frac{1}{M_2} \sum_{j=0}^{M_2-1} \frac{d_2}{r_2} \overline{\mathcal{G}}(K_k^j, \overline{L}(K_k^j)) V_k^j$, $\overline{\Sigma}_{K_k, \overline{L}(K_k)} = \frac{1}{M_2} \sum_{j=0}^{M_2-1} \overline{\Sigma}_{K_k, \overline{L}(K_k)}^j$.

10:     *NPG* update: $K_{k+1} = K_k - \alpha \overline{\nabla}_K \mathcal{G}(K_k, \overline{L}(K_k)) \overline{\Sigma}_{K_k, \overline{L}(K_k)}^{-1}$.

11: **end for**

12: Return $K_K$.

---

where $\alpha > 0$ is the stepsize, $k \geq 0$ is the iteration index, and $\overline{\nabla}_K \mathcal{G}(K, \overline{L}(K))$ and $\overline{\Sigma}_{K, \overline{L}(K)}$ are the estimated PG and state correlation matrices, obtained from Algorithm 2. Similar to the modification in §4.1, we estimate the correlation matrix $\overline{\Sigma}(K, L(K))$ using the state sequence generated by the *unperturbed* gain matrices, $(K, \overline{L}(K))$, to avoid the estimation bias induced by the perturbations on the gain matrices. In the following theorem (Theorem 4.2), with its complete version and proof being deferred to §B.14, we have that the IR property in Theorem 3.7 also holds in the derivative-free setting, with high probability. We provide here a sketch of the proof of Theorem 4.2.

**Theorem 4.2** *(IR: Derivative-Free Setting) For any $K_0 \in \mathcal{K}$, define the following strict subset of $\mathcal{K}$:*

$$\widehat{\mathcal{K}} := \left\{ K \mid (3.8) \text{ admits a solution } P_{K,L(K)} \geq 0, \text{ and } P_{K,L(K)} \leq P_{K_0,L(K_0)} + \frac{\lambda_{\min}(H_{K_0,L(K_0)})}{2\|D\|^2} \cdot I \right\}.$$

*Let $\delta_2 \in (0,1)$ and parameters of the minibatch zeroth-order algorithm (cf. Algorithm 2) satisfy*

$$\text{Minibatch Size: } M_2 \sim \widetilde{\mathcal{O}}(K^4), \quad \text{Stepsize: } \alpha \sim \mathcal{O}(1), \quad \text{Smoothing Radius: } r_2 \sim \mathcal{O}(K^{-1}),$$
$$\text{Inner-Loop Accuracy: } \epsilon_1 \sim \mathcal{O}(K^{-2}), \qquad \delta_1 \sim \mathcal{O}(\delta_2 M_2^{-1} K^{-1}).$$

*Then, it holds with probability at least $1 - \delta_2$ that $K_k \in \widehat{\mathcal{K}} \subset \mathcal{K}$ for all $k \in \{1, \cdots, K\}$.*

*Proof Sketch.* Due to lack of coercivity, the objective function of the outer loop cannot act as a barrier function to guarantee that the iterates of PG updates stay in $\mathcal{K}$ (cf. Figure 1), and thus new candidates are needed. To this end, we construct two compact sets $\widehat{\mathcal{K}}$ and $\mathcal{K}_0$ for a given $K_0 \in \mathcal{K}$, as shown in blue and red in Figure 3, respectively, where $\mathcal{K}_0 := \left\{ K \mid K \in \mathcal{K}, \text{ and } P_{K,L(K)} \leq P_{K_0,L(K_0)} \right\}$. Clearly, $K_0 \in \partial \mathcal{K}_0$ and a sequence of iterates $\{K_k\}$ that stays in $\widehat{\mathcal{K}}$ or $\mathcal{K}_0$ is also uniformly separated from $\partial \mathcal{K}$ (dashed lines in Figure 3). Denote the gain matrix after one step of the exact NPG update (3.17) as $\widetilde{K}_1$. Then, under an appropriate stepsize, Theorem 3.7 demonstrates that $P_{\widetilde{K}_1, L(\widetilde{K}_1)} \geq 0$ exists and satisfies $P_{\widetilde{K}_1, L(\widetilde{K}_1)} \leq P_{K_0, L(K_0)}$ almost surely, implying $\widetilde{K}_1 \in \mathcal{K}_0$ almost surely. In contrast, when the model is not known, one step of the ZO-NPG update (4.3) using estimated gradients sampled through system trajectories could drive the gain matrix outside of $\mathcal{K}_0$ (even worse, outside of $\mathcal{K}$) due to the induced statistical errors. Moreover, these errors accumulate over all the iterations, raising significant challenges to find a uniform "margin" that is needed to safely select the parameters of Algorithm 2.

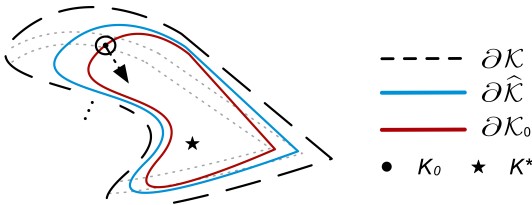

Figure 3: Illustrating the proof idea for Theorem 4.2. For any $K_0 \in \mathcal{K}$, we construct two compact sets $\widehat{\mathcal{K}}$ and $\mathcal{K}_0$ shown in blue and red, respectively, that are independent of the contour lines of the objective function. Our analysis proves that the iterates following (4.3) stay within $\widehat{\mathcal{K}}$ with high probability, thus uniformly separated from $\partial\mathcal{K}$, with high probability.

To overcome this challenge, we establish some arguments stronger than that in Theorem 3.7, i.e., $K_k$ stays in $\widehat{\mathcal{K}}$, for all $k$, with high probability. We first show that with polynomial samples, the estimated NPG could be accurate enough such that $K_1$, the iterate after applying one step of (4.3), is close to $\widetilde{K}_1$, and thus stays in $\widehat{\mathcal{K}}$, with high probability. The same arguments could be iteratively applied to all future iterations, because starting from any $K_k \in \mathcal{K}$ and choosing an appropriate stepsize, Theorem 3.7 guarantees that the iterates following the exact NPG direction are monotonically moving toward the interior of $\mathcal{K}$. Also, there exist parameters of Algorithm 2 such that the NPG estimates could be arbitrarily close to the exact ones. These two properties together imply that we can control, with high probability, the rate under which the iterates following (4.3) is moving "outward", i.e. toward $\partial\mathcal{K}$. Therefore, we manage to demonstrate that even in the worse case, the iterates of the ZO-NPG update will not travel beyond $\partial\widehat{\mathcal{K}}$ (the blue line), with high probability. Since $\widehat{\mathcal{K}}$ is compact, we can then safely choose the parameters of Algorithm 2 when analyzing the convergence rate of (4.3). ∎

Theorem 4.2 appears to be the first IR result of PO in robust control in the derivative-free setting, with previous work [16] focusing only on the case with exact PG accesses. With Theorem 4.2 at hand, we now present the sample complexity of ZO-NPG (4.3), deferring its proof to §B.15.

**Theorem 4.3** *(Outer-Loop Sample Complexity) For any $K_0 \in \mathcal{K}$, let $\epsilon_2 \leq \phi/2$, $\delta_2, \epsilon_1, \delta_1, M_2, r_2, \alpha$ in Algorithm 2 have the same order as the requirements introduced in Theorem 4.2. Then, it holds with probability at least $1 - \delta_2$ that the sequence $\{K_k\}$, $k \in \{0, \cdots, K\}$, converges with $\mathcal{O}(1/K)$ rate in the sense that $K^{-1} \sum_{k=0}^{K-1} \|F_{K_k, L(K_k)}\|_F^2 \leq \epsilon_2$ with $K = \text{Tr}(P_{K_0, L(K_0)} - P_{K^*, L(K^*)})/[\alpha\phi\epsilon_2]$.*

Hence, ZO-NPG converges to the $\epsilon_2$-neighborhood of the Nash equilibrium with probability at least $1 - \delta_2$, after using $M_2 \cdot K \sim \widetilde{\mathcal{O}}(\epsilon_2^{-5} \cdot \log(\delta_2^{-1}))$ samples. Moreover, the convergent gain matrix $K_K$ also solves the LEQG and the LQ disturbance attenuation problems (cf. Lemma 2.2).

Compared with the $\widetilde{\mathcal{O}}(\epsilon^{-2})$ rate of PG methods for solving LQR [19] and our inner-loop (cf. Theorem 4.1), the $\widetilde{\mathcal{O}}(\epsilon^{-5})$ rate of ZO-NPG for solving our outer-loop is less efficient, due to the much richer landscape presented in Lemma 3.5. In particular, the objective function of LQR is coercive (leading to the "compactness" of sub-level sets and the "smoothness" over the sub-level sets, and enabling the use of "any" descent direction of the objective to ensure feasibility) and PL. In contrast, none of these properties hold true for the objective function of our outer loop, and a descent direction of the objective value may still drive the iterates out of $\mathcal{K}$, which will lead to unbounded/undefined value.

To guarantee that the ZO-NPG iterates will stay within $\mathcal{K}$ (which is necessary for the iterates to be well-defined), we need to use large number of samples ($\widetilde{\mathcal{O}}(\epsilon^{-5})$ in $\epsilon$ and polynomial in other parameters) to obtain a very accurate approximation of the exact NPG update (3.17), similar to what has been done in [11], to ensure that $P_{K, L(K)}$ is monotonically non-increasing in the p.s.d. sense with a high probability along iterations. For LQR, any descent direction of the objective value, which is a scalar, suffices to guarantee that the PG iterates will stay in the feasible set. Therefore, for LQR, the $\widetilde{\mathcal{O}}(\epsilon^{-2})$ rate is expected for the smooth (since the iterates will not leave the feasible set, which is compact) and PL objective functions; while for our robust control setting, due to the more stringent requirements on the estimation accuracy and the non-PL objective function, the $\widetilde{\mathcal{O}}(\epsilon^{-5})$ rate is reasonable. In fact, to our knowledge, there is no sample complexity lower-bound applicable to our outer loop (stochastic nonconvex optimization with no global smoothness nor coercivity nor PL condition, with an inner-loop oracle).

## Acknowledgments

K. Zhang, X. Zhang, and T. Başar were supported in part by the US Army Research Laboratory (ARL) Cooperative Agreement W911NF-17-2-0196, and in part by the Office of Naval Research (ONR) MURI Grant N00014-16-1-2710. B. Hu was supported by the National Science Foundation (NSF) award CAREER-2048168. The authors would like to thank Dhruv Malik and Na Li for helpful discussions and feedback. The authors would also like to thank the anonymous reviewers of NeurIPS for valuable comments and suggestions.

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
