# Supplementary Materials for "Derivative-Free Policy Optimization for Linear Risk-Sensitive and Robust Control Design: Implicit Regularization and Sample Complexity"

## A   Extended Literature Review, Formulations, and Discussions

### A.1   Related Work

**Risk-Sensitive/Robust Control and Dynamic Games.**   $\mathcal{H}_\infty$-robust control has been one of the most fundamental research fields in control theory, addressing worst-case controller design for linear plants in the presence of unknown disturbances and uncertainties. Frequency-domain and time-domain/state-space formulations of the $\mathcal{H}_\infty$-robust control problem were first introduced in [34] and [35], respectively. Based on the time-domain approach, the precise equivalence relationships between controllers in the disturbance attenuation (which belongs to a class of $\mathcal{H}_\infty$-robust control) problem, risk-sensitive linear control, and zero-sum LQ dynamic games have been studied extensively [22, 23, 24, 36, 26, 25]. Specifically, [22] first demonstrated the equivalence of the controllers between LEQG and zero-sum LQ (differential) games; [23] studied the relationship between the mixed $\mathcal{H}_2/\mathcal{H}_\infty$ design problem, a sub-problem of the $\mathcal{H}_\infty$-robust control problem, and LEQG; [24, 36] investigated the disturbance attenuation problem through a dynamic games approach, for both finite- and infinite-horizons, time-invariant and -varying, deterministic and stochastic settings. We refer the readers to [26, 25] for comprehensive studies of $\mathcal{H}_\infty$-robust control, dynamic game theory, risk-sensitive control, and also their precise interconnections.

**Policy Optimization for LQ Control.**   Applying PG methods to LQ control problems has been investigated in both control and learning communities extensively. In contrast to the early work on this topic [37, 38], the recent study focuses more on theoretical aspects such as global convergence and sample complexity of these PG methods [11, 12, 13, 14, 15, 39, 17, 16, 40, 19, 41, 33]. Specifically, [11] was the first work to show the global convergence of PG methods for the LQR problem. Initial sample complexity results for the derivative-free PG methods based on zeroth-order optimization techniques were also reported in [11]. Subsequently, [14] characterized the optimization landscape of the LQR in detail, and provided an initial extension toward the distributive LQR. Building upon these works, [19] enhanced the sample complexity result by adopting a two-point zeroth-order optimization method, in contrast to the single-point method adopted in [11]. The sample complexity result of the two-point method was further improved recently in both continuous- [17] and discrete-time LQR [33]. Moreover, [41] studied the global convergence of PG methods for the LQR where both the state and control weighting matrices are indefinite, but without addressing the derivative-free setting; [15, 39] derived global convergence of PG methods for zero-sum LQ dynamic games, also pointing to the need to investigate the indefinite LQR. The most relevant works that addressed LQ control with *robustness/risk-sensitivity concerns* using policy optimization methods were [40, 42, 16]. The work in [40] considered LQR with multiplicative noises, which enjoys a similar landscape as standard LQR. The work in [42] studied finite-horizon risk-sensitive nonlinear control, and established stationary-point convergence when using iterative LEQG and performing optimization over control actions directly. [16] examined the mixed $\mathcal{H}_2/\mathcal{H}_\infty$ control design problem, a class of $\mathcal{H}_\infty$-robust control problems, with a different landscape from LQR. However, no sample complexity results were provided in either [42] or [16]. Very recently, [32] has also studied the convergence of policy gradient method for finite-horizon LQR problems. Interestingly, the landscape analysis independently developed in [32] matches some of our findings for the inner-loop subproblem of zero-sum LQ dynamic games, while we consider a more general setting with time-varying system dynamics and a possibly indefinite state-weighting matrix. Other recent results on PO for LQ control include [12, 43, 18, 13, 20, 44, 21]. We also refer the readers to [45, 46] for analyses of model-based methods in solving LQ control problems, and to [47, 48] for online control of LQ systems.

**Nonconvex-Nonconcave Minimax Optimization & MARL.** Solving minimax problems using PG methods, especially gradient descent-ascent (GDA), has been one of the most popular sub-fields in the optimization community recently. Convergence of the first-order GDA algorithm (and its variants) has been studied in [49, 50, 51, 52] for general nonconvex-nonconcave objectives, in [53] for the setting with an additional two-sided Polyak-Łojasiewicz (PL) condition, and in [54] for the setting under the weak Minty variational inequality condition. In fact, [55] has shown very recently the computational hardness of solving nonconvex-nonconcave constrained minimax problems with

first-order methods. To the best of our knowledge, *zeroth-order/derivative-free* methods, as studied in our paper, have not yet been investigated for nonconvex-nonconcave minimax optimization problems. On the other hand, in the multi-agent RL (MARL) regime, policy optimization for solving zero-sum Markov/dynamic games naturally leads to a nonconvex-nonconcave minimax problem [15, 56, 57], whose convergence guarantee has remained open [56] until very recently, except for the aforementioned LQ setting [15, 39]. In particular, [57] has established the first non-asymptotic global convergence of independent policy gradient methods for tabular (finite-state-action) zero-sum Markov games, where GDA with two-timescale stepsizes is used. In stark contrast to these works, the crucial global *smoothness* assumption (or property that automatically holds in the tabular MARL setting) on the objective therein does not hold in our control setting with *unbounded* and *continuous* spaces. A careful characterization of the iterate trajectory is thus required in order to establish global convergence and sample complexity results.

**Risk-Sensitive/Robust RL.** Risk-sensitivity/robustness to model uncertainty/misspecification in RL has attracted significant research efforts, following the frameworks of robust adversarial RL (RARL) and robust Markov decision process (RMDP). Tracing back to [58], early works on RARL exploited the same game-theoretic perspective as in $\mathcal{H}_\infty$-robust control, by modeling the uncertainty as a fictitious adversary against the nominal agent. This worst-case (minimax) design concept then enabled actor-critic type algorithms with exceptional empirical performance, but rather sparse theoretical footprints [59]. Very recently, theoretical investigations on the convergence and stability of RARL have been carried out in [60] within the LQ setting. In parallel, the RMDP framework has been proposed in [61, 62], and the robustness of RL algorithms under such framework has been further studied in [63, 64, 65] for the *finite* MDP settings, in contrast to our continuous control tasks. Other recent advances on risk-sensitive/robust RL/MDP include [66, 67].

**Contributions Over Existing Works.** Seminal works [11, 19] presented sample complexity results for the infinite-horizon deterministic LQR, and [32] studied the finite-horizon stochastic LQR. The main technical challenge between our robust control setting and [11, 19, 32] is the lack of coercivity of the objective function (cf. Lemma 3.5). In particular, the objective value may remain finite while approaching the boundary of the robustness constraint set $\mathcal{K}$ (cf. (3.9)), and a descent direction of the objective value may still drive the iterates out of $\mathcal{K}$, which will lead to an unbounded/undefined value. To address this "landscape" challenge in the model-based setting, [15] used an extra projection step, which is restrictive since it requires model information as well as non-standard assumptions. This projection step has been removed in [39] and [16], again in the case when the model is known. Specifically, the Riccati-based arguments in [39] are tailored for the zero-sum game setting and require assumptions that are non-standard for robust control. The implicit regularization arguments in [16] align better with the robust control literature, by leveraging standard tools such as the KYP lemma. However, the arguments in [16] only apply to the setting with a known model, as they rely on a specialized "perturbation technique" which may potentially generate arbitrarily small "margins" between the iterates and the boundary of the robustness feasible set. Such an argument cannot be directly applied to our model-free setting where a uniform margin is required for provable tolerance of statistical errors, when "samples" are used to estimate the policy gradients. Our work has overcome the technical challenges mentioned above, by developing the first implicit regularization results when applying model-free PG methods for robust controller designs.

## A.2 Notations

For a square matrix $X$ of proper dimension, we use $\text{Tr}(X)$ to denote its trace. We also use $\|X\|$ and $\|X\|_F$ to denote, respectively, the operator norm and the Frobenius norm of $X$. If $X$ is further symmetric, we use $X > 0$ to denote that $X$ is positive definite. Similarly, $X \geq 0$, $X \leq 0$, and $X < 0$ are used to denote $X$ being positive semi-definite, negative semi-definite, and negative definite, respectively. Further, again for a symmetric matrix $X$, $\lambda_{\min}(X)$ and $\lambda_{\max}(X)$ are used to denote, respectively, the smallest and the largest eigenvalues of $X$. Moreover, we use $\langle X, Y \rangle := \text{Tr}(X^\top Y)$ to denote the standard matrix inner product and use $diag(X_0, \cdots, X_N)$ to denote the block-diagonal matrix with $X_0, \cdots, X_N$ on the diagonal block entries. In the case where $X_0 = X_1 = \cdots = X_N = X$, we further denote it as $diag(X^N)$. We use $x \sim \mathcal{N}(\mu, \Sigma)$ to denote a Gaussian random variable with mean $\mu$ and covariance $\Sigma$ and use $\|v\|$ to denote the Euclidean norm of a vector $v$. Lastly, we use $\boldsymbol{I}$ and $\boldsymbol{0}$ to denote the identity and zero matrices with appropriate dimensions.

## A.3 Extended Backgrounds

In this section, we first introduce two classic settings in risk-sensitive and robust control, namely LEQG, and LQ disturbance attenuation. We then discuss, in §2.2, the challenges one confronts when attempting to address the above two problems directly using derivative-free PG methods by sampling system trajectories. Fortunately, solving zero-sum LQ (stochastic) dynamic games, a benchmark setting in MARL, via derivative-free PG methods by sampling system trajectories provides a workaround to address these problems all in a unified way, due to the well-known equivalence relationships between zero-sum LQ dynamic games and the two aforementioned classes of problems [25], which we will also discuss in §A.3.3.

### A.3.1 Linear Exponential Quadratic Gaussian

We first consider a fundamental setting of risk-sensitive optimal control, known as the LEQG problem [22, 27, 28], in the finite-horizon setting. The time-varying (linear) systems dynamics are described by:

$$x_{t+1} = A_t x_t + B_t u_t + w_t, \quad t \in \{0, \cdots, N-1\},$$

where $x_t \in \mathbb{R}^m$ represents the system state; $u_t \in \mathbb{R}^d$ is the control input; $w_t \in \mathbb{R}^m$ is an independent (across time) Gaussian random noise drawn from $w_t \sim \mathcal{N}(\mathbf{0}, W)$ for some $W > 0$; the initial state $x_0 \sim \mathcal{N}(\mathbf{0}, X_0)$ is a Gaussian random vector for some $X_0 > 0$, independent of the sequence $\{w_t\}$; and $A_t, B_t$ are time-varying system matrices with appropriate dimensions. The objective function is given by

$$\mathcal{J}(\{u_t\}) := \frac{2}{\beta} \log \mathbb{E} \exp \left[ \frac{\beta}{2} \Big( \sum_{t=0}^{N-1} (x_t^\top Q_t x_t + u_t^\top R_t u_t) + x_N^\top Q_N x_N \Big) \right],$$

where $Q_t \geq 0$ and $R_t > 0$, for all $t \in \{0, \cdots, N-1\}$, and $Q_N \geq 0$ are symmetric weighting matrices; and $\beta > 0$ is a parameter capturing the degree of risk-sensitivity, which is upper-bounded by some $\beta^* > 0$ [23, 28, 29].

The goal in the LEQG problem is to find the $\mathcal{J}$-minimizing optimal control policy $\mu_t^* : (\mathbb{R}^m \times \mathbb{R}^d)^t \times \mathbb{R}^m \to \mathbb{R}^d$ that maps, at time $t$, the history of state-control pairs up to time $t$ and the current state $x_t$ to the control $u_t$, this being so for all $t \in \{0, \cdots, N-1\}$. It has been shown in [22] that $\mu_t^*$ has a linear state-feedback form $\mu_t^*(x_t) = -K_t^* x_t$, where $K_t^* \in \mathbb{R}^{d \times m}$, for all $t \in \{0, \cdots, N-1\}$. Therefore, it suffices to search $K_t^*$ in the matrix space $\mathbb{R}^{d \times m}$ for all $t \in \{0, \cdots, N-1\}$, without losing any optimality. The resulting policy optimization problem is then represented as (by a slight abuse of notation with regard to $\mathcal{J}$):

$$\min_{\{K_t\}} \mathcal{J}(\{K_t\}) := \frac{2}{\beta} \log \mathbb{E} \exp \left[ \frac{\beta}{2} \Big( \sum_{t=0}^{N-1} \big( x_t^\top (Q_t + K_t^\top R_t K_t) x_t \big) + x_N^\top Q_N x_N \Big) \right]. \tag{A.1}$$

To characterize the solution to (A.1), we first introduce the following time-varying Riccati difference equation (RDE):

$$P_{K_t} = Q_t + K_t^\top R_t K_t + (A_t - B_t K_t)^\top \widetilde{P}_{K_{t+1}} (A_t - B_t K_t), \quad t \in \{0, \cdots, N-1\}, \quad P_{K_N} = Q_N, \tag{A.2}$$

where $\widetilde{P}_{K_{t+1}} := P_{K_{t+1}} + \beta P_{K_{t+1}} (W^{-1} - \beta P_{K_{t+1}})^{-1} P_{K_{t+1}}$. Then, (A.1) can be expressed by the solution to (A.2) and the exact PG can be analyzed, as follows, with its proof being deferred to §B.1.

**Lemma A.1** *(Closed-Form Objective Function and PG) For any sequence of $\{K_t\}$ such that (A.2) generates a sequence of positive semi-definite (p.s.d.) sequence $\{P_{K_t}\}$ satisfying $X_0^{-1} - \beta P_{K_0} > 0$ and $W^{-1} - \beta P_{K_t} > 0$, for all $t \in \{1, \cdots, N\}$, the objective function $\mathcal{J}(\{K_t\})$ can be expressed as*

$$\mathcal{J}(\{K_t\}) = -\frac{1}{\beta} \log \det(\mathbf{I} - \beta P_{K_0} X_0) - \frac{1}{\beta} \sum_{t=1}^{N} \log \det(\mathbf{I} - \beta P_{K_t} W). \tag{A.3}$$

As $\beta \to 0$, (A.3) *reduces to* $\mathrm{Tr}(P_{K_0} X_0) + \sum_{t=1}^{N} \mathrm{Tr}(P_{K_t} W)$, *which is the objective function of the* finite-horizon *linear-quadratic-Gaussian (LQG) problem. Moreover, the PG of* (A.3) *at time t has the form:*

$$\nabla_{K_t} \mathcal{J}\big(\{K_t\}\big) := \frac{\partial \mathcal{J}\big(\{K_t\}\big)}{\partial K_t} = 2\Big[(R_t + B_t^\top \widetilde{P}_{K_{t+1}} B_t)K_t - B_t^\top \widetilde{P}_{K_{t+1}} A_t\Big]\Sigma_{K_t}, \quad t \in \{0, \cdots, N-1\}, \quad \text{(A.4)}$$

*where*

$$\Sigma_{K_t} := \prod_{i=0}^{t-1} \Big[(I - \beta P_{K_{i+1}} W)^{-\top}(A_i - B_i K_i)\Big] \cdot X_0^{\frac{1}{2}}(I - \beta X_0^{\frac{1}{2}} P_{K_0} X_0^{\frac{1}{2}})^{-1} X_0^{\frac{1}{2}}$$

$$\cdot \prod_{i=0}^{t-1} \Big[(A_i - B_i K_i)^\top (I - \beta P_{K_{i+1}} W)^{-1}\Big]$$

$$+ \sum_{\tau=1}^{t} \Bigg\{ \prod_{i=\tau}^{t-1} \Big[(I - \beta P_{K_{i+1}} W)^{-\top}(A_i - B_i K_i)\Big] \cdot W^{\frac{1}{2}}(I - \beta W^{\frac{1}{2}} P_{K_\tau} W^{\frac{1}{2}})^{-1} W^{\frac{1}{2}}$$

$$\cdot \prod_{i=\tau}^{t-1} \Big[(A_i - B_i K_i)^\top (I - \beta P_{K_{i+1}} W)^{-1}\Big]\Bigg\}.$$

### A.3.2 LQ Disturbance Attenuation

Second, we introduce the *optimal* LQ disturbance attenuation problem, again for finite-horizon settings, with time-varying (deterministic) dynamical systems described by

$$x_{t+1} = A_t x_t + B_t u_t + D_t w_t, \quad z_t = C_t x_t + E_t u_t, \quad t \in \{0, \cdots, N-1\},$$

where $x_t \in \mathbb{R}^m$ is the system state; $u_t \in \mathbb{R}^d$ is the control input; $w_t \in \mathbb{R}^n$ is the (unknown) disturbance input; $z_t \in \mathbb{R}^l$ is the controlled output; $A_t$, $B_t$, $C_t$, $D_t$, $E_t$ are system matrices with appropriate dimensions; and $x_0 \in \mathbb{R}^m$ is unknown. In addition, we assume that $E_t^\top [C_t \ E_t] = [0 \ R_t]$ for some $R_t > 0$ to eliminate the cross-weightings between control input $u$ and system state $x$, which provides no loss of generality, and is standard in the literature [23, 25]. As shown in §3.5.1 of [25], a simple procedure can transform the general problem into a form where there are no coupling terms between $u$ and $x$, satisfying the above "normalization" assumption that we have made. Subsequently, we introduce the $\ell^2$-norms of the vectors $\omega := \big[x_0^\top C_0^\top, w_0^\top, \cdots, w_{N-1}^\top\big]^\top$ and $z := \big[z_0^\top, \cdots, z_{N-1}^\top, x_N^\top Q_N^{1/2}\big]^\top$ as $\|\omega\| := \big\{x_0^\top C_0^\top C_0 x_0 + \sum_{t=0}^{N-1} \|w_t\|^2\big\}^{1/2}$ and $\|z\| = \big\{\mathcal{C}(\{u_t\}, \{w_t\})\big\}^{1/2} := \big\{x_N^\top Q_N x_N + \sum_{t=0}^{N-1} \|z_t\|^2\big\}^{1/2}$, where $Q_N \geq 0$ is symmetric, and $\|\cdot\|$ in the two expressions denote appropriate Euclidean norms. Then, the robustness of a designed controller can be guaranteed by a constraint on the ratio between $\|z\|$ and $\|\omega\|$. Specifically, the goal of the optimal LQ disturbance attenuation problem is to find the optimal control policy $\mu_t^* : (\mathbb{R}^m \times \mathbb{R}^d)^t \times \mathbb{R}^m \to \mathbb{R}^d$ that maps, at time $t$, the history of state-control pairs until time $t$ and the current state $x_t$, denoted as $h_t := \{x_0, \cdots, x_t, u_0, \cdots, u_{t-1}\}$, to the control $u_t$, such that

$$\gamma^* = \inf_{\{\mu_t\}} \sup_{x_0, \{w_t\}} \frac{\big\{\mathcal{C}(\{\mu_t(h_t)\}, \{w_t\})\big\}^{1/2}}{\|\omega\|} = \sup_{x_0, \{w_t\}} \frac{\big\{\mathcal{C}(\{\mu_t^*(h_t)\}, \{w_t\})\big\}^{1/2}}{\|\omega\|}, \quad \text{(A.5)}$$

where $\gamma^* = \sqrt{(\beta^*)^{-1}} > 0$ is the *optimal* (*minimax*) level of disturbance attenuation at the output [23, 25], and recall that $\beta^*$ is the upper bound for the risk-sensitivity parameter in LEQG. It is known that under closed-loop perfect-state information pattern, the existence of a sequence $\{\mu_t^*\}$, $t \in \{0, \cdots, N-1\}$, is always guaranteed and $\mu_t^*$ is linear state-feedback, i.e., $\mu^*(h_t) = -K_t^* x_t$ ([25]; Theorem 3.5). Thus, it suffices to search for $K_t^* \in \mathbb{R}^{d \times m}$, for all $t \in \{0, \cdots, N-1\}$, to achieve the optimal attenuation.

The problem (A.5) can be challenging to solve to the optimum [25]. Moreover, in practice, due to the inevitable uncertainty in modeling the system, the robust controller that achieves the optimal attenuation level can be too sensitive to the model uncertainty to use. Hence, a reasonable surrogate

of the optimal LQ disturbance attenuation problem is the following: Given a $\gamma > \gamma^*$, find a control policy $\mu_t = -K_t x_t$, for all $t \in \{0, \cdots, N-1\}$, that solves

$$\min_{\{\mu_t\}} \mathcal{J}(\{\mu_t(x_t)\}) := \mathbb{E}_{\{\xi_t\}} \left[ \sum_{t=0}^{N-1} \left( x_t^\top Q_t x_t + u_t^\top R_t u_t \right) + x_N^\top Q_N x_N \right] \quad \text{s.t.} \sup_{x_0, \{w_t\}} \frac{\left\{ \mathcal{C}(\{\mu_t(x_t)\}, \{w_t\}) \right\}^{1/2}}{\|\omega\|} < \gamma, \tag{A.6}$$

where $Q_t := C_t^\top C_t$ for $t \in \{0, \cdots, N-1\}$, $\mathcal{J}(\{\mu_t(x_t)\})$ is the LQG cost of the system $x_{t+1} = A_t x_t + B_t u_t + D_t \xi_t$, and $\xi_t \sim \mathcal{N}(0, I)$ is independent Gaussian noise with unit intensity [30]. Note that the LQ disturbance attenuation problem in (A.6), and particularly with $x_0 = 0$, is also called the mixed $\mathcal{H}_2/\mathcal{H}_\infty$ problem [35, 30, 68, 16], when the system is time-invariant, and the horizon of the problem $N$ is $\infty$. To solve (A.6) using policy optimization methods, we introduce the time-varying RDE:

$$P_{K_t} = Q_t + K_t^\top R_t K_t + (A_t - B_t K_t)^\top \widetilde{P}_{K_{t+1}} (A_t - B_t K_t), \quad t \in \{0, \cdots, N-1\}, \quad P_{K_N} = Q_N, \tag{A.7}$$

where $\widetilde{P}_{K_{t+1}} := P_{K_{t+1}} + P_{K_{t+1}} (\gamma^2 I - D_t^\top P_{K_{t+1}} D_t)^{-1} P_{K_{t+1}}$. Then, two upper bounds of $\mathcal{J}$ and their corresponding PGs can be expressed in terms of solutions to (A.7) (if exist) [30]. In particular, (A.8) below is closely related to the objective function of LEQG in (A.1), and (A.9) is connected to the objective function of zero-sum LQ dynamic games to be introduced shortly.

**Lemma A.2** *(Closed-Forms for the Objective Function and PGs) For any sequence of control gains $\{K_t\}$ such that (A.7) generates a sequence of p.s.d. solutions $\{P_{K_{t+1}}\}$ satisfying $\gamma^2 I - D_t^\top P_{K_{t+1}} D_t > 0$, for all $t \in \{0, \cdots, N-1\}$, and $\gamma^2 I - P_{K_0} > 0$, then two common upper bounds of the objective function $\mathcal{J}$ in (A.6) are:*

$$\overline{\mathcal{J}}(\{K_t\}) = -\gamma^2 \log \det(I - \gamma^{-2} P_{K_0}) - \gamma^2 \sum_{t=1}^{N} \log \det(I - \gamma^{-2} P_{K_t} D_{t-1} D_{t-1}^\top), \tag{A.8}$$

$$\overline{\mathcal{J}}(\{K_t\}) = \text{Tr}(P_{K_0}) + \sum_{t=1}^{N} \text{Tr}(P_{K_t} D_{t-1} D_{t-1}^\top). \tag{A.9}$$

*The PGs of (A.8) and (A.9) at time $t$ can be expressed as*

$$\nabla_{K_t} \overline{\mathcal{J}}(\{K_t\}) := \frac{\partial \overline{\mathcal{J}}(\{K_t\})}{\partial K_t} = 2 \left[ (R_t + B_t^\top \widetilde{P}_{K_{t+1}} B_t) K_t - B_t^\top \widetilde{P}_{K_{t+1}} A_t \right] \Sigma_{K_t}, \quad t \in \{0, \cdots, N-1\}, \tag{A.10}$$

*where $\Sigma_t$, $t \in \{0, \cdots, N-1\}$, for the PGs of (A.8) and (A.9) are expressed, respectively, as*

$$\Sigma_{K_t} := \prod_{i=0}^{t-1} \left[ (I - \gamma^{-2} P_{K_{i+1}} D_i D_i^\top)^{-\top} (A_i - B_i K_i) \right] \cdot (I - \gamma^{-2} P_{K_0})^{-1} \cdot \prod_{i=0}^{t-1} \left[ (A_i - B_i K_i)^\top (I - \gamma^{-2} P_{K_{i+1}} D_i D_i^\top)^{-1} \right]$$

$$+ \sum_{\tau=1}^{t} \left\{ \prod_{i=\tau}^{t-1} \left[ (I - \gamma^{-2} P_{K_{i+1}} D_i D_i^\top)^{-\top} (A_i - B_i K_i) \right] \cdot D_{\tau-1} (I - \gamma^{-2} D_{\tau-1}^\top P_{K_\tau} D_{\tau-1})^{-1} D_{\tau-1}^\top \right.$$

$$\left. \cdot \prod_{i=\tau}^{t-1} \left[ (A_i - B_i K_i)^\top (I - \gamma^{-2} P_{K_{i+1}} D_i D_i^\top)^{-1} \right] \right\},$$

$$\Sigma_{K_t} := \prod_{i=0}^{t-1} \left[ (I - \gamma^{-2} P_{K_{i+1}} D_i D_i^\top)^{-\top} (A_i - B_i K_i)(A_i - B_i K_i)^\top (I - \gamma^{-2} P_{K_{i+1}} D_i D_i^\top)^{-1} \right]$$

$$+ \sum_{\tau=1}^{t} \left\{ \prod_{i=\tau}^{t-1} \left[ (I - \gamma^{-2} P_{K_{i+1}} D_i D_i^\top)^{-\top} (A_i - B_i K_i) \right] \cdot D_{\tau-1} D_{\tau-1}^\top \right.$$

$$\left. \cdot \prod_{i=\tau}^{t-1} \left[ (A_i - B_i K_i)^\top (I - \gamma^{-2} P_{K_{i+1}} D_i D_i^\top)^{-1} \right] \right\}.$$

The proof of Lemma A.2 is deferred to §B.2. Before moving on to the next section, we provide a remark regarding the challenges of addressing LEQG or the LQ disturbance attenuation problem directly using derivative-free PG methods.

**Remark A.3** *(Challenges) It seems tempting to tackle LEQG or the LQ disturbance attenuation problem directly using derivative-free PG methods, e.g., vanilla PG or natural PG methods by sampling the system trajectories as in [11]. However, there are several challenges. First, when attempting to sample the LEQG objective* (A.1) *using system trajectories (and then to estimate PGs), the bias of gradient estimates can be difficult to control uniformly with a fixed sample size per iterate, as the* log *function is not globally Lipschitz[2]. Moreover, for the vanilla PG method, even with exact PG accesses, the iterates may not preserve a certain level of risk-sensitivity/disturbance attenuation, i.e., remain feasible, along the iterations, which can make the disturbance input drive the cost to arbitrarily large values (see the numerical examples in [16] for the infinite-horizon LTI setting). This failure will only be exacerbated in the derivative-free setting with accesses to only noisy PG estimates. Lastly, developing a derivative-free natural PG method (which we will show in §3.3 that it can preserve a prescribed attenuation level along the iterations) is also challenging, because the expressions of $\Sigma_{K_t}$ in Lemmas A.1 and A.2 cannot be sampled from system trajectories directly. Therefore, the game-theoretic approach to be introduced next is rather one workaround.*

### A.3.3 An Equivalent Dynamic Game Formulation

Lastly, and more importantly, we describe an equivalent dynamic game formulation to the LEQG and the LQ disturbance attenuation problems introduced above. Under certain conditions to be introduced in Lemma A.6, the saddle-point gain matrix of the minimizing player in the dynamic game (if exists) also addresses the LEQG and the LQ disturbance attenuation problem, providing an alternative route to overcome the challenges reported in Remark A.3. Specifically, we consider a zero-sum LQ stochastic dynamic game (henceforth, game) model with closed-loop perfect-state information pattern, which can also be viewed as a benchmark setting of MARL for two competing agents [69, 70, 15, 39, 71], as the role played by LQR for single-agent RL [3]. The linear time-varying system dynamics follow

$$x_{t+1} = A_t x_t + B_t u_t + D_t w_t + \xi_t, \quad t \in \{0, \cdots, N-1\}, \tag{A.11}$$

where $x_t \in \mathbb{R}^m$ is the system state, $u_t := \mu_{u,t}(h_{u,t}) \in \mathbb{R}^d$ (resp., $w_t := \mu_{w,t}(h_{w,t}) \in \mathbb{R}^n$) is the control input of the minimizing (resp., maximizing) player[3] and $\mu_{u,t} : (\mathbb{R}^m \times \mathbb{R}^d)^t \times \mathbb{R}^m \to \mathbb{R}^d$ (resp., $\mu_{w,t} : (\mathbb{R}^m \times \mathbb{R}^n)^t \times \mathbb{R}^m \to \mathbb{R}^n$) is the control policy that maps, at time $t$, the history of state-control pairs up to time $t$, and state at time $t$, denoted as $h_{u,t} := \{x_0, \cdots, x_t, u_0, \cdots, u_{t-1}\}$ (resp., $h_{w,t} := \{x_0, \cdots, x_t, w_0, \cdots, w_{t-1}\}$), to the control $u_t$ (resp., $w_t$). The independent process noises are denoted by $\xi_t \sim \mathcal{D}$ and $A_t, B_t, D_t$ are system matrices with proper dimensions. Further, we assume $x_0 \sim \mathcal{D}$ to be independent of the process noises, and the distribution $\mathcal{D}$ has zero mean and a positive-definite covariance. We also assume that $\|x_0\|, \|\xi_t\| \leq \vartheta$, for all $t \in \{0, \cdots, N-1\}$, almost surely, for some constant $\vartheta$[4]. The goal of the minimizing (resp. maximizing) player is to minimize (resp. maximize) a quadratic objective function, namely to solve the zero-sum game

$$\inf_{\{u_t\}} \sup_{\{w_t\}} \mathbb{E}_{x_0, \xi_0, \cdots, \xi_{N-1}} \left[ \sum_{t=0}^{N-1} \left( x_t^\top Q_t x_t + u_t^\top R_t^u u_t - w_t^\top R_t^w w_t \right) + x_N^\top Q_N x_N \right], \tag{A.12}$$

where $Q_t \geq 0$, $R_t^u, R_t^w > 0$ are symmetric weighting matrices and the system transitions follow (A.11). Whenever the solution to (A.12) exists such that the inf and sup operators in (A.12) are interchangeable, then the value (A.12) is the *value* of the game, and the corresponding policies of the players are known as saddle-point policies. To characterize the solution to (A.12), we first introduce the following time-varying RDE:

$$P_t^* = Q_t + A_t^\top P_{t+1}^* \Lambda_t^{-1} A_t, \quad t \in \{0, \cdots, N-1\}, \tag{A.13}$$

---

[2]One workaround for mitigating this bias might be to remove the log operator in the objective (A.1), and only minimize the terms after the $\mathbb{E}\exp(\cdot)$ function. However, it is not clear yet if derivative-free methods provably converge for this objective, as exp function is neither globally smooth nor Lipschitz. We have left this direction in our future work, as our goal in the present work is to provide a unified way to solve all three classes of problems.

[3]Hereafter we will use player and agent interchangeably.

[4]The assumption on the boundedness of distribution is only for ease of analysis [19, 18]. Extensions to sub-Gaussian distributions are standard and do not affect the sample complexity result, as noted by [19].

where $\Lambda_t := \boldsymbol{I} + \left(B_t(R_t^u)^{-1}B_t^\top - D_t(R_t^w)^{-1}D_t^\top\right)P_{t+1}^*$ and $P_N^* = Q_N$. For the stochastic game with closed-loop perfect-state information pattern that we considered, we already know from [26] that whenever a saddle point exists, the saddle-point control policies are linear state-feedback (i.e., $\mu_{u,t}^*(h_{u,t}) = -K_t^* x_t$ and $\mu_{w,t}^*(h_{w,t}) = -L_t^* x_t$), and the gain matrices $K_t^*$ and $L_t^*$, for $t \in \{0, \cdots, N-1\}$, are unique and can be expressed by

$$K_t^* = (R_t^u)^{-1}B_t^\top P_{t+1}^* \Lambda_t^{-1} A_t, \qquad (A.14) \qquad\qquad L_t^* = -(R_t^w)^{-1}D_t^\top P_{t+1}^* \Lambda_t^{-1} A_t, \qquad (A.15)$$

where $P_{t+1}^*$, $t \in \{0, \cdots, N-1\}$, is the sequence of p.s.d. matrices generated by (A.13). We next introduce a standard assumption that suffices to ensure the existence of the value of the game, following from Theorem 3.2 of [25] and Theorem 6.7 of [26].

**Assumption A.4** $R_t^w - D_t^\top P_{t+1}^* D_t > 0$, *for all* $t \in \{0, \cdots, N-1\}$, *where* $P_{t+1}^* \geq 0$ *is generated by* (A.13).

Under Assumption A.4, the value in (A.12) is attained by the controller sequence $\left(\{-K_t^* x_t^*\}, \{-L_t^* x_t^*\}\right)$, where $K_t^*$ and $L_t^*$ are given in (A.14) and (A.15), respectively, and $x_{t+1}^*$ is the corresponding state trajectory generated by $x_{t+1}^* = \Lambda_t^{-1} A_t x_t^* + \xi_t$, for $t \in \{0, \cdots, N-1\}$ and with $x_0^* = x_0$ [25]. Thus, the solution to (A.12) can be found by searching $K_t^*$ and $L_t^*$ in $\mathbb{R}^{d \times m} \times \mathbb{R}^{n \times m}$, for all $t \in \{0, \cdots, N-1\}$. The resulting minimax policy optimization problem is

$$\min_{\{K_t\}} \max_{\{L_t\}} \mathcal{G}\left(\{K_t\}, \{L_t\}\right) := \mathbb{E}_{x_0, \xi_0, \cdots, \xi_{N-1}} \left[ \sum_{t=0}^{N-1} x_t^\top \left(Q_t + K_t^\top R_t^u K_t - L_t^\top R_t^w L_t\right) x_t + x_N^\top Q_N x_N \right], \quad (A.16)$$

subject to the system transition $x_{t+1} = (A_t - B_t K_t - D_t L_t)x_t + \xi_t$, for $t \in \{0, \cdots, N-1\}$. Some further notes regarding Assumption A.4 are provided in the remark below.

**Remark A.5** *(Unique Feedback Nash (Saddle-point) Equilibrium) By [25, 26], Assumption 2.1 (A.4) is sufficient to guarantee the* existence *of feedback Nash (equivalently, saddle-point) equilibrium in zero-sum LQ dynamic games (game), under which it is also unique. Besides sufficiency, Assumption 2.1 (A.4) is also "almost necessary", and "quite tight" as noted in Remark 6.8 of [26]. In particular, if the sequence of matrices $R_t^w - D_t^\top P_{t+1}^* D_t$ for $t \in \{0, \cdots, N-1\}$ in Assumption 2.1 (A.4) admits any negative eigenvalues, then the upper value of the game becomes unbounded. In the context of LEQG and LQ disturbance attenuation (LQDA), suppose we set the system parameters of LEQG, LQDA, and game according to Lemma 2.2 (A.6), and in particular set $\beta^{-1}\boldsymbol{I}$ in LEQG, $\gamma^2\boldsymbol{I}$ in LQDA, and $\boldsymbol{R}^w$ in the game to be the same. Then these three problems are equivalent as far as their optimum solutions go (which is what we seek). Due to this equivalence, Assumption 2.1 (A.4) is a sufficient and "almost necessary" condition for the existence of a solution to the equivalent LEQG/LQDA. Specifically, if the sequence of matrices $R_t^w - D_t^\top P_{t+1}^* D_t$ for $t \in \{0, \cdots, N-1\}$ in Assumption 2.1 (A.4) admits any negative eigenvalues, then no state-feedback controller can achieve a $\gamma$-level of disturbance attenuation in the equivalent LQDA, and no state-feedback controller can achieve a $\beta$-degree of risk-sensitivity in the equivalent LEQG.*

Lastly, we formally state the well-known equivalence conditions between LEQG, LQ disturbance attenuation, and zero-sum LQ stochastic dynamic games in the following lemma, and provide a short proof in §B.3.

**Lemma A.6** *(Connections) For any fixed $\gamma > \gamma^*$ in the LQ disturbance attenuation problem, we can introduce an equivalent LEQG problem and an equivalent zero-sum LQ dynamic game. Specifically, if we set $\beta^{-1}\boldsymbol{I}, R_t, C_t^\top C_t, W$ in LEQG, $\gamma^2\boldsymbol{I}, R_t, C_t^\top C_t, D_t D_t^\top$ in the LQ disturbance attenuation problem, and $R_t^w, R_t^u, Q_t, D_t D_t^\top$ in the game to be the same, for all $t \in \{0, \cdots, N-1\}$, then the optimal gain matrices in LEQG, the gain matrix in the LQ disturbance attenuation problem, and the saddle-point gain matrix for the minimizing player in the game are the same.*

By the above connections and Remark A.3, we will hereafter focus on the stochastic zero-sum LQ dynamic game between a minimizing controller and a maximizing disturbance, using PG methods with exact gradient accesses in §3, and derivative-free PG methods by sampling system trajectories in §4. As a result, the minimizing controller we obtain solves all three classes of problems introduced above altogether.

## A.4 Compact Formulations

In this section, we re-derive the game formulations in §2.3 (and equivalently, §A.3.3), with linear state-feedback policies for the two players, using the following compact notations:

$$\boldsymbol{x} = \left[x_0^\top, \cdots, x_N^\top\right]^\top, \quad \boldsymbol{u} = \left[u_0^\top, \cdots, u_{N-1}^\top\right]^\top, \quad \boldsymbol{w} = \left[w_0^\top, \cdots, w_{N-1}^\top\right]^\top, \quad \boldsymbol{\xi} = \left[x_0^\top, \xi_0^\top, \cdots, \xi_{N-1}^\top\right]^\top,$$

$$\boldsymbol{A} = \begin{bmatrix} \mathbf{0}_{m \times mN} & \mathbf{0}_{m \times m} \\ diag(A_0, \cdots, A_{N-1}) & \mathbf{0}_{mN \times m} \end{bmatrix}, \quad \boldsymbol{B} = \begin{bmatrix} \mathbf{0}_{m \times dN} \\ diag(B_0, \cdots, B_{N-1}) \end{bmatrix}, \quad \boldsymbol{D} = \begin{bmatrix} \mathbf{0}_{m \times nN} \\ diag(D_0, \cdots, D_{N-1}) \end{bmatrix},$$

$$\boldsymbol{Q} = \left[diag(Q_0, \cdots, Q_N)\right], \quad \boldsymbol{R}^u = \left[diag(R_0^u, \cdots, R_{N-1}^u)\right], \quad \boldsymbol{R}^w = \left[diag(R_0^w, \cdots, R_{N-1}^w)\right],$$

$$\boldsymbol{K} = \left[diag(K_0, \cdots, K_{N-1}) \quad \mathbf{0}_{dN \times m}\right], \quad \boldsymbol{L} = \left[diag(L_0, \cdots, L_{N-1}) \quad \mathbf{0}_{nN \times m}\right].$$

With the above definitions, the game in §2.3 is characterized by the transition dynamics

$$\boldsymbol{x} = \boldsymbol{A}\boldsymbol{x} + \boldsymbol{B}\boldsymbol{u} + \boldsymbol{D}\boldsymbol{w} + \boldsymbol{\xi} = (\boldsymbol{A} - \boldsymbol{B}\boldsymbol{K} - \boldsymbol{D}\boldsymbol{L})\boldsymbol{x} + \boldsymbol{\xi}, \tag{A.17}$$

where $\boldsymbol{x}$ is the concatenated system state, $\boldsymbol{u} = -\boldsymbol{K}\boldsymbol{x}$ (respectively, $\boldsymbol{w} = -\boldsymbol{L}\boldsymbol{x}$) is the linear state-feedback controller of the minimizing (resp., maximizing) player, and $\boldsymbol{A}, \boldsymbol{B}, \boldsymbol{D}$ are the system matrices in their corresponding compact forms. Note that $\boldsymbol{A}$ is essentially a nilpotent matrix with degree of $N+1$ (i.e. a lower triangular matrix with zeros along the main diagonal) such that $\boldsymbol{A}^{N+1} = \mathbf{0}$. Also, $\boldsymbol{\xi}$ is a vector that concatenates the independently sampled initial state $x_0 \in \mathcal{D}$ and process noises $\xi_t \in \mathcal{D}, t \in \{0, \cdots, N-1\}$. The objective of the minimizing (resp. maximizing) player is to minimize (resp. maximize) the objective function re-written in terms of the compact notation, that is to solve

$$\inf_{\boldsymbol{K}} \sup_{\boldsymbol{L}} \mathcal{G}(\boldsymbol{K}, \boldsymbol{L}) := \mathbb{E}_{\boldsymbol{\xi}} \left[ \boldsymbol{x}^\top \left( \boldsymbol{Q} + \boldsymbol{K}^\top \boldsymbol{R}^u \boldsymbol{K} - \boldsymbol{L}^\top \boldsymbol{R}^w \boldsymbol{L} \right) \boldsymbol{x} \right] \tag{A.18}$$

subject to the transition dynamics (A.17).

**An Illustrative Example:** Let us now consider a scalar example with $N, n, m, d = 1$ with the following concatenated notations:

$$\boldsymbol{x} = \begin{bmatrix} x_0 \\ x_1 \end{bmatrix}, \quad \boldsymbol{\xi} = \begin{bmatrix} x_0 \\ \xi_0 \end{bmatrix}, \quad (\boldsymbol{u}, \boldsymbol{w}, \boldsymbol{R}^u, \boldsymbol{R}^w) = (u_0, w_0, R_0^u, R_0^w), \quad \boldsymbol{A} = \begin{bmatrix} 0 & 0 \\ A_0 & 0 \end{bmatrix}, \quad \boldsymbol{B} = \begin{bmatrix} 0 \\ B_0 \end{bmatrix}, \quad \boldsymbol{D} = \begin{bmatrix} 0 \\ D_0 \end{bmatrix},$$

$$\boldsymbol{Q} = \begin{bmatrix} Q_0 & 0 \\ 0 & Q_1 \end{bmatrix}, \quad \boldsymbol{K} = \begin{bmatrix} K_0 & 0 \end{bmatrix}, \quad \boldsymbol{L} = \begin{bmatrix} L_0 & 0 \end{bmatrix}.$$

Then, the system dynamics of the game and the linear state-feedback controllers are represented as

$$\boldsymbol{x} = \boldsymbol{A}\boldsymbol{x} + \boldsymbol{B}\boldsymbol{u} + \boldsymbol{D}\boldsymbol{w} + \boldsymbol{\xi} \iff \begin{bmatrix} x_0 \\ x_1 \end{bmatrix} = \begin{bmatrix} 0 & 0 \\ A_0 & 0 \end{bmatrix} \cdot \begin{bmatrix} x_0 \\ x_1 \end{bmatrix} + \begin{bmatrix} 0 \\ B_0 \end{bmatrix} \cdot u_0 + \begin{bmatrix} 0 \\ D_0 \end{bmatrix} \cdot w_0 + \begin{bmatrix} x_0 \\ \xi_0 \end{bmatrix},$$

$$\boldsymbol{u} = -\boldsymbol{K}\boldsymbol{x} \iff u_0 = -\begin{bmatrix} K_0 & 0 \end{bmatrix} \cdot \begin{bmatrix} x_0 \\ x_1 \end{bmatrix}, \qquad \boldsymbol{w} = -\boldsymbol{L}\boldsymbol{x} \iff w_0 = -\begin{bmatrix} L_0 & 0 \end{bmatrix} \cdot \begin{bmatrix} x_0 \\ x_1 \end{bmatrix}$$

$$\boldsymbol{x} = (\boldsymbol{A} - \boldsymbol{B}\boldsymbol{K} - \boldsymbol{D}\boldsymbol{L})\boldsymbol{x} + \boldsymbol{\xi} \iff \begin{bmatrix} x_0 \\ x_1 \end{bmatrix} = \left( \begin{bmatrix} 0 & 0 \\ A_0 & 0 \end{bmatrix} - \begin{bmatrix} 0 \\ B_0 \end{bmatrix} \cdot \begin{bmatrix} K_0 & 0 \end{bmatrix} - \begin{bmatrix} 0 \\ D_0 \end{bmatrix} \cdot \begin{bmatrix} L_0 & 0 \end{bmatrix} \right) \cdot \begin{bmatrix} x_0 \\ x_1 \end{bmatrix} + \begin{bmatrix} x_0 \\ \xi_0 \end{bmatrix}$$

$$\iff \begin{bmatrix} x_0 \\ x_1 \end{bmatrix} = \begin{bmatrix} 0 & 0 \\ A_0 - B_0 K_0 - D_0 L_0 & 0 \end{bmatrix} \cdot \begin{bmatrix} x_0 \\ x_1 \end{bmatrix} + \begin{bmatrix} x_0 \\ \xi_0 \end{bmatrix} = \begin{bmatrix} x_0 \\ (A_0 - B_0 K_0 - D_0 L_0)x_0 + \xi_0 \end{bmatrix}.$$

Moreover, we define $A_{K_0, L_0} := A_0 - B_0 K_0 - D_0 L_0$ and establish the equivalence between the recursive Lyapunov equation (3.3) and its compact form (3.4) such that

$$\boldsymbol{P}_{\boldsymbol{K}, \boldsymbol{L}} = (\boldsymbol{A} - \boldsymbol{B}\boldsymbol{K} - \boldsymbol{D}\boldsymbol{L})^\top \boldsymbol{P}_{\boldsymbol{K}, \boldsymbol{L}} (\boldsymbol{A} - \boldsymbol{B}\boldsymbol{K} - \boldsymbol{D}\boldsymbol{L}) + \boldsymbol{Q} + \boldsymbol{K}^\top \boldsymbol{R}^u \boldsymbol{K} - \boldsymbol{L}^\top \boldsymbol{R}^w \boldsymbol{L}$$

$$\iff \begin{bmatrix} P_{K_0, L_0} & 0 \\ 0 & P_{K_1, L_1} \end{bmatrix} = \begin{bmatrix} 0 & 0 \\ A_{K_0, L_0} & 0 \end{bmatrix}^\top \begin{bmatrix} P_{K_0, L_0} & 0 \\ 0 & P_{K_1, L_1} \end{bmatrix} \begin{bmatrix} 0 & 0 \\ A_{K_0, L_0} & 0 \end{bmatrix} + \begin{bmatrix} Q_0 + K_0^\top R_0^u K_0 - L_0 R_0^w L_0 & 0 \\ 0 & Q_1 \end{bmatrix}$$

$$\iff \begin{bmatrix} P_{K_0, L_0} & 0 \\ 0 & P_{K_1, L_1} \end{bmatrix} = \begin{bmatrix} A_{K_0, L_0}^\top P_{K_1, L_1} A_{K_0, L_0} + Q_0 + K_0^\top R_0^u K_0 - L_0 R_0^w L_0 & 0 \\ 0 & Q_1 \end{bmatrix}.$$

Next, we note that the solution to (3.5), $\Sigma_{K,L}$, is also a recursive formula in the compact form, such that

$$\Sigma_{K,L} = (A - BK - DL)\Sigma_{K,L}(A - BK - DL)^\top + \Sigma_0$$

$$\Longleftrightarrow \mathbb{E}\begin{bmatrix} x_0 x_0^\top & 0 \\ 0 & x_1 x_1^\top \end{bmatrix} = \begin{bmatrix} 0 & 0 \\ A_{K_0,L_0} & 0 \end{bmatrix} \mathbb{E}\begin{bmatrix} x_0 x_0^\top & 0 \\ 0 & x_1 x_1^\top \end{bmatrix} \begin{bmatrix} 0 & 0 \\ A_{K_0,L_0} & 0 \end{bmatrix}^\top + \mathbb{E}\begin{bmatrix} x_0 x_0^\top & 0 \\ 0 & \xi_0 \xi_0^\top \end{bmatrix}$$

$$\Longleftrightarrow \mathbb{E}\begin{bmatrix} x_0 x_0^\top & 0 \\ 0 & x_1 x_1^\top \end{bmatrix} = \mathbb{E}\begin{bmatrix} x_0 x_0^\top & 0 \\ 0 & A_{K_0,L_0} x_0 x_0^\top A_{K_0,L_0}^\top + \xi_0 \xi_0^\top \end{bmatrix}.$$

This completes the illustrative example.

## A.5 Extended Discussions

**Remark A.7** *(Double-Loop Scheme) Our double-loop scheme, which has also been used before in [15, 39], is pertinent to the descent-multi-step-ascent scheme [72] and the gradient descent-ascent (GDA) scheme with two-timescale stepsizes [51, 73, 74] for solving nonconvex-(non)concave minimax optimization problems, with the number of multi-steps of ascent going to infinity, or the ratio between the fast and slow stepsizes going to infinity (the $\tau$-GDA scheme with $\tau \to \infty$ [51, 74]). Another variant is the alternating GDA (AGDA) scheme, which has been investigated in [53] to address nonconvex-nonconcave problems with two-sided PL condition. However, unlike this literature, one key condition for establishing convergence, the* global smoothness *of the objective function, does not hold in our control setting with* unbounded *decision spaces and objective functions. In fact, as per Lemma 3.3, a non-judicious choice of $K$ may lead to an undefined inner-loop maximization problem with unbounded objective. Hence, one has to carefully control the update-rule here, to ensure that the iterates do not yield unbounded/undefined values along iterations. As we will show in §D.4, it is not hard to construct cases where descent-multi-step-ascent/AGDA/$\tau$-GDA diverges even with infinitesimal stepsizes, similar to the negative results reported in the infinite-horizon setting in [60]. These results suggest that it seems challenging to provably show that other candidate update schemes converge to the global NE in zero-sum LQ games, except the double-loop one, as we will show next.*

**Remark A.8** *(IR) Suppose that the initial control gain matrix satisfies $K_0 \in \mathcal{K}$. Then Lemma A.6 shows that $K_0$ is the control gain matrix that attains a $\gamma$-level of disturbance attenuation. By the implicit regularization property in Theorem 3.7, every iterate $K_k \in \mathcal{K}$ for all $k \geq 0$ following the NPG (3.17) or the GN (3.18) update rules will thus preserve this $\gamma$-level of disturbance attenuation throughout the policy optimization (learning) process. Theorem 3.7 thus provides some provable robustness guarantees for two specific policy search directions, (3.17) and (3.18), which is important for safety-critical control systems in the presence of adversarial disturbances, since otherwise, the system performance index can be driven to arbitrarily large values.*

**Remark A.9** *When solving LEQG and LQ disturbance attenuation (LQDA) problems using the proposed double-loop derivative-free PG methods (cf. Algorithms 1 and 2), we exploited the equivalence relationships in Lemma 2.2 to construct and solve an equivalent zero-sum LQ game. We comment on how to construct such an equivalent game problem in the model-free setting where only oracle-level accesses to the LEQG and LQDA models are available.*

*Suppose that one would like to solve the LQDA problem by solving the equivalent zero-sum game; then, the only information needed is oracle-level accesses to the LQDA model (cf. §2.2). In particular, for a fixed sequence of gains $K_t$ and any sequence of disturbances $w_t$, we assume that this LQDA oracle can return the $\|z\|$ as defined in §2.2. Then, the LQDA oracle with an additionally injected sequence of independent Gaussian noise (with any positive definite covariance matrix) suffices to serve as the oracle for our double-loop derivative-free PG algorithms (for the stochastic game). Therefore, no explicit knowledge on the system parameters $(A_t, B_t, D_t, C_t, E_t, \gamma)$ is needed.*

*However, if one would like to solve the LEQG problem by solving an equivalent zero-sum game, one will need knowledge of $W$ (to build-up the black-box sampling oracle/simulator, but still, the exact value of $W$ is not revealed to the learning agent) in addition to oracle-level accesses to the original LEQG model (cf. 2.1). Also, explicit knowledge of the parameters $(A_t, B_t, Q_t, R_t, \beta)$ is not required. We would like to note that the assumption on the knowledge (and/or the availability of the*

*estimate) of W is reasonable for a large family of control applications where system dynamics and disturbance can be studied separately. For example, consider the risk-sensitive control of a wind turbine. The turbine dynamics and the wind information can be gathered separately. One can identify W by looking at the past wind data. Similar situations hold for many aerospace applications where the properties of process noise (e.g. wind gust) can be estimated beforehand.*

## B Proofs of Main Results

### B.1 Proof of Lemma A.1

**Proof** Since $x_0 \sim \mathcal{N}(\mathbf{0}, X_0)$, $w_0, \cdots, w_{N-1} \sim \mathcal{N}(\mathbf{0}, W)$, $P_{K_t} \geq 0$ for all $t$, $X_0^{-1} - \beta P_{K_0} > 0$, and $W^{-1} - \beta P_{K_t} > 0$, for all $t \in \{1, \cdots, N\}$, we have by Lemma C.1 of [16] that the objective function of LEQG can be represented by (A.3). Also, the conditions that $X_0^{-1} - \beta P_{K_0} > 0$ and $W^{-1} - \beta P_{K_t} > 0$, for all $t \in \{0, \cdots, N\}$, ensure the existence of the expression $\Sigma_{K_t}$, which can be verified through applying matrix inversion lemma as in (B.1). Thus, the existence of the expression $\nabla_{K_t} \mathcal{J}(\{K_t\})$ is also ensured, for all $t \in \{0, \cdots, N-1\}$. Let us now use $\nabla_{K_t}$ to denote the derivative w.r.t. $K_t$. Then, for $t \in \{1, \cdots, N-1\}$, we have

$$
\nabla_{K_t} \mathcal{J}(\{K_t\}) = -\frac{1}{\beta} \text{Tr}\left\{ (I - \beta P_{K_0} X_0)^{-\top} \cdot \left[ \nabla_{K_t} (I - \beta P_{K_\tau} X_0) \right]^\top \right\} - \frac{1}{\beta} \sum_{\tau=1}^{t} \text{Tr}\left\{ (I - \beta P_{K_\tau} W)^{-\top} \cdot \left[ \nabla_{K_t} (I - \beta P_{K_\tau} W) \right]^\top \right\}
$$

$$
= \text{Tr}\left[ (I - \beta P_{K_0} X_0)^{-1} \cdot \nabla_{K_t} (P_{K_0} X_0) \right] + \sum_{\tau=1}^{t} \text{Tr}\left[ (I - \beta P_{K_\tau} W)^{-1} \cdot \nabla_{K_t} (P_{K_\tau} W) \right]
$$

$$
= \text{Tr}\left[ \nabla_{K_t} P_{K_0} \cdot X_0 (I - \beta P_{K_0} X_0)^{-1} \right] + \sum_{\tau=1}^{t} \text{Tr}\left[ \nabla_{K_t} P_{K_\tau} \cdot W (I - \beta P_{K_\tau} W)^{-1} \right]
$$

$$
= \text{Tr}\left[ \nabla_{K_t} P_{K_0} \cdot X_0^{\frac{1}{2}} (I - \beta X_0^{\frac{1}{2}} P_{K_0} X_0^{\frac{1}{2}})^{-1} X_0^{\frac{1}{2}} \right] + \sum_{\tau=1}^{t} \text{Tr}\left[ \nabla_{K_t} P_{K_\tau} \cdot W^{\frac{1}{2}} (I - \beta W^{\frac{1}{2}} P_{K_\tau} W^{\frac{1}{2}})^{-1} W^{\frac{1}{2}} \right],
$$

where the first equality is due to $\nabla_X \log \det X = X^{-\top}$, the chain rule, and the fact that $P_{K_\tau}$, for all $\tau \in \{t+1, \cdots, N\}$, is independent of $K_t$. The second and third equalities are due to the cyclic property of matrix trace, $\text{Tr}(A^\top B^\top) = \text{Tr}(AB)$, and that $X_0, W$ are independent of $K_t$. The last equality uses matrix inversion lemma, such that for $V \in \{X_0, W\}$:

$$
V \cdot (I - \beta P_K V)^{-1} = V^{\frac{1}{2}} (I - \beta V^{\frac{1}{2}} P_K V^{\frac{1}{2}})^{-1} V^{\frac{1}{2}} \tag{B.1}
$$

Then, by the definition of the RDE in (A.2) and defining $M_0 := X_0^{\frac{1}{2}} (I - \beta X_0^{\frac{1}{2}} P_{K_0} X_0^{\frac{1}{2}})^{-1} X_0^{\frac{1}{2}}$, $M_t := W^{\frac{1}{2}} (I - \beta W^{\frac{1}{2}} P_{K_t} W^{\frac{1}{2}})^{-1} W^{\frac{1}{2}}$ for all $t \in \{1, \cdots, N-1\}$, we have:

$$
\forall t \in \{0, \cdots, N-1\}, \quad \nabla_{K_t} \mathcal{J}(\{K_t\}) = \sum_{\tau=0}^{t} \text{Tr}\left[ \nabla_{K_t} P_{K_\tau} \cdot M_\tau \right]
$$

$$
= \sum_{\tau=0}^{t-1} \text{Tr}\left[ \nabla_{K_t} P_{K_\tau} \cdot M_\tau \right] + 2\left[ (R_t + B_t^\top \widetilde{P}_{K_{t+1}} B_t) K_t - B_t^\top \widetilde{P}_{K_{t+1}} A_t M_t \right]
$$

$$
+ \text{Tr}\left[ (A_t - B_t K_t)^\top \cdot \nabla_{K_t} \widetilde{P}_{K_{t+1}} \cdot (A_t - B_t K_t) M_t \right]
$$

$$
= \sum_{\tau=0}^{t-1} \text{Tr}\left[ \nabla_{K_t} P_{K_\tau} \cdot M_\tau \right] + 2\left[ (R_t + B_t^\top \widetilde{P}_{K_{t+1}} B_t) K_t - B_t^\top \widetilde{P}_{K_{t+1}} A_t M_t \right],
$$

where the last equality is due to $\widetilde{P}_{K_{t+1}}$ not depending on $K_t$ and thus $\nabla_{K_t} \widetilde{P}_{K_{t+1}} = \mathbf{0}$. Then, for all $\tau \in \{1, \cdots, t-1\}$, we first recall the definition that $\widetilde{P}_{K_\tau} = P_{K_\tau} + \beta P_{K_\tau} (W^{-1} - \beta P_{K_\tau})^{-1} P_{K_\tau}$, and then

take the derivatives on both sides with respect to $K_t$ to get

$$\nabla_{K_t}\widetilde{P}_{K_\tau} = \nabla_{K_t}\left[P_{K_\tau} + \beta P_{K_\tau}(W^{-1} - \beta P_{K_\tau})^{-1}P_{K_\tau}\right] = \nabla_{K_t}\left[(I - \beta P_{K_\tau}W)^{-1}P_{K_\tau}\right]$$

$$= (I - \beta P_{K_\tau}W)^{-1}\cdot\nabla_{K_t}P_{K_\tau}\cdot\beta W(I - \beta P_{K_\tau}W)^{-1}P_{K_\tau} + (I - \beta P_{K_\tau}W)^{-1}\cdot\nabla_{K_t}P_{K_\tau}$$

$$= (I - \beta P_{K_\tau}W)^{-1}\cdot\nabla_{K_t}P_{K_\tau}\cdot\left[W^{\frac{1}{2}}(\beta^{-1}I - W^{\frac{1}{2}}P_{K_\tau}W^{\frac{1}{2}})^{-1}W^{\frac{1}{2}}P_{K_\tau} + I\right]$$

$$= (I - \beta P_{K_\tau}W)^{-1}\cdot\nabla_{K_t}P_{K_\tau}\cdot(I - \beta WP_{K_\tau})^{-1} = (I - \beta P_{K_\tau}W)^{-1}\cdot\nabla_{K_t}P_{K_\tau}\cdot(I - \beta P_{K_\tau}W)^{-\top},$$

where the second and the fifth equalities use matrix inversion lemma, the third equality uses $\nabla_X(P^{-1}) = -P^{-1}\cdot\nabla_X P\cdot P^{-1}$, and the fourth equality uses (B.1). Now, for any $\tau \in \{0,\cdots,t-1\}$, we can iteratively unroll the RDE (A.2) to obtain that

$$\text{Tr}\left[\nabla_{K_t}P_{K_\tau}\cdot M_\tau\right] = 2\left[(R_t + B_t^\top\widetilde{P}_{K_{t+1}}B_t)K_t - B_t^\top\widetilde{P}_{K_{t+1}}A_t\right]$$

$$\cdot\prod_{i=\tau}^{t-1}\left[(I - \beta P_{K_{i+1}}W)^{-\top}(A_i - B_iK_i)\right]\cdot M_\tau\cdot\prod_{i=\tau}^{t-1}\left[(A_i - B_iK_i)^\top(I - \beta P_{K_{i+1}}W)^{-1}\right].$$

Taking a summation proves the PG expression in (A.4). ∎

## B.2 Proof of Lemma A.2

**Proof** We first prove the policy gradient of (A.8). By the conditions that $\gamma^2 I - D_t^\top P_{K_{t+1}}D_t > 0$, for all $t \in \{0,\cdots,N-1\}$, and $\gamma^2 I - P_{K_0} > 0$, the expression $\Sigma_{K_t}$ exists and thus the expression $\nabla_{K_t}\overline{\mathcal{J}}(\{K_t\})$ exists, for all $t \in \{0,\cdots,N-1\}$. Then, the PG expression of (A.8) follows from the proof of Lemma A.1 by replacing $\beta^{-1}$ and $W$ therein with $\gamma^2$ and $D_tD_t^\top$, respectively, and also replacing the $X_0$ in the expression of $\Sigma_{K_t}$ by $I$. Next, we prove the PG of (A.9) w.r.t. $K_t$. Let us now use $\nabla_{K_t}$ to denote the derivative w.r.t. $K_t$. Then, for $t \in \{0,\cdots,N-1\}$, we have

$$\nabla_{K_t}\overline{\mathcal{J}}(\{K_t\}) = \text{Tr}\left[\nabla_{K_t}P_{K_0}\right] + \sum_{\tau=1}^{t}\text{Tr}\left[\nabla_{K_t}(P_{K_\tau}D_{\tau-1}D_{\tau-1}^\top)\right] = \text{Tr}\left[\nabla_{K_t}P_{K_0}\right] + \sum_{\tau=1}^{t}\text{Tr}\left[D_{\tau-1}^\top\cdot\nabla_{K_t}P_{K_\tau}\cdot D_{\tau-1}\right],$$

where the first equality is due to that $P_{K_\tau}$, for all $\tau \in \{t+1,\cdots,N\}$, does not depend on $K_t$. Recalling the RDE in (A.7), we can unroll the recursive formula to obtain

$$\text{Tr}\left[\nabla_{K_t}P_{K_0}\right] = 2\left[(R_t + B_t^\top\widetilde{P}_{K_{t+1}}B_t)K_t - B_t^\top\widetilde{P}_{K_{t+1}}A_t\right]$$

$$\cdot\prod_{i=0}^{t-1}\left[(I - \gamma^{-2}P_{K_{i+1}}D_iD_i^\top)^{-\top}(A_i - B_iK_i)(A_i - B_iK_i)^\top(I - \gamma^{-2}P_{K_{i+1}}D_iD_i^\top)^{-1}\right].$$

Moreover, for any $\tau \in \{1,\cdots,t\}$, we have

$$\text{Tr}\left[D_{\tau-1}^\top\cdot\nabla_{K_t}P_{K_\tau}\cdot D_{\tau-1}\right] = 2\left[(R_t + B_t^\top\widetilde{P}_{K_{t+1}}B_t)K_t - B_t^\top\widetilde{P}_{K_{t+1}}A_t\right]$$

$$\cdot\prod_{i=\tau}^{t-1}\left[(I - \gamma^{-2}P_{K_{i+1}}D_iD_i^\top)^{-\top}(A_i - B_iK_i)\right]\cdot D_{\tau-1}D_{\tau-1}^\top\cdot\prod_{i=\tau}^{t-1}\left[(A_i - B_iK_i)^\top(I - \gamma^{-2}P_{K_{i+1}}D_iD_i^\top)^{-1}\right].$$

Taking a summation proves the expression of the policy gradient in (A.10). ∎

## B.3 Proof of Lemmas 2.2 and A.6

**Proof** The equivalence relationships are proved in three parts. Firstly, by Chapter 6.4, page 306 of [26], the discrete-time zero-sum LQ dynamic game with additive stochastic disturbance (zero-mean, independent sequence), and with closed-loop perfect-state information pattern, admits the same saddle-point gain matrices for the minimizer and the maximizer as the deterministic version of the game (that is, without the stochastic driving term).

Subsequently, by Theorem 4.1 of [24], we can introduce a corresponding (deterministic) zero-sum LQ dynamic game with $R_t^w = \gamma^2 I$ for any fixed $\gamma > \gamma^*$ in the LQ disturbance attenuation problem. Then,

if we replace $R_t^w, R_t^u, Q_t$ in the deterministic version of the game presented in §2.3 by $\gamma^2 I, R_t, C_t^\top C_t$ in §2.2, respectively, for all $t \in \{0, \cdots, N-1\}$, and also set other shared parameters in §2.3 and §2.2 to be the same, the saddle-point gain matrix for the minimizing player in that game is equivalent to the gain matrix in the disturbance attenuation problem.

Lastly, by §VI.B of [22] and if we replace $R_t^w, R_t^u, Q_t, D_t D_t^\top$ in the deterministic version of the game presented in §2.3 by $\beta^{-1} I, R_t, C_t^\top C_t, W$ in §2.1 for all $t \in \{0, \cdots, N-1\}$, and also set other shared parameters in §2.3 to be the same as the ones in §2.1, then the deterministic zero-sum LQ dynamic game is equivalent to the LEQG problem with perfect state measurements, in the sense that the optimal gain matrix in the latter is the same as the saddle-point gain matrix for the minimizing player in the former. This completes the proof. ∎

## B.4 Proof of Lemma 3.1

**Proof** We first prove that for a fixed $K \in \mathcal{K}$, the optimization problem with respect to $L$ can be nonconcave, by establishing an example of the game. Here, $\mathcal{K}$ is the feasible set as defined in (3.9), which ensures that for a fixed $K$, the solution to the maximization problem with respect to $L$ is well defined. Specifically, we consider a 3-dimensional linear time-invariant system with $A_t = A$, $B_t = B$, $D_t = D$, $Q_t = Q$, $R_t^u = R^u$, and $R_t^w = R^w$ for all $t$, where

$$A = \begin{bmatrix} 1 & 0 & -5 \\ -1 & 1 & 0 \\ 0 & 0 & 1 \end{bmatrix}, \quad B = \begin{bmatrix} 1 & -10 & 0 \\ 0 & 3 & 1 \\ -1 & 0 & 2 \end{bmatrix}, \quad D = \begin{bmatrix} 0.5 & 0 & 0 \\ 0 & 0.2 & 0 \\ 0 & 0 & 0.2 \end{bmatrix},$$

$$Q = \begin{bmatrix} 2 & -1 & 0 \\ -1 & 2 & -1 \\ 0 & -1 & 2 \end{bmatrix}, \quad R^u = \begin{bmatrix} 4 & -1 & 0 \\ -1 & 4 & -2 \\ 0 & -2 & 3 \end{bmatrix}, \quad R^w = 5 \cdot I.$$

Also, we set $\Sigma_0 = I$, $N = 5$, and choose the time-invariant control gain matrices $K_t = K$, $L_t^1 = L^1$, $L_t^2 = L^2$, $L_t^3 = L^3 = \frac{L^1 + L^2}{2}$ for all $t$, where

$$K = \begin{bmatrix} -0.12 & -0.01 & 0.62 \\ -0.21 & 0.14 & 0.15 \\ -0.06 & 0.05 & 0.42 \end{bmatrix}, \quad L^1 = \begin{bmatrix} -0.86 & 0.97 & 0.14 \\ -0.82 & 0.36 & 0.51 \\ 0.98 & 0.08 & -0.20 \end{bmatrix}, \quad L^2 = \begin{bmatrix} -0.70 & -0.37 & 0.09 \\ -0.54 & -0.28 & 0.23 \\ 0.74 & 0.62 & -0.51 \end{bmatrix}.$$

The concatenated matrices $A, B, D, Q, R^u, R^w, K, L^1, L^2, L^3$ are generated following the definitions in §3. Then, we first note that $K \in \mathcal{K}$ holds, as $P_{K, L(K)} \geq 0$ exists and one can calculate that $\lambda_{\min}(R^w - D^\top P_{K, L(K)} D) = 0.5041 > 0$. Subsequently, we can verify that $(\mathcal{G}(K, L^1) + \mathcal{G}(K, L^2))/2 - \mathcal{G}(K, L^3) = 6.7437 > 0$. Thus, we can conclude that there exists a $K \in \mathcal{K}$ such that the objective function is nonconcave with respect to $L$. Next, we prove the other argument by choosing the following time-invariant control gain matrices $L_t = L$, $K_t^1 = K^1$, $K_t^2 = K^2$, $K_t^3 = K^3$ for all $t$, where

$$L = \mathbf{0}, \quad K^1 = \begin{bmatrix} 1.44 & 0.31 & -1.18 \\ 0.03 & -0.13 & -0.39 \\ 0.36 & -1.71 & 0.24 \end{bmatrix}, \quad K^2 = \begin{bmatrix} -0.08 & -0.16 & -1.96 \\ -0.13 & -1.12 & 1.28 \\ 1.67 & -0.91 & 1.71 \end{bmatrix}, \quad K^3 = \frac{K^1 + K^2}{2}.$$

Following similar steps, we can verify that $(\mathcal{G}(K^1, L) + \mathcal{G}(K^2, L))/2 - \mathcal{G}(K^3, L) = -1.2277 \times 10^5 < 0$. Therefore, there exists $L$ such that the objective function is nonconvex with respect to $K$. ∎

## B.5 Proof of Lemma 3.2

**Proof** For $t \in \{0, \cdots, N-1\}$, we can define the cost-to-go function starting from time $t$ to $N$ as $\mathcal{G}_t$. Recalling the definition of $P_{K_t, L_t}$ in (3.3), we can write $\mathcal{G}_t$ as

$$\mathcal{G}_t = \mathbb{E}_{x_t, \xi_t, \ldots, \xi_{N-1}} \left\{ x_t^\top P_{K_t, L_t} x_t + \sum_{\tau=t}^{N-1} \xi_\tau^\top P_{K_{\tau+1}, L_{\tau+1}} \xi_\tau \right\} = \mathbb{E}_{x_t} \left\{ x_t^\top P_{K_t, L_t} x_t \right\} + \mathbb{E}_{\xi_t, \ldots, \xi_{N-1}} \left\{ \sum_{\tau=t}^{N-1} \xi_\tau^\top P_{K_{\tau+1}, L_{\tau+1}} \xi_\tau \right\}$$

$$= \mathbb{E}_{x_t} \left\{ x_t^\top (Q_t + K_t^\top R_t^u K_t - L_t^\top R_t^w L_t) x_t + x_t^\top (A_t - B_t K_t - D_t L_t)^\top P_{K_{t+1}, L_{t+1}} (A_t - B_t K_t - D_t L_t) x_t \right\}$$

$$+ \mathbb{E}_{\xi_t, \ldots, \xi_{N-1}} \left\{ \sum_{\tau=t}^{N-1} \xi_\tau^\top P_{K_{\tau+1}, L_{\tau+1}} \xi_\tau \right\}.$$

We denote $\nabla_{L_t} \mathcal{G}_t := \frac{\partial \mathcal{G}_t}{\partial L_t}$ for notational simplicity, and note that the last term of the above equation does not depend on $L_t$. Then, we can show that $\nabla_{L_t} \mathcal{G}_t = 2 \left[ (-R_t^w + D_t^\top P_{K_{t+1}, L_{t+1}} D_t) L_t - D_t^\top P_{K_{t+1}, L_{t+1}} (A_t - B_t K_t) \right] \mathbb{E}_{x_t} \{ x_t x_t^\top \}$. By writing the gradient of each time step compactly and taking expectations over $x_0$ and the additive noises $\{\xi_0, \ldots, \xi_{N-1}\}$ along the trajectory, we can derive the exact PG of $\mathcal{G}(K, L)$ with respect to $L$ as $\nabla_L \mathcal{G}(K, L) = 2 \left[ (-R^w + D^\top P_{K,L} D) L - D^\top P_{K,L} (A - BK) \right] \Sigma_{K,L}$, where $\Sigma_{K,L}$ is the solution to (3.5). Using similar techniques, we can derive, for a fixed $L$, the PG of $\mathcal{G}(K, L)$ with respect to $K$ to be $\nabla_K \mathcal{G}(K, L) = 2 \left[ (R^u + B^\top P_{K,L} B) K - B^\top P_{K,L} (A - DL) \right] \Sigma_{K,L}$. Since $\Sigma_0$ being full-rank ensures that $\Sigma_{K,L}$ is also full-rank and we have $\nabla_K \mathcal{G}(K, L) = \nabla_L \mathcal{G}(K, L) = 0$ at the stationary points, the PG forms derived above imply

$$K = (R^u + B^\top P_{K,L} B)^{-1} B^\top P_{K,L} (A - DL), \qquad L = (-R^w + D^\top P_{K,L} D)^{-1} D^\top P_{K,L} (A - BK), \quad \text{(B.2)}$$

where by assumption, we have $R^w - D^\top P_{K,L} D > 0$, and $R^u + B^\top P_{K,L} B > 0$ is further implied by $P_{K,L} \geq 0$ (and thus they are both invertible). Therefore, we can solve (B.2) to obtain

$$K = (R^u)^{-1} B^\top P_{K,L} \left[ I + \left( B(R^u)^{-1} B^\top - D(R^w)^{-1} D^\top \right) P_{K,L} \right]^{-1} A,$$

$$L = -(R^w)^{-1} D^\top P_{K,L} \left[ I + \left( B(R^u)^{-1} B^\top - D(R^w)^{-1} D^\top \right) P_{K,L} \right]^{-1} A,$$

which are (A.14) and (A.15) in their corresponding compact forms. Moreover, at the Nash equilibrium, the solution to (A.13) in its compact form, denoted as $P^*$, is also the solution to the Lyapunov equation

$$P^* = Q + (K^*)^\top R^u K^* - (L^*)^\top R^w L^* + (A - BK^* - DL^*)^\top P^* (A - BK^* - DL^*).$$

Lastly, under Assumption A.4, the solution to the RDE (A.13) is uniquely computed. As a result, the stationary point of $\mathcal{G}(K, L)$ is also the unique Nash equilibrium of the game. This completes the proof. ∎

## B.6 Proof of Lemma 3.3

**Proof** Let the sequence of outer-loop control gains $\{K_t\}$, $t \in \{0, \cdots, N-1\}$, be fixed. To facilitate the analysis, we first construct a sequence of auxiliary zero-sum LQ dynamic games, denoted as $\{\Gamma_{d_t}\}$, $t \in \{0, \cdots, N-1\}$, where each $\Gamma_{d_t}$ has closed-loop state-feedback information structure. Specifically, for any $t$, $\Gamma_{d_t}$ starts from some arbitrary $\overline{x}_0$ and follows a deterministic system dynamics $\overline{x}_{\tau+1} = \overline{A}_\tau \overline{x}_\tau + \overline{B}_\tau \overline{u}_\tau + \overline{D}_\tau \overline{w}_\tau$ for a finite horizon of $N - t$, where

$$\overline{A}_\tau := A_{t+\tau} - B_{t+\tau} K_{t+\tau}, \quad \overline{B}_\tau := 0, \quad \overline{D}_\tau := D_{t+\tau}, \quad \overline{u}_\tau = -\overline{K}_\tau \overline{x}_\tau = -K_{t+\tau} \overline{x}_\tau, \quad \overline{w}_\tau = -\overline{L}_\tau \overline{x}_\tau.$$

The weighting matrices of $\Gamma_{d_t}$ are chosen to be $\overline{Q}_\tau := Q_{t+\tau}$, $\overline{R}_\tau^u := R_{t+\tau}^u$, and $\overline{R}_\tau^w := R_{t+\tau}^w$ for all $\tau \in \{0, \cdots, N-t-1\}$ and $\overline{Q}_{N-t} = Q_N$. Moreover, for any sequence of control gains $\{\overline{L}_\tau\}$, $\tau \in \{0, \cdots, N-t-1\}$ in $\Gamma_{d_t}$, we define $\{\overline{P}_{\overline{L}_\tau}\}$, $\tau \in \{0, \cdots, N-t\}$, as the sequence of solutions generated by the recursive Lyapunov equation

$$\overline{P}_{\overline{L}_\tau} = \overline{Q}_\tau + \overline{K}_\tau^\top \overline{R}_\tau^u \overline{K}_\tau - \overline{L}_\tau^\top \overline{R}_\tau^w \overline{L}_\tau + (\overline{A}_\tau - \overline{D}_\tau \overline{L}_\tau)^\top \overline{P}_{\overline{L}_{\tau+1}} (\overline{A}_\tau - \overline{D}_\tau \overline{L}_\tau), \quad \overline{P}_{\overline{L}_{N-t}} = \overline{Q}_{N-t}. \quad \text{(B.3)}$$

Then, for any $t \in \{0, \cdots, N-1\}$, Theorem 3.2 of [25] suggests that $\Gamma_{d_t}$ admits a unique feedback saddle-point if the sequence of p.s.d. solutions $\{\overline{P}_\tau\}$, $\tau \in \{0, \cdots, N-t\}$, computed by the recursive Riccati equation

$$\overline{P}_\tau = \overline{Q}_\tau + \overline{K}_\tau^\top \overline{R}_\tau^u \overline{K}_\tau + \overline{A}_\tau^\top \overline{P}_{\tau+1} \overline{A}_\tau + \overline{A}_\tau^\top \overline{P}_{\tau+1} \overline{D}_\tau (\overline{R}_\tau^w - \overline{D}_\tau^\top \overline{P}_{\tau+1} \overline{D}_\tau)^{-1} \overline{D}_\tau^\top \overline{P}_{\tau+1} \overline{A}_\tau, \quad \overline{P}_{N-t} = \overline{Q}_{N-t}. \tag{B.4}$$

exists and satisfies

$$\overline{R}_\tau^w - \overline{D}_\tau^\top \overline{P}_{\tau+1} \overline{D}_\tau > 0, \quad \tau \in \{0, \cdots, N-t-1\}. \tag{B.5}$$

Whenever exist, the saddle-point solutions at time $\tau$, for any $\tau \in \{0, \cdots, N-t-1\}$, are given by

$$\overline{u}_\tau^* = \mathbf{0}, \quad \overline{w}_\tau^* = -\overline{L}_\tau^* \overline{x}_\tau = (\overline{R}_\tau^w - \overline{D}_\tau^\top \overline{P}_{\tau+1} \overline{D}_\tau)^{-1} \overline{D}_\tau^\top \overline{P}_{\tau+1} \overline{A}_\tau \overline{x}_\tau. \tag{B.6}$$

Moreover, the saddle-point value of $\Gamma_{d_t}$ is $\overline{x}_0^\top \overline{P}_0 \overline{x}_0$ and the upper value of $\Gamma_{d_t}$ becomes unbounded when the matrix $\overline{R}_\tau^w - \overline{D}_\tau^\top \overline{P}_{\tau+1} \overline{D}_\tau$ has at least one negative eigenvalue for at least one $\tau \in \{0, \cdots, N-t-1\}$. Further, since the choice of $\overline{x}_0$ was arbitrary, it holds that $\overline{P}_0 \geq \overline{P}_{\overline{L}_0}$ for any sequence of $\{\overline{L}_\tau\}$, $\tau \in \{0, \cdots, N-t-1\}$. Because the above properties hold for all $\Gamma_{d_t}$, $t \in \{0, \cdots, N-1\}$, we can generate a compact matrix $P_{K,L(K)} \in \mathbb{R}^{m(N+1) \times m(N+1)}$ by putting $\overline{P}_0$ for each $\Gamma_{d_t}$, $t \in \{0, \cdots, N-1\}$, sequentially on the diagonal and putting $Q_N$ at the end of the diagonal. The resulting $P_{K,L(K)}$ then solves (3.8), and satisfies $P_{K,L(K)} \geq P_{K,L}$, for all $L \in \mathcal{S}(n, m, N)$, where $P_{K,L}$ is computed by (3.4).

Subsequently, we construct an auxiliary zero-sum LQ dynamic game $\Gamma_s$ with closed-loop state-feedback information pattern, using compact notations. In particular, $\Gamma_s$ has a horizon of $N$ and a (stochastic) system dynamics being $\overline{x} = \overline{A}\overline{x} + \overline{B}\overline{u} + \overline{D}\overline{w} + \overline{\xi}$, where the initial state is some arbitrary $\overline{x}_0$ and

$$\overline{A} := A - BK, \quad \overline{B} := \mathbf{0}, \quad \overline{D} := D, \quad \overline{\xi} := \left[x_0^\top, \xi_0^\top, \cdots, \xi_{N-1}^\top\right]^\top, \quad \overline{u} = -\overline{K}\overline{x}, \quad \overline{w} = -\overline{L}\overline{x}.$$

We assume that $x_0, \xi_0, \cdots, \xi_{N-1} \sim \mathcal{D}$ are zero-mean and independent random variables. The weighting matrices of $\Gamma_s$ are chosen to be $\overline{Q} := Q$, $\overline{R}^u := R^u$, and $\overline{R}^w := R^w$. By Corollary 6.4 of [26], $\Gamma_s$ is equivalent to $\Gamma_{d_0}$ above in that the Riccati equation (B.4) and the saddle-point gain matrices (B.6) are the same for $\Gamma_s$ and $\Gamma_{d_0}$.

Note that for a fixed outer-loop control policy $\mathbf{u} = -K\mathbf{x}$, the inner-loop subproblem is the same as $\Gamma_s$, in that $\Gamma_s$ essentially absorbs the fixed outer-loop control input into system dynamics. Therefore, the properties of $\Gamma_s$, which are equivalent to those of $\Gamma_{d_0}$, apply to our inner-loop subproblem. Specifically, (3.8) is a compact representation of (B.4) and (3.9) is the set of $K$ such that (B.4) admits a sequence of p.s.d. solutions satisfying (B.5). Therefore, $K \in \mathcal{K}$ is a sufficient and almost necessary condition for the solution to the inner-loop to be well defined, in that if $K \notin \overline{\mathcal{K}}$, the inner-loop objective $\mathcal{G}(K, \cdot)$ can be driven to arbitrarily large values. Whenever the solution to the inner-loop exists, it is unique and takes the form of (3.10). This completes the proof. ∎

## B.7 Proof of Lemma 3.4

**Proof** Let $K \in \mathcal{K}$ be fixed. First, the nonconcavity of the inner-loop objective $\mathcal{G}(K, \cdot)$ follows directly from Lemma 3.1. Then, we have by (3.9) and the definitions of the compact matrices that $P_{K_t,L(K_t)} \geq 0$ exists and satisfies $R_t^w - D_t^\top P_{K_{t+1},L(K_{t+1})} D_t > 0$, for all $t \in \{0, \cdots, N-1\}$. Also, for any sequence of $\{L_t\}$, the sequence of solutions to the recursive Lyapunov equation (3.3), $\{P_{K_t,L_t}\}$, always exists and is unique: by Lemma 3.3, it holds that $P_{K_t,L(K_t)} \geq P_{K_t,L_t}$ and thus $R_t^w - D_t^\top P_{K_{t+1},L_{t+1}} D_t > 0$ for all $t \in \{0, \cdots, N-1\}$. Now, for any $t \in \{0, \cdots, N-1\}$, we have by (3.3) that

$$\begin{aligned}
P_{K_t,L_t} &= (A_t - B_t K_t - D_t L_t)^\top P_{K_{t+1},L_{t+1}} (A_t - B_t K_t - D_t L_t) + Q_t + K_t^\top R_t^u K_t - L_t^\top R_t^w L_t \\
&= (A_t - B_t K_t)^\top P_{K_{t+1},L_{t+1}} (A_t - B_t K_t) + Q_t + K_t^\top R_t^u K_t - (A_t - B_t K_t)^\top P_{K_{t+1},L_{t+1}} D_t L_t \\
&\quad - L_t^\top D_t^\top P_{K_{t+1},L_{t+1}} (A_t - B_t K_t) - L_t^\top (R_t^w - D_t^\top P_{K_{t+1},L_{t+1}} D_t) L_t. \tag{B.7}
\end{aligned}$$

Note that as $\|L_t\| \to +\infty$, the quadratic term in (B.7), with leading matrix $-(R_t^w - D_t^\top P_{K_{t+1},L_{t+1}} D_t) < 0$, dominates other terms. Thus, as $\|L_t\| \to +\infty$ for some $t \in \{0, \cdots, N-1\}$, $\lambda_{\min}(P_{K_t,L_t}) \to -\infty$,

which further makes $\lambda_{\min}(P_{K,L}) \to -\infty$. In compact forms, this means that $\lambda_{\min}(P_{K,L}) \to -\infty$ as $\|L\| \to +\infty$. By (3.2), the inner-loop objective for a fixed $K \in \mathcal{K}$ and any $L \in \mathcal{S}(n,m,N)$ has the form of $\mathcal{G}(K,L) = \mathrm{Tr}(P_{K,L}\Sigma_0)$, where $\Sigma_0 > 0$ is full-rank. This proves the coercivity of the inner-loop objective $\mathcal{G}(K,\cdot)$.

Moreover, for a fixed $K \in \mathcal{K}$, as $P_{K,L}$ is a polynomial of $L$, $\mathcal{G}(K,\cdot)$ is continuous in $L$. Combined with the coercivity property and the upper-boundedness of $\mathcal{G}(K,\cdot)$ for $K \in \mathcal{K}$, we can conclude the compactness of the superlevel set (3.12). The local Lipschitz and smoothness properties of $\mathcal{G}(K,\cdot)$ follow from Lemmas 4 and 5 of [19], with their $A, B, Q, R, K$ matrices replaced by our $A - BK, D, Q + K^\top R^u K, -R^w, L$, respectively. Note that the property $\|\Sigma_K\| \le \frac{C(K)}{\sigma_{\min}(Q)}$ in the proof of Lemma 16 in [19] does not hold in our setting as we only assume $Q \ge 0$ and $Q + K^\top R^u K$ may not be full-rank. Instead, we utilize the fact that $\mathcal{L}_K(a)$ is compact, and thus there exists a uniform constant $c_{\Sigma,a} := \max_{L \in \mathcal{L}_K(a)} \|\Sigma_{K,L}\|$ such that $\|\Sigma_{K,L}\| \le c_{\Sigma,a}$ for all $L \in \mathcal{L}_K(a)$. Subsequently, we note that for all $L \in \mathcal{L}_K(a)$, the operator norms of the PG (3.13) and the Hessian, which are continuous functions of $L$, can be uniformly bounded by some constants $l_{K,a} > 0$ and $\psi_{K,a} > 0$, respectively. That is,

$$l_{K,a} := \max_{L \in \mathcal{L}_K(a)} \left\| \nabla_L \mathcal{G}(K,L) \right\|, \quad \psi_{K,a} := \max_{L \in \mathcal{L}_K(a)} \left\| \nabla_L^2 \mathcal{G}(K,L) \right\| = \max_{L \in \mathcal{L}_K(a)} \sup_{\|X\|_F = 1} \left\| \nabla_L^2 \mathcal{G}(K,L)[X,X] \right\|,$$

where $\nabla_L^2 \mathcal{G}(K,L)[X,X]$ denotes the action of the Hessian on a matrix $X \in \mathcal{S}(n,m,N)$. The expression of $\nabla_L^2 \mathcal{G}(K,L)[X,X]$ follows from Proposition 3.10 of [14], with their $R, B, X, Y, A_K$ being replaced by our $-R^w, D, P_{K,L}, \Sigma_{K,L}, A - BK - DL$, respectively. This proves the global Lipschitzness and smoothness of $\mathcal{G}(K,\cdot)$ over the compact superlevel set $\mathcal{L}_K(a)$.

To prove the PL condition, we first characterize the difference between $P_{K,L}$ and $P_{K,L(K)}$ as

$$P_{K,L(K)} - P_{K,L} = A_{K,L(K)}^\top (P_{K,L(K)} - P_{K,L})A_{K,L(K)} + (L(K) - L)^\top E_{K,L}$$
$$+ E_{K,L}^\top (L(K) - L) - (L(K) - L)^\top H_{K,L}(L(K) - L),$$

where $A_{K,L(K)} := A - BK - DL(K)$. By Lemma 3.3 and $K \in \mathcal{K}$, we can conclude that for all $L$, the inequality $H_{K,L} > 0$ holds because $P_{K,L(K)} \ge P_{K,L}$ for all $L \in \mathcal{S}(n,m,N)$. Then, by Proposition 2.1(b) of [14], we have that for every $\varphi > 0$,

$$(L(K) - L)^\top E_{K,L} + E_{K,L}^\top (L(K) - L) \le \frac{1}{\varphi}(L(K) - L)^\top (L(K) - L) + \varphi E_{K,L}^\top E_{K,L}.$$

Choosing $\varphi = 1/\lambda_{\min}(H_{K,L})$ yields

$$(L(K) - L)^\top E_{K,L} + E_{K,L}^\top (L(K) - L) - (L(K) - L)^\top H_{K,L}(L(K) - L) \le \frac{E_{K,L}^\top E_{K,L}}{\lambda_{\min}(H_{K,L})} \le \frac{E_{K,L}^\top E_{K,L}}{\lambda_{\min}(H_{K,L(K)})},$$
(B.8)

where the last inequality is due to $H_{K,L(K)} \le H_{K,L}, \forall L \in \mathcal{S}(n,m,N)$. Let $Y$ be the solution to $Y = A_{K,L(K)}^\top Y A_{K,L(K)} + \frac{E_{K,L}^\top E_{K,L}}{\lambda_{\min}(H_{K,L(K)})}$. From (B.8), we further know that $P_{K,L(K)} - P_{K,L} \le Y$ and $Y = \sum_{t=0}^N \left[ A_{K,L(K)}^\top \right]^t \frac{E_{K,L}^\top E_{K,L}}{\lambda_{\min}(H_{K,L(K)})} \left[ A_{K,L(K)} \right]^t$. Therefore, letting $\Sigma_{K,L(K)} = \sum_{t=0}^N [A_{K,L(K)}]^t \Sigma_0 [A_{K,L(K)}^\top]^t$, we have

$$\mathcal{G}(K,L(K)) - \mathcal{G}(K,L) = \mathrm{Tr}\left( \Sigma_0 (P_{K,L(K)} - P_{K,L}) \right) \le \mathrm{Tr}(\Sigma_0 Y)$$

$$= \mathrm{Tr}\left( \Sigma_0 \left( \sum_{t=0}^N \left[ A_{K,L(K)}^\top \right]^t \frac{E_{K,L}^\top E_{K,L}}{\lambda_{\min}(H_{K,L(K)})} \left[ A_{K,L(K)} \right]^t \right) \right) \le \frac{\|\Sigma_{K,L(K)}\|}{\lambda_{\min}(H_{K,L(K)})} \mathrm{Tr}(E_{K,L}^\top E_{K,L}) \quad \text{(B.9)}$$

$$\le \frac{\|\Sigma_{K,L(K)}\|}{\lambda_{\min}(H_{K,L(K)})\lambda_{\min}^2(\Sigma_{K,L})} \mathrm{Tr}(\Sigma_{K,L}^\top E_{K,L}^\top E_{K,L} \Sigma_{K,L}) \le \frac{\|\Sigma_{K,L(K)}\|}{4\lambda_{\min}(H_{K,L(K)})\phi^2} \mathrm{Tr}(\nabla_L \mathcal{G}(K,L)^\top \nabla_L \mathcal{G}(K,L)).$$

Inequality (B.9) follows from the cyclic property of matrix trace and the last inequality is due to $\Sigma_0 \le \Sigma_{K,L}$. Hence,

$$\mathcal{G}(K,L(K)) - \mathcal{G}(K,L) \le \frac{\|\Sigma_{K,L(K)}\|}{4\lambda_{\min}(H_{K,L(K)})\phi^2} \mathrm{Tr}(\nabla_L \mathcal{G}(K,L)^\top \nabla_L \mathcal{G}(K,L)).$$

Substituting in $\mu_K := 4\lambda_{\min}(H_{K,L(K)})\phi^2/\|\Sigma_{K,L(K)}\|$ completes the proof. ∎

## B.8 Proof of Lemma 3.5

**Proof** The nonconvexity proof is done by constructing a time-invariant example, which chooses $\Sigma_0 = I$ and $N = 5$. We further choose the system matrices to be $A_t = A$, $B_t = B$, $D_t = D$, $Q_t = Q$, $R_t^u = R^u$, and $R_t^w = R^w$ for all $t$, where

$$A = \begin{bmatrix} 1 & 0 & -5 \\ -1 & 1 & 0 \\ 0 & 0 & 1 \end{bmatrix}, \quad B = \begin{bmatrix} 1 & -10 & 0 \\ 0 & 3 & 1 \\ -1 & 0 & 2 \end{bmatrix}, \quad D = \begin{bmatrix} 0.5 & 0 & 0 \\ 0 & 0.5 & 0 \\ 0 & 0 & 0.5 \end{bmatrix}$$

$$Q = \begin{bmatrix} 3 & -1 & 0 \\ -1 & 2 & -1 \\ 0 & -1 & 1 \end{bmatrix}, \quad R^u = \begin{bmatrix} 2 & 1 & 1 \\ 1 & 3 & -1 \\ 1 & -1 & 3 \end{bmatrix},$$

and $R^w = 7.22543 \cdot I$. We also choose the time-invariant gain matrices $K_t^1 = K^1$, $K_t^2 = K^2$, and $K_t^3 = K^3$, where

$$K^1 = \begin{bmatrix} -0.8750 & 1.2500 & -2.5000 \\ -0.1875 & 0.1250 & 0.2500 \\ -0.4375 & 0.6250 & -0.7500 \end{bmatrix}, \quad K^2 = \begin{bmatrix} -0.8786 & 1.2407 & -2.4715 \\ -0.1878 & 0.1237 & 0.2548 \\ -0.4439 & 0.5820 & -0.7212 \end{bmatrix}, \quad K^3 = \frac{K^1 + K^2}{2}.$$

The concatenated matrices $A, B, D, Q, R^u, R^w, K^1, K^2, K^3$ are generated following the definitions in §3. Subsequently, we can prove that $K^1, K^2, K^3 \in \mathcal{K}$ by verifying that the recursive Riccati equation (3.8) yields p.s.d. solutions for $K^1, K^2, K^3$, respectively, and $\lambda_{\min}(H_{K^1,L(K^1)}) = 4.3496 \times 10^{-6}$, $\lambda_{\min}(H_{K^2,L(K^2)}) = 0.1844$, $\lambda_{\min}(H_{K^3,L(K^3)}) = 0.0926$. Then, we can compute that $\left(\mathcal{G}(K^1, L(K^1)) + \mathcal{G}(K^2, L(K^2))\right)/2 - \mathcal{G}(K^3, L(K^3)) = -0.0224 < 0$. Thus, we can conclude that $\mathcal{G}(K, L(K))$ is nonconvex in $K$. Subsequently, we show that the outer loop is noncoercive on $\mathcal{K}$ by a scalar example with $N = 2$ and time-invariant system matrices being $A_t = 2$, $B_t = D_t = Q_t = R_t^u = 1$, and $R_t^w = 5$, for all $t$. Then, we consider the gain matrix $K^\epsilon = \begin{bmatrix} 2-\epsilon & 0 & 0 \\ 0 & 2-\epsilon & 0 \end{bmatrix} \in \mathcal{K}$, $\forall \epsilon \in \left(0, \frac{16}{9}\right)$. It can be observed that $\lim_{\epsilon \to 0+} \mathcal{G}(K^\epsilon, L(K^\epsilon)) = 11 < \infty$, while $\lim_{\epsilon \to 0+} K^\epsilon \in \partial\mathcal{K}$, for $\epsilon \in (0, \frac{16}{9})$. Therefore, $\mathcal{G}(K, L(K))$ is not coercive. Lastly, by Lemma 3.2, the stationary point of the outer loop in $\mathcal{K}$, denoted as $(K^*, L(K^*))$, is unique and constitutes the unique Nash equilibrium of the game. ∎

## B.9 Proof of Theorem 3.6

**Proof** We prove the global convergence of three inner-loop PG updates as follows:

**PG:** For a fixed $K \in \mathcal{K}$ and an arbitrary $L_0$ that induces a finite $\mathcal{G}(K, L_0)$, we first define superlevel sets $\mathcal{L}_K(\mathcal{G}(K, L_0))$ and $\mathcal{L}_K(a)$ as in (3.12), where $a < \mathcal{G}(K, L_0)$ is an arbitrary constant. Clearly, it holds that $L_0 \in \mathcal{L}_K(\mathcal{G}(K, L_0)) \subset \mathcal{L}_K(a)$ and thus $\mathcal{L}_K(\mathcal{G}(K, L_0))$ is nonempty as well as compact (as shown before in Lemma 3.4). Next, denote the closure of the complement of $\mathcal{L}_K(a)$ as $\overline{(\mathcal{L}_K(a))^c}$, which is again nonempty and also disjoint with $\mathcal{L}_K(\mathcal{G}(K, L_0))$, i.e., $\mathcal{L}_K(\mathcal{G}(K, L_0)) \cap \overline{(\mathcal{L}_K(a))^c} = \varnothing$, due to $a$ being strictly less than $\mathcal{G}(K, L_0)$. Hence, one can deduce (see for example Lemma A.1 of [75]) that there exists a Hausdorff distance $\delta_a > 0$ between $\mathcal{L}_K(\mathcal{G}(K, L_0))$ and $\overline{(\mathcal{L}_K(a))^c}$ such that for a given $L \in \mathcal{L}_K(\mathcal{G}(K, L_0))$, all $L'$ satisfying $\|L' - L\|_F \leq \delta_a$ also satisfy $L' \in \mathcal{L}_K(a)$.

Now, since $L_0 \in \mathcal{L}_K(\mathcal{G}(K, L_0))$, we have $\|\nabla_L \mathcal{G}(K, L_0)\|_F \leq l_{K,\mathcal{G}(K,L_0)}$, where $l_{K,\mathcal{G}(K,L_0)}$ is the global Lipschitz constant over $\mathcal{L}_K(\mathcal{G}(K, L_0))$. Therefore, it suffices to choose $\eta_0 \leq \frac{\delta_a}{l_{K,\mathcal{G}(K,L_0)}}$ to ensure that the one-step "fictitious" PG update satisfies $L_0 + \eta_0 \nabla_L \mathcal{G}(K, L_0) \in \mathcal{L}_K(a)$. By Lemma 3.4, we can apply the descent lemma (for minimization problems) [76] to derive:

$$\mathcal{G}(K, L_0) - \mathcal{G}(K, L_0 + \eta_0 \nabla_L \mathcal{G}(K, L_0)) \leq -\eta_0 \left\langle \nabla_L \mathcal{G}(K, L_0), \nabla_L \mathcal{G}(K, L_0) \right\rangle + \frac{\eta_0^2 \psi_{K,a}}{2} \|\nabla_L \mathcal{G}(K, L_0)\|_F^2.$$

Thus, we can additionally require $\eta_0 \leq 1/\psi_{K,a}$ to guarantee that the objective is non-decreasing (i.e., $L_0 + \eta_0 \nabla_L \mathcal{G}(K, L_0) \in \mathcal{L}_K(\mathcal{G}(K, L_0))$). This also implies that starting from $L_0 + \eta_0 \nabla_L \mathcal{G}(K, L_0)$,

taking another "fictitious" PG update step of $\eta_0 \nabla_L \mathcal{G}(K, L_0)$ with $\eta_0 \leq \min\{\delta_a/l_{K,\mathcal{G}(K,L_0)}, 1/\psi_{K,a}\}$ ensures that $L_0 + 2\eta_0 \nabla_L \mathcal{G}(K, L_0) \in \mathcal{L}_K(\mathcal{G}(K, L_0))$. Applying the same argument iteratively certifies that we can actually take a much larger stepsize $\widetilde{\eta}_0$, with the only requirement being $\widetilde{\eta}_0 \leq 1/\psi_{K,a}$, which guarantees that after one-step of the "actual" PG update, $L_1 = L_0 + \widetilde{\eta}_0 \nabla_L \mathcal{G}(K, L_0)$, it holds that $L_1 \in \mathcal{L}_K(\mathcal{G}(K, L_0))$. This is because $1/\psi_{K,a}$ can be covered by a finite times of $\delta_a/l_{K,\mathcal{G}(K,L_0)} > 0$. Now, since $L_1 \in \mathcal{L}_K(\mathcal{G}(K, L_0))$, we can repeat the above arguments for all future iterations and show that with a fixed stepsize $\eta$ satisfying $\eta \leq 1/\psi_{K,a}$, the iterates of the PG update (3.13) satisfies for all $l \geq 0$ that

$$\mathcal{G}(K, L_l) - \mathcal{G}(K, L_{l+1}) \leq -\eta \langle \nabla_L \mathcal{G}(K, L_l), \nabla_L \mathcal{G}(K, L_l) \rangle + \frac{\eta^2 \psi_{K,a}}{2} \|\nabla_L \mathcal{G}(K, L_l)\|_F^2 \leq 0.$$

By the choice of $\eta$, we characterize the convergence rate of the PG update (3.13) by exploiting the PL condition from Lemma 3.4, such that

$$\mathcal{G}(K, L_l) - \mathcal{G}(K, L_{l+1}) \leq -\frac{1}{2\psi_{K,a}} \|\nabla_L \mathcal{G}(K, L_l)\|_F^2 \leq -\frac{\mu_K}{2\psi_{K,a}} \Big( \mathcal{G}(K, L(K)) - \mathcal{G}(K, L_l) \Big),$$

where $\mu_K$ is the global PL constant for a given $K \in \mathcal{K}$. Thus, $\mathcal{G}(K, L(K)) - \mathcal{G}(K, L_{l+1}) \leq \Big( 1 - \frac{\mu_K}{2\psi_{K,a}} \Big) \Big( \mathcal{G}(K, L(K)) - \mathcal{G}(K, L_l) \Big)$. This completes the proof for the linear convergence of the objective. Next, we show the convergence of the control gain matrix to $L(K)$. We let $q := 1 - \frac{\mu_K}{2\psi_{K,a}}$ and present the following comparison

$$\mathcal{G}(K, L(K)) - \mathcal{G}(K, L_l) = \text{Tr}\Big[ (P_{K,L(K)} - P_{K,L_l}) \Sigma_0 \Big]$$

$$= \text{Tr}\Big[ \sum_{t=0}^{N} (A_{K,L_l}^\top)^t \Big( (L(K) - L_l)^\top H_{K,L(K)} (L(K) - L_l) \Big) (A_{K,L_l})^t \Sigma_0 \Big]$$

$$= \text{Tr}\Big[ (L(K) - L_l)^\top H_{K,L(K)} (L(K) - L_l) \Sigma_{K,L_l} \Big]$$

$$\geq \phi \cdot \lambda_{\min}(H_{K,L(K)}) \cdot \|L(K) - L_l\|_F^2,$$

where the second equality is due to that $E_{K,L(K)} = 0$ from the property of stationary points, and the last inequality is due to $\Sigma_{K,L_l} \geq \Sigma_0, \forall l \geq 0$. Then, we can conclude that

$$\|L(K) - L_l\|_F \leq \sqrt{\lambda_{\min}^{-1}(H_{K,L(K)}) \cdot \Big( \mathcal{G}(K, L(K)) - \mathcal{G}(K, L_l) \Big)}$$

$$\leq q^{\frac{l}{2}} \cdot \sqrt{\lambda_{\min}^{-1}(H_{K,L(K)}) \cdot \Big( \mathcal{G}(K, L(K)) - \mathcal{G}(K, L_0) \Big)}. \tag{B.10}$$

This completes the convergence proof of the gain matrix.

**NPG:** We argue that for a fixed $K \in \mathcal{K}$, the inner-loop iterates following the NPG update (3.14) with a certain stepsize choice satisfy $P_{K,L_{l+1}} \geq P_{K,L_l}$ in the p.s.d. sense, for all $l \geq 0$. By the definition of $P_{K,L}$ in (3.4), we can derive the following comparison between $P_{K,L_l}$ and $P_{K,L_{l+1}}$:

$$P_{K,L_{l+1}} - P_{K,L_l} = A_{K,L_{l+1}}^\top (P_{K,L_{l+1}} - P_{K,L_l}) A_{K,L_{l+1}} + 4\eta E_{K,L_l}^\top E_{K,L_l} - 4\eta^2 E_{K,L_l}^\top H_{K,L_l} E_{K,L_l}$$

$$= A_{K,L_{l+1}}^\top (P_{K,L_{l+1}} - P_{K,L_l}) A_{K,L_{l+1}} + 4\eta E_{K,L_l}^\top (I - \eta H_{K,L_l}) E_{K,L_l},$$

where $A_{K,L} := A - BK - DL$. As a result, we can require $\eta \leq \frac{1}{\|H_{K,L_l}\|}$ to guarantee that $P_{K,L_{l+1}} \geq P_{K,L_l}$. Moreover, Lemma 3.3 suggests that for a fixed $K \in \mathcal{K}$, it holds that $P_{K,L} \leq P_{K,L(K)}$ in the p.s.d. sense for all $L$. Thus, if we require $\eta \leq \min_l \frac{1}{\|H_{K,L_l}\|}$, then $\{P_{K,L_l}\}_{l \geq 0}$ constitutes a monotonically non-decreasing sequence in the p.s.d. sense. Moreover, since $K \in \mathcal{K}$ and $\{P_{K,L_l}\}_{l \geq 0}$ monotonically non-decreasing, the sequence $\{H_{K,L_l}\}_{l \geq 0}$ is monotonically non-increasing (in the p.s.d. sense) and each $H_{K,L_l}$ is both symmetric and positive-definite. Therefore, we can choose a uniform stepsize $\eta = \frac{1}{2\|H_{K,L_0}\|} \leq \min_l \frac{1}{\|H_{K,L_l}\|}$ and show the convergence rate of the NPG update (3.14) as follows:

$$\mathcal{G}(K, L_{l+1}) - \mathcal{G}(K, L_l) = \text{Tr}\Big( (P_{K,L_{l+1}} - P_{K,L_l}) \Sigma_0 \Big) \geq \text{Tr}\Big( (4\eta E_{K,L_l}^\top E_{K,L_l} - 4\eta^2 E_{K,L_l}^\top H_{K,L_l} E_{K,L_l}) \Sigma_0 \Big)$$

$$\geq \frac{\phi}{\|H_{K,L_0}\|} \text{Tr}(E_{K,L_l}^\top E_{K,L_l}) \geq \frac{\phi \lambda_{\min}(H_{K,L(K)})}{\|H_{K,L_0}\| \|\Sigma_{K,L(K)}\|} \Big( \mathcal{G}(K, L(K)) - \mathcal{G}(K, L_l) \Big),$$

where the last inequality follows from (B.9) . Thus,

$$\mathcal{G}(K, L(K)) - \mathcal{G}(K, L_{l+1}) \le \Big(1 - \frac{\phi \lambda_{\min}(H_{K,L(K)})}{\|H_{K,L_0}\|\|\Sigma_{K,L(K)}\|}\Big)(\mathcal{G}(K, L(K)) - \mathcal{G}(K, L_l)).$$

This implies the globally linear convergence of the objective. The convergence proof of the gain matrix is similar to the one presented for the vanilla PG update. The only difference is to have $q := 1 - \frac{\phi \lambda_{\min}(H_{K,L(K)})}{\|H_{K,L_0}\|\|\Sigma_{K,L(K)}\|}$ instead.

**GN:** Similar to the proof for NPG, we first find a stepsize such that $\{P_{K,L_l}\}_{l \ge 0}$ constitutes a monotonically non-decreasing sequence bounded above by $P_{K,L(K)}$ in the p.s.d. sense (based on Lemma 3.3). Taking the difference between $P_{K,L_l}$ and $P_{K,L_{l+1}}$ and substituting in (3.15) yield

$$P_{K,L_{l+1}} - P_{K,L_l} = A_{K,L_{l+1}}^\top (P_{K,L_{l+1}} - P_{K,L_l})A_{K,L_{l+1}} + 4\eta E_{K,L_l}^\top H_{K,L_l}^{-1} E_{K,L_l} - 4\eta^2 E_{K,L_l}^\top H_{K,L_l}^{-1} E_{K,L_l}$$

$$= A_{K,L_{l+1}}^\top (P_{K,L_{l+1}} - P_{K,L_l})A_{K,L_{l+1}} + 4\eta E_{K,L_l}^\top (H_{K,L_l}^{-1} - \eta H_{K,L_l}^{-1})E_{K,L_l}.$$

Therefore, $P_{K,L_{l+1}} \ge P_{K,L_l}$ can be ensured by choosing $\eta \le 1$. Subsequently, we characterize the convergence rate of the GN update with $\eta \le 1/2$ as follows

$$\mathcal{G}(K, L_{l+1}) - \mathcal{G}(K, L_l) = \mathrm{Tr}\Big((P_{K,L_{l+1}} - P_{K,L_l})\Sigma_0\Big) \ge \mathrm{Tr}\Big((4\eta E_{K,L_l}^\top H_{K,L_l}^{-1} E_{K,L_l} - 4\eta^2 E_{K,L_l}^\top H_{K,L_l}^{-1} E_{K,L_l})\Sigma_0\Big)$$

$$\ge \frac{\phi}{\|H_{K,L_l}\|}\mathrm{Tr}(E_{K,L_l}^\top E_{K,L_l}) \ge \frac{\phi \lambda_{\min}(H_{K,L(K)})}{\|H_{K,L_0}\|\|\Sigma_{K,L(K)}\|}\Big(\mathcal{G}(K, L(K)) - \mathcal{G}(K, L_l)\Big),$$

where the last inequality follows from (B.9) and the fact that $\{H_{K,L_l}\}_{l \ge 0}$ is a monotonically non-increasing sequence lower bounded by $H_{K,L(K)} > 0$, in the p.s.d. sense. Thus,

$$\mathcal{G}(K, L(K)) - \mathcal{G}(K, L_{l+1}) \le \Big(1 - \frac{\phi \lambda_{\min}(H_{K,L(K)})}{\|H_{K,L_0}\|\|\Sigma_{K,L(K)}\|}\Big)(\mathcal{G}(K, L(K)) - \mathcal{G}(K, L_l))$$

and the globally linear convergence rate of the objective is proved. Lastly, the locally Q-quadratic convergence rate for the GN update directly follows from [14, 41] and the convergence proof of the gain matrix is the same as the one for the NPG update. This completes the proof. ∎

## B.10  Proof of Theorem 3.7

**Proof** We first introduce the following cost difference lemma for the outer-loop problem, whose proof is deferred to §C.1.

**Lemma B.1** *(Cost Difference Lemma for Outer-Loop) Suppose that for two sequences of control gain matrices $\{K_t\}$ and $\{K_t'\}$ and a given $t \in \{0, \cdots, N-1\}$, there exist p.s.d. solutions to (3.3) at time $t+1$, denoted as $P_{K_{t+1}, L(K_{t+1})}$ and $P_{K_{t+1}', L(K_{t+1}')}$, respectively. Also, suppose that the following inequalities are satisfied:*

$$R_t^w - D_t^\top P_{K_{t+1}, L(K_{t+1})} D_t > 0, \quad R_t^w - D_t^\top P_{K_{t+1}', L(K_{t+1}')} D_t > 0.$$

*Then, there exist p.s.d. solutions to (3.3) at time $t$, denoted as $P_{K_t, L(K_t)}$ and $P_{K_t', L(K_t')}$, and their difference can be quantified as*

$$P_{K_t', L(K_t')} - P_{K_t, L(K_t)} = A_{K_t', L(K_t')}^\top (P_{K_{t+1}', L(K_{t+1}')} - P_{K_{t+1}, L(K_{t+1})})A_{K_t', L(K_t')}$$

$$+ \mathcal{R}_{K_t, K_t'} - \Xi_{K_t, K_t'}^\top (R_t^w - D_t^\top P_{K_{t+1}, L(K_{t+1})} D_t)^{-1} \Xi_{K_t, K_t'}, \tag{B.11}$$

*where*

$$\widetilde{P}_{K_{t+1}, L(K_{t+1})} := P_{K_{t+1}, L(K_{t+1})} + P_{K_{t+1}, L(K_{t+1})} D_t(R_t^w - D_t^\top P_{K_{t+1}, L(K_{t+1})} D_t)^{-1} D_t^\top P_{K_{t+1}, L(K_{t+1})} \tag{B.12}$$

$$F_{K_t, L(K_t)} := (R_t^u + B_t^\top \widetilde{P}_{K_{t+1}, L(K_{t+1})} B_t)K_t - B_t^\top \widetilde{P}_{K_{t+1}, L(K_{t+1})} A_t$$

$$L(K_t) := (-R_t^w + D_t^\top P_{K_{t+1}, L(K_{t+1})} D_t)^{-1} D_t^\top P_{K_{t+1}, L(K_{t+1})}(A_t - B_t K_t)$$

$$\Xi_{K_t, K_t'} := -(R_t^w - D_t^\top P_{K_{t+1}, L(K_{t+1})} D_t)L(K_t') - D_t^\top P_{K_{t+1}, L(K_{t+1})}(A_t - B_t K_t')$$

$$\mathcal{R}_{K_t, K_t'} := (A_t - B_t K_t')^\top \widetilde{P}_{K_{t+1}, L(K_{t+1})}(A_t - B_t K_t') - P_{K_t, L(K_t)} + Q_t + (K_t')^\top R_t^u(K_t')$$

$$= (K_t' - K_t)^\top F_{K_t, L(K_t)} + F_{K_t, L(K_t)}^\top (K_t' - K_t) + (K_t' - K_t)^\top (R_t^u + B_t^\top \widetilde{P}_{K_{t+1}, L(K_{t+1})} B_t)(K_t' - K_t). \tag{B.13}$$

Subsequently, we prove that starting from any $K \in \mathcal{K}$, the next iterate $K'$ is guaranteed to satisfy $K' \in \mathcal{K}$ following the NPG update (3.17) or the GN update (3.18) with proper stepsizes. Note that (3.8) is the compact form of the following recursive Riccati equation

$$P_{K_t, L(K_t)} = Q_t + K_t^\top R_t^u K_t + (A_t - B_t K_t)^\top \widetilde{P}_{K_{t+1}, L(K_{t+1})}(A_t - B_t K_t), \quad t \in \{0, \cdots, N-1\} \quad \text{(B.14)}$$

where $\widetilde{P}_{K_{t+1}, L(K_{t+1})}$ is as defined in (B.12) and $P_{K_N, L(K_N)} = Q_N$. Thus, $K \in \mathcal{K}$ is equivalent to that (B.14) admits a solution $P_{K_{t+1}, L(K_{t+1})} \geq 0$ and $R_t^w - D_t^\top P_{K_{t+1}, L(K_{t+1})} D_t > 0$, for all $t \in \{0, \cdots, N-1\}$. Since $K \in \mathcal{K}$ and $P_{K_N, L(K_N)} = P_{K_N', L(K_N')} = Q_N$ for any $K'$, we have

$$R_{N-1}^w - D_{N-1}^\top P_{K_N', L(K_N')} D_{N-1} = R_{N-1}^w - D_{N-1}^\top P_{K_N, L(K_N)} D_{N-1} = R_{N-1}^w - D_{N-1}^\top Q_N D_{N-1} > 0.$$

That is, $R_{N-1}^w - D_{N-1}^\top P_{K_N', L(K_N')} D_{N-1}$ is invertible and a solution to (B.14) with $t = N-1$, denoted as $P_{K_{N-1}', L(K_{N-1}')}$, exists and is both p.s.d. and unique.

Subsequently, we invoke Lemma B.1 to find a stepsize such that a solution to (B.14) with $t = N-2$ also exists after one-step NPG update (3.17). By (B.14), it suffices to ensure $R_{N-2}^w - D_{N-2}^\top P_{K_{N-1}', L(K_{N-1}')} D_{N-2} > 0$ (thus invertible) in order to guarantee the existence of $P_{K_{N-2}', L'(K_{N-2})}$. By Lemma B.1 and $P_{K_N, L(K_N)} = P_{K_N', L(K_N')} = Q_N$, we have

$$P_{K_{N-1}', L(K_{N-1}')} - P_{K_{N-1}, L(K_{N-1})} = \mathcal{R}_{K_{N-1}, K_{N-1}'} - \Xi_{K_{N-1}, K_{N-1}'}^\top (R_{N-1}^w - D_{N-1}^\top P_{K_N, L(K_N)} D_{N-1})^{-1} \Xi_{K_{N-1}, K_{N-1}'}$$
$$\leq \mathcal{R}_{K_{N-1}, K_{N-1}'},$$

where the last inequality is due to the fact that $R_{N-1}^w - D_{N-1}^\top P_{K_N, L(K_N)} D_{N-1} > 0$. Then, we substitute the NPG update rule $K_{N-1}' = K_{N-1} - 2\alpha_{K_{N-1}} F_{K_{N-1}, L(K_{N-1})}$ for $t = N-1$ into (B.13) to get

$$P_{K_{N-1}', L(K_{N-1}')} - P_{K_{N-1}, L(K_{N-1})} \leq (K_{N-1}' - K_{N-1})^\top F_{K_{N-1}, L(K_{N-1})} + F_{K_{N-1}, L(K_{N-1})}^\top (K_{N-1}' - K_{N-1})$$
$$+ (K_{N-1}' - K_{N-1})^\top (R_{N-1}^u + B_{N-1}^\top \widetilde{P}_{K_N, L(K_N)} B_{N-1})(K_{N-1}' - K_{N-1})$$
$$= -4\alpha_{K_{N-1}} F_{K_{N-1}, L(K_{N-1})}^\top F_{K_{N-1}, L(K_{N-1})} + 4\alpha_{K_{N-1}}^2 F_{K_{N-1}, L(K_{N-1})}^\top (R_{N-1}^u + B_{N-1}^\top \widetilde{P}_{K_N, L(K_N)} B_{N-1}) F_{K_{N-1}, L(K_{N-1})}$$
$$= -4\alpha_{K_{N-1}} F_{K_{N-1}, L(K_{N-1})}^\top \Big( I - \alpha_{K_{N-1}} (R_{N-1}^u + B_{N-1}^\top \widetilde{P}_{K_N, L(K_N)} B_{N-1}) \Big) F_{K_{N-1}, L(K_{N-1})}.$$

Therefore, choosing $\alpha_{K_{N-1}} \in [0, 1/\|R_{N-1}^u + B_{N-1}^\top \widetilde{P}_{K_N, L(K_N)} B_{N-1}\|]$ suffices to ensure that $P_{K_{N-1}', L(K_{N-1}')} \leq P_{K_{N-1}, L(K_{N-1})}$. Hence,

$$R_{N-2}^w - D_{N-2}^\top P_{K_{N-1}', L(K_{N-1}')} D_{N-2} \geq R_{N-2}^w - D_{N-2}^\top P_{K_{N-1}, L(K_{N-1})} D_{N-2} > 0,$$

where the last inequality comes from $K \in \mathcal{K}$. Therefore, the existence of $P_{K_{N-2}', L'(K_{N-2})} \geq 0$ is proved if the above requirement on $\alpha_{K_{N-1}}$ is satisfied. We can apply this argument iteratively backward since for all $t \in \{1, \cdots, N\}$, $P_{K_t', L(K_t')} \leq P_{K_t, L(K_t)}$ implies $R_{t-1}^w - D_{t-1}^\top P_{K_t', L(K_t')} D_{t-1} \geq R_{t-1}^w - D_{t-1}^\top P_{K_t, L(K_t)} D_{t-1} > 0$ and thus Lemma B.1 can be applied for all $t$. Moreover, $P_{K_t', L(K_t')} \leq P_{K_t, L(K_t)}$ is guaranteed to hold for all $t \in \{1, \cdots, N\}$ if we require the stepsize of the NPG update to satisfy

$$\alpha_{K_t} \in [0, 1/\|R_t^u + B_t^\top \widetilde{P}_{K_{t+1}, L(K_{t+1})} B_t\|], \quad \forall t \in \{0, \cdots, N-1\}.$$

Equivalently, the above conditions can also be represented using the compact forms introduced in §3. In particular, $G_{K, L(K)}$ is a concatenated matrix with blocks of $R_t^u + B_t^\top \widetilde{P}_{K_{t+1}, L(K_{t+1})} B_t$, $t \in \{0, \cdots, N-1\}$, on the diagonal. Thus, we have $\|G_{K, L(K)}\| \geq \|R_t^u + B_t^\top \widetilde{P}_{K_{t+1}, L(K_{t+1})} B_t\|$, for all $t \in \{0, \cdots, N-1\}$. Adopting the compact notations, we have for any $K \in \mathcal{K}$, if the stepsize $\alpha_K \in [0, 1/\|G_{K, L(K)}\|]$, then

$$P_{K', L(K')} - P_{K, L(K)} \leq (K' - K)^\top F_{K, L(K)} + F_{K, L(K)}^\top (K' - K) + (K' - K)^\top G_{K, L(K)} (K' - K)$$
$$= -4\alpha_K F_{K, L(K)}^\top F_{K, L(K)} + 4\alpha_K^2 F_{K, L(K)}^\top G_{K, L(K)} F_{K, L(K)}$$
$$= -4\alpha_K F_{K, L(K)}^\top \Big( I - \alpha_K G_{K, L(K)} \Big) F_{K, L(K)} \leq 0. \quad \text{(B.15)}$$

Since we have already shown that $P_{K_t', L(K_t')} \geq 0$ exists for all $t \in \{0, \cdots, N-1\}$ and $P_{K', L(K')}$ is a concatenation of blocks of $P_{K_t', L(K_t')}$, we can conclude that $K' \in \mathcal{K}$. Applying the above analysis iteratively proves that the sequence $\{P_{K_k, L(K_k)}\}_{k \geq 0}$ following the NPG update (3.17) with a

given $K_0 \in \mathcal{K}$ and a constant stepsize $\alpha \leq \min_{k \geq 0} \frac{1}{\|G_{K_k,L(K_k)}\|}$ is monotonically non-increasing in the p.s.d. sense, and thus $K_k \in \mathcal{K}$ for all $k \geq 0$ given a $K_0 \in \mathcal{K}$. Furthermore, we can choose $\alpha \leq \frac{1}{\|G_{K_0,L(K_0)}\|} \leq \min_{k \geq 0} \frac{1}{\|G_{K_k,L(K_k)}\|}$ because $\{G_{K_k,L(K_k)}\}$ is monotonically non-increasing and each $G_{K_k,L(K_k)}$ is symmetric and positive-definite. This completes the proof for the NPG update (3.17).

For the GN update (3.18), all our arguments for the proof above still hold but instead we invoke Lemma B.1 with the recursive update rule being $K'_t = K_t - 2\alpha_{K_t}(R^u_t + B^\top_t \widetilde{P}_{K_{t+1},L(K_{t+1})}B_t)^{-1}F_{K_t,L(K_t)}$, for all $t \in \{0, \cdots, N-1\}$. In particular, we have the matrix difference at time $t = N-1$, denoted as $P_{K'_{N-1},L(K'_{N-1})} - P_{K_{N-1},L(K_{N-1})}$, being

$$P_{K'_{N-1},L(K'_{N-1})} - P_{K_{N-1},L(K_{N-1})} \leq (K'_{N-1} - K_{N-1})^\top F_{K_{N-1},L(K_{N-1})} + F^\top_{K_{N-1},L(K_{N-1})}(K'_{N-1} - K_{N-1})$$
$$+ (K'_{N-1} - K_{N-1})^\top (R^u_{N-1} + B^\top_{N-1}\widetilde{P}_{K_N,L(K_N)}B_{N-1})(K'_{N-1} - K_{N-1})$$
$$= -4\alpha_{K_{N-1}} F^\top_{K_{N-1},L(K_{N-1})}\Big((1 - \alpha_{K_{N-1}}) \cdot (R^u_{N-1} + B^\top_{N-1}\widetilde{P}_{K_N,L(K_N)}B_{N-1})^{-1}\Big)F_{K_{N-1},L(K_{N-1})}.$$

Therefore, choosing $\alpha_{K_{N-1}} \in [0,1]$ suffices to ensure that $P_{K'_{N-1},L(K'_{N-1})} \leq P_{K_{N-1},L(K_{N-1})}$. Applying the iterative arguments for the proof of the NPG update and using the compact notations yields that for any fixed $K \in \mathcal{K}$, if the stepsize satisfies $\alpha_K \in [0,1]$, then $P_{K',L(K')} \geq 0$ exists and

$$P_{K',L(K')} - P_{K,L(K)} \leq (K' - K)^\top F_{K,L(K)} + F^\top_{K,L(K)}(K' - K) + (K' - K)^\top G_{K,L(K)}(K' - K)$$
$$= -4\alpha_K F^\top_{K,L(K)} G^{-1}_{K,L(K)} F_{K,L(K)} + 4\alpha^2_K F^\top_{K,L(K)} G^{-1}_{K,L(K)} F_{K,L(K)}$$
$$= -4\alpha_K F^\top_{K,L(K)}\Big((1 - \alpha_K) \cdot G^{-1}_{K,L(K)}\Big)F_{K,L(K)} \leq 0. \tag{B.16}$$

This proves that $K' \in \mathcal{K}$. Now, we can apply the above analysis iteratively to prove that the sequence $\{P_{K_k,L(K_k)}\}_{k \geq 0}$ following the GN update (3.18) with a given $K_0 \in \mathcal{K}$ and a constant stepsize $\alpha \in [0,1]$ is monotonically non-increasing in the p.s.d. sense. Thus. we have $K_k \in \mathcal{K}$ for all $k \geq 0$ given a $K_0 \in \mathcal{K}$. This completes the proof. ∎

## B.11 Proof of Theorem 3.8

**Proof** For the NPG update (3.17), let $\alpha \leq \frac{1}{2\|G_{K_0,L(K_0)}\|}$ and suppose $K_0 \in \mathcal{K}$. Theorem 3.7 suggests that $P_{K_k,L(K_k)} \geq P_{K^*,L(K^*)} \geq 0$ exists for all $k \geq 0$ and the sequence $\{P_{K_k,L(K_k)}\}_{k \geq 0}$ is non-increasing in the p.s.d. sense. As a result, we have by (B.15) that

$$\text{Tr}\Big((P_{K_{k+1},L(K_{k+1})} - P_{K_k,L(K_k)})\Sigma_0\Big) \leq -\frac{\phi}{\|G_{K_0,L(K_0)}\|}\text{Tr}(F^\top_{K_k,L(K_k)}F_{K_k,L(K_k)}) \leq -2\alpha\phi\text{Tr}(F^\top_{K_k,L(K_k)}F_{K_k,L(K_k)}).$$
$$\tag{B.17}$$

Therefore, from iterations $k = 0$ to $K - 1$, summing over both sides of (B.17) and dividing by $K$ yield,

$$\frac{1}{K}\sum_{k=0}^{K-1}\text{Tr}(F^\top_{K_k,L(K_k)}F_{K_k,L(K_k)}) \leq \frac{\text{Tr}(P_{K_0,L(K_0)} - P_{K^*,L(K^*)})}{2\alpha\phi \cdot K}. \tag{B.18}$$

Namely, the sequence of natural gradient norm square $\{\text{Tr}(F^\top_{K_k,L(K_k)}F_{K_k,L(K_k)})\}_{k \geq 0}$ converges on average with a globally $\mathcal{O}(1/K)$ rate. The convergent stationary point is also the unique Nash equilibrium of the game, by Lemma 3.5. This completes the convergence proof of the NPG update. Similarly, for the GN update with $\alpha \leq 1/2$ and suppose $K_0 \in \mathcal{K}$, we can obtain from (B.16) that

$$\text{Tr}\Big((P_{K_{k+1},L(K_{k+1})} - P_{K_k,L(K_k)})\Sigma_0\Big) \leq -\frac{2\alpha\phi}{\|G_{K_0,L(K_0)}\|}\text{Tr}(F^\top_{K_k,L(K_k)}F_{K_k,L(K_k)}). \tag{B.19}$$

As before, we sum up (B.19) from $k = 0$ to $K - 1$ and divide both sides by $K$ to obtain

$$\frac{1}{K}\sum_{k=0}^{K-1}\text{Tr}(F^\top_{K_k,L(K_k)}F_{K_k,L(K_k)}) \leq \frac{\|G_{K_0,L(K_0)}\|\,\text{Tr}(P_{K_0,L(K_0)} - P_{K^*,L(K^*)})}{2\alpha\phi \cdot K}.$$

This proves that the sequence of natural gradient norm squares, $\{\mathrm{Tr}(F_{K_k,L(K_k)}^\top F_{K_k,L(K_k)})\}_{k\geq 0}$, converges on average with a globally $\mathcal{O}(1/K)$ rate. Also, The convergent stationary point is the unique Nash equilibrium. Finally, faster local rates can be shown by following the techniques presented in Theorem 4.6 of [16]. ∎

## B.12 Proof of Theorem 4.1 (PG)

**Proof** We first provide a complete version of Theorem 4.1 for the PG update (4.1) and introduce a helper lemma, whose proof is deferred to §C.2.

**Theorem B.2** *(Inner-Loop Sample Complexity for PG) For a fixed $K \in \mathcal{K}$ and an arbitrary $L_0$ that induces a finite $\mathcal{G}(K, L_0)$, define a superlevel set $\mathcal{L}_K(a)$ as in (3.12), where $a \leq \mathcal{G}(K, L_0)$ is an arbitrary constant. Let $\epsilon_1, \delta_1 \in (0, 1)$, and $M_1, r_1, \eta > 0$ in Algorithm 1 satisfy*

$$M_1 \geq \left( \frac{d_1}{r_1} \Big( \mathcal{G}(K, L(K)) + \frac{l_{K,a}}{\rho_{K,a}} \Big) \sqrt{\log\Big(\frac{2d_1 L}{\delta_1}\Big)} \right)^2 \frac{1024}{\mu_K \epsilon_1},$$

$$r_1 \leq \min\left\{ \frac{\theta_{K,a}\mu_K}{8\psi_{K,a}} \sqrt{\frac{\epsilon_1}{240}}, \frac{1}{8\psi_{K,a}^2} \sqrt{\frac{\epsilon_1 \mu_K}{30}}, \rho_{K,a} \right\}, \quad \eta \leq \min\left\{ 1, \frac{1}{8\psi_{K,a}}, \rho_{K,a} \cdot \Big[ \frac{\sqrt{\mu_K}}{32} + \psi_{K,a} + l_{K,a} \Big]^{-1} \right\},$$

*where $\theta_{K,a} = \min\left\{ 1/[2\psi_{K,a}], \rho_{K,a}/l_{K,a} \right\}$; $l_{K,a}, \psi_{K,a}, \mu_K$ are defined in Lemma 3.4; $\rho_{K,a} := \min_{L \in \mathcal{L}_K(a)} \rho_{K,L} > 0$; and $d_1 = nmN$. Then, with probability at least $1 - \delta_1$ and a total number of iterations $L = \frac{8}{\eta\mu_K} \log(\frac{2}{\epsilon_1})$, the inner-loop ZO-PG update (4.1) outputs some $\overline{L}(K) := L_L$ such that $\mathcal{G}(K, \overline{L}(K)) \geq \mathcal{G}(K, L(K)) - \epsilon_1$, and $\|L(K) - \overline{L}(K)\|_F \leq \sqrt{\lambda_{\min}^{-1}(H_{K,L(K)}) \cdot \epsilon_1}$.*

**Lemma B.3** *For a given $K \in \mathcal{K}$ and a given $L$, let $\epsilon_1, \delta_1 \in (0, 1)$ and let the batchsize $M_{1,L} > 0$ and the smoothing radius $r_{1,L} > 0$ of a $M_{1,L}$-sample one-point minibatch gradient estimator satisfy*

$$M_{1,L} \geq \left( \frac{d_1}{r_{1,L}} \Big( \mathcal{G}(K, L) + \frac{l_{K,L}}{\rho_{K,L}} \Big) \sqrt{\log\Big(\frac{2d_1}{\delta_1}\Big)} \right)^2 \frac{1024}{\mu_K \epsilon_1}, \quad r_{1,L} \leq \min\left\{ \frac{\theta_{K,L}\mu_K}{8\psi_{K,L}} \sqrt{\frac{\epsilon_1}{240}}, \frac{1}{8\psi_{K,L}^2} \sqrt{\frac{\epsilon_1 \mu_K}{30}}, \rho_{K,L} \right\},$$

*where $\theta_{K,L} = \min\left\{ \frac{1}{2\psi_{K,L}}, \frac{\rho_{K,L}}{l_{K,L}} \right\}$; $l_{K,L}, \psi_{K,L}, \rho_{K,L}$ are the local curvature parameters in Lemma 3.4; and $d_1 = nmN$. Also, let the stepsize $\eta_L > 0$ satisfy*

$$\eta_L \leq \min\left\{ 1, \frac{1}{8\psi_{K,L}}, \rho_{K,L} \cdot \Big[ \frac{\sqrt{\mu_K}}{32} + \psi_{K,L} + l_{K,L} \Big] \right\}.$$

*Then, the gain matrix after applying one step of (4.1) on $L$, denoted as $L'$, satisfies with probability at least $1 - \delta_1$ that $\mathcal{G}(K, L) \leq \mathcal{G}(K, L')$ and*

$$\mathcal{G}(K, L(K)) - \mathcal{G}(K, L') \leq \left( 1 - \frac{\eta_L \mu_K}{8} \right) \cdot (\mathcal{G}(K, L(K)) - \mathcal{G}(K, L)) + \frac{\eta_L \mu_K \epsilon_1}{16}.$$

We now prove the sample complexity result for a given $L_0 \in \mathcal{L}_K(a)$. First, we use $\Delta_l := \mathcal{G}(K, L(K)) - \mathcal{G}(K, L_l)$ to denote the optimality gap at iteration $l$, where $l \in \{0, \cdots, L\}$. By Lemma B.3, if we require $\epsilon_1, \delta_1 \in (0, 1)$, the parameters $M_{1,L_0}, r_{1,L_0}, \eta_{L_0} > 0$ in Algorithm 1 to satisfy

$$M_{1,L_0} \geq \left( \frac{d_1}{r_1} \Big( \mathcal{G}(K, L_0) + \frac{l_{K,L_0}}{\rho_{K,L_0}} \Big) \sqrt{\log\Big(\frac{2d_1 L}{\delta_1}\Big)} \right)^2 \frac{1024}{\mu_K \epsilon_1},$$

$$r_{1,L_0} \leq \min\left\{ \frac{\theta_{K,L_0}\mu_K}{8\psi_{K,L_0}} \sqrt{\frac{\epsilon_1}{240}}, \frac{1}{8\psi_{K,L_0}^2} \sqrt{\frac{\epsilon_1 \mu_K}{30}}, \rho_{K,L_0} \right\}, \quad \eta_{L_0} \leq \min\left\{ 1, \frac{1}{8\psi_{K,L_0}}, \frac{\rho_{K,L_0}}{\frac{\sqrt{\mu_K}}{32} + \psi_{K,L_0} + l_{K,L_0}} \right\},$$

then we ensure with probability at least $1 - \delta_1/L$ that $\mathcal{G}(K, L_0) \leq \mathcal{G}(K, L_1)$, i.e., $L_1 \in \mathcal{L}_K(a)$. Moreover, for any $L_l$, where $l \in \{0, \cdots, L-1\}$, there exist $M_{1,L_l}, r_{1,L_l}, \eta_{L_l} > 0$ as defined in Lemma B.3 that guarantee $\mathcal{G}(K, L_l) \leq \mathcal{G}(K, L_{l+1})$ with probability at least $1 - \delta_1/L$. Now, we choose uniform constants $M_1, r_1, \eta > 0$ such that

$$M_1 \geq \left( \frac{d_1}{r_1} \left( \mathcal{G}(\boldsymbol{K}, \boldsymbol{L}(\boldsymbol{K})) + \frac{l_{K,a}}{\rho_{K,a}} \right) \sqrt{\log\left( \frac{2d_1 L}{\delta_1} \right)} \right)^2 \frac{1024}{\mu_K \epsilon_1} \geq \max_{l \in \{0, \cdots, L-1\}} M_{1, L_l},$$

$$r_1 \leq \min \left\{ \frac{\theta_{K,a} \mu_K}{8 \psi_{K,a}} \sqrt{\frac{\epsilon_1}{240}}, \; \frac{1}{8 \psi_{K,a}^2} \sqrt{\frac{\epsilon_1 \mu_K}{30}}, \; \rho_{K,a} \right\} \leq \min_{l \in \{0, \cdots, L-1\}} r_{1, L_l},$$

$$\eta \leq \min \left\{ 1, \; \frac{1}{8 \psi_{K,a}}, \; \frac{\rho_{K,a}}{\frac{\sqrt{\mu_K}}{32} + \psi_{K,a} + l_{K,a}} \right\} \leq \min_{l \in \{0, \cdots, L-1\}} \eta_{L_l},$$

where $\theta_{K,a} = \min\left\{ 1/[2\psi_{K,a}], \rho_{K,a}/l_{K,a} \right\}$, $\rho_{K,a} = \min_{L \in \mathcal{L}_K(a)} \rho_{K,L} > 0$ is due to the compactness of $\mathcal{L}_K(a)$, and $l_{K,a}, \psi_{K,a} < \infty$ are defined in Lemma 3.4 that satisfy

$$l_{K,a} \geq \max_{L \in \mathcal{L}_K(a)} l_{K,L}, \quad \psi_{K,a} \geq \max_{L \in \mathcal{L}_K(a)} \psi_{K,L}.$$

Then, we can guarantee with probability at least $1 - \delta_1$ that the value of the objective function, following the ZO-PG update (4.1), is monotonically non-decreasing. That is, we have with probability at least $1 - \delta_1$ that $\boldsymbol{L}_l \in \mathcal{L}_K(a)$, for all $l \in \{0, \cdots, L\}$, when a $\boldsymbol{L}_0 \in \mathcal{L}_K(a)$ is given. By Lemma B.3 and the above choices of $M_1, r_1, \eta$, we also ensure with probability at least $1 - \delta_1$ that $\Delta_l \leq \left( 1 - \frac{\eta \mu_K}{8} \right) \cdot \Delta_{l-1} + \frac{\eta \mu_K \epsilon_1}{16}$, for all $l \in \{1, \cdots, L-1\}$. Thus, we can show with probability at least $1 - \delta_1$ that

$$\Delta_L \leq \left( 1 - \frac{\eta \mu_K}{8} \right) \cdot \Delta_{L-1} + \eta \frac{\mu_K \epsilon_1}{16} \leq \left( 1 - \frac{\eta \mu_K}{8} \right)^L \cdot \Delta_0 + \sum_{i=1}^{L-1} \left( 1 - \frac{\eta \mu_K}{8} \right)^i \eta \frac{\mu_K \epsilon_1}{16} \leq \left( 1 - \frac{\eta \mu_K}{8} \right)^L \Delta_0 + \frac{\epsilon_1}{2}.$$

As a result, when $L = \frac{8}{\eta \mu_K} \log(\frac{2}{\epsilon_1})$, the inequality $\mathcal{G}(\boldsymbol{K}, \boldsymbol{L}_L) \leq \mathcal{G}(\boldsymbol{K}, \boldsymbol{L}(\boldsymbol{K})) - \epsilon_1$ holds with probability at least $1 - \delta_1$. This proves the convergence of the generated values of the objective function for the ZO-PG update (4.1). Lastly, we demonstrate the convergence of the gain matrix to $\boldsymbol{L}(\boldsymbol{K})$ by our results in §B.9. Based on (B.10) and the convergence of the objective function values, we have with probability at least $1 - \delta_1$ that

$$\|\boldsymbol{L}(\boldsymbol{K}) - \boldsymbol{L}_L\|_F \leq \sqrt{\lambda_{\min}^{-1}(\boldsymbol{H}_{K, L(K)}) \cdot \left( \mathcal{G}(\boldsymbol{K}, \boldsymbol{L}(\boldsymbol{K})) - \mathcal{G}(\boldsymbol{K}, \boldsymbol{L}_L) \right)} \leq \sqrt{\lambda_{\min}^{-1}(\boldsymbol{H}_{K, L(K)}) \cdot \epsilon_1}. \tag{B.20}$$

This completes the proof. ∎

## B.13 Proof of Theorem 4.1 (NPG)

**Proof** We first provide a complete version of Theorem 4.1 for the NPG update (4.2) and introduce a helper lemma, whose proof is deferred to §C.3.

**Theorem B.4** (*Inner-Loop Sample Complexity for NPG*) *For a fixed $\boldsymbol{K} \in \mathcal{K}$ and an arbitrary $\boldsymbol{L}_0$ that induces a finite $\mathcal{G}(\boldsymbol{K}, \boldsymbol{L}_0)$, define a superlevel set $\mathcal{L}_K(a)$ as in (3.12), where $a \leq \mathcal{G}(\boldsymbol{K}, \boldsymbol{L}_0)$ is an arbitrary constant. Let $\epsilon_1, \delta_1 \in (0, 1)$, and $M_1, r_1, \eta > 0$ in Algorithm 1 satisfy*

$$M_1 \geq \max\left\{ \left( \mathcal{G}(\boldsymbol{K}, \boldsymbol{L}(\boldsymbol{K})) + \frac{l_{K,a}}{\rho_{K,a}} \right)^2 \cdot \frac{64 d_1^2 (\overline{\varkappa}_a + 1)^2}{\phi^2 r_1^2 \mu_K \epsilon_1}, \; \frac{2 \overline{\varkappa}_a^2}{\phi^2} \right\} \cdot \log\left( \frac{4L \max\{d_1, d_\Sigma\}}{\delta_1} \right),$$

$$r_1 \leq \min\left\{ \frac{\phi \sqrt{\mu_K \epsilon_1}}{16 \overline{\varkappa}_a \psi_{K,a}}, \; \frac{\phi \mu_K \theta_{K,a} \sqrt{\epsilon_1 / 2}}{32 \overline{\varkappa}_a \psi_{K,a}}, \; \rho_{K,a} \right\},$$

$$\eta \leq \min\left\{ \frac{\phi^2}{32 \psi_{K,a} \overline{\varkappa}_a}, \; \frac{1}{2 \psi_{K,a}}, \; \rho_{K,a} \cdot \left[ \frac{\sqrt{\mu_K}}{4(\underline{\varkappa}_a + 1)} + \frac{2 \psi_{K,a}}{\phi} + l_{K,a} + \frac{\phi l_{K,a}}{2} \right]^{-1} \right\},$$

*where $\theta_{K,a} = \min\left\{ 1/[2\psi_{K,a}], \rho_{K,a}/l_{K,a} \right\}$; $l_{K,a}, \psi_{K,a}, \mu_K$ are defined in Lemma 3.4; $\rho_{K,a}, \overline{\varkappa}_a, \underline{\varkappa}_a$ are uniform constants over $\mathcal{L}_K(a)$ defined in §B.13; $d_\Sigma = m^2(N+1)$; and $d_1 = nmN$. Then, with probability at least $1 - \delta_1$ and a total number of iterations $L = \frac{8 \overline{\varkappa}_a}{\eta \mu_K} \log(\frac{2}{\epsilon_1})$, the inner-loop ZO-NPG update (4.2) outputs some $\overline{\boldsymbol{L}}(\boldsymbol{K}) := \boldsymbol{L}_L$ such that $\mathcal{G}(\boldsymbol{K}, \overline{\boldsymbol{L}}(\boldsymbol{K})) \geq \mathcal{G}(\boldsymbol{K}, \boldsymbol{L}(\boldsymbol{K})) - \epsilon_1$, and $\|\boldsymbol{L}(\boldsymbol{K}) - \overline{\boldsymbol{L}}(\boldsymbol{K})\|_F \leq \sqrt{\lambda_{\min}^{-1}(\boldsymbol{H}_{K, L(K)}) \cdot \epsilon_1}$.*

**Lemma B.5** *For a given $K \in \mathcal{K}$ and a given $L$, let $\epsilon_1, \delta_1 \in (0,1)$ and let the batchsize $M_{1,L} > 0$ and the smoothing radius $r_{1,L} > 0$ of a $M_{1,L}$-sample one-point minibatch gradient estimator satisfy*

$$M_{1,L} \geq \max\left\{ \left( \mathcal{G}(K, L(K)) + \frac{l_{K,L}}{\rho_{K,L}} \right)^2 \cdot \frac{64 d_1^2 (\varkappa + 1)^2}{\phi^2 r_{1,L}^2 \mu_K \epsilon_1}, \frac{2\varkappa^2}{\phi^2} \right\} \cdot \log\left( \frac{4 \max\{d_1, d_\Sigma\}}{\delta_1} \right),$$

$$r_{1,L} \leq \min\left\{ \frac{\phi \sqrt{\mu_K \epsilon_1}}{16 \varkappa \psi_{K,L}}, \frac{\phi \mu_K \theta_{K,L} \sqrt{\epsilon_1 / 2}}{32 \varkappa \psi_{K,L}}, \rho_{K,L} \right\},$$

*where $\theta_{K,L} = \min\left\{ 1/[2\psi_{K,L}], \rho_{K,L}/l_{K,L} \right\}$; $l_{K,L}, \psi_{K,L}, \rho_{K,L}$ are the local curvature parameters in Lemma 3.4; $d_1 = nmN$; $d_\Sigma = m^2(N+1)$; $\varkappa := c_{\Sigma_{K,L}} + \frac{\phi}{2}$; and $c_{\Sigma_{K,L}}$ is a polynomial of $\|A\|_F$, $\|B\|_F$, $\|D\|_F$, $\|K\|_F$, $\|L\|_F$ that is linear in $c_0$, defined in Lemma C.1. Also, let the stepsize $\eta_L > 0$ satisfy*

$$\eta_L \leq \min\left\{ \frac{\phi^2}{32 \psi_{K,L} \varkappa}, \frac{1}{2\psi_{K,L}}, \rho_{K,L} \cdot \left[ \frac{\sqrt{\mu_K}}{4(\varkappa + 1)} + \frac{2\psi_{K,L}}{\phi} + l_{K,L} + \frac{\phi l_{K,L}}{2} \right]^{-1} \right\}.$$

*Then, the gain matrix after applying one step of (4.2) on $L$, denoted as $L'$, satisfies with probability at least $1 - \delta_1$ that $\mathcal{G}(K, L) \leq \mathcal{G}(K, L')$ and*

$$\mathcal{G}(K, L(K)) - \mathcal{G}(K, L') \leq \left( 1 - \frac{\eta_L \mu_K}{8\varkappa} \right) \cdot (\mathcal{G}(K, L(K)) - \mathcal{G}(K, L)) + \frac{\eta_L \mu_K \epsilon_1}{16\varkappa}.$$

Based on Lemma B.5, the rest of the proof mostly follows the proof of Theorem 4.1 in §B.12. Specifically, we can choose uniform constants $M_1, r_1, \eta > 0$ such that

$$M_1 \geq \max\left\{ \left( \mathcal{G}(K, L(K)) + \frac{l_{K,a}}{\rho_{K,a}} \right)^2 \cdot \frac{64 d_1^2 (\overline{\varkappa}_a + 1)^2}{\phi^2 r_1^2 \mu_K \epsilon_1}, \frac{2\overline{\varkappa}_a^2}{\phi^2} \right\} \cdot \log\left( \frac{4L \max\{d_1, d_\Sigma\}}{\delta_1} \right) \geq \max_{l \in \{0, \cdots, L-1\}} M_{1,L_l},$$

$$r_1 \leq \min\left\{ \frac{\phi \sqrt{\mu_K \epsilon_1}}{16 \overline{\varkappa}_a \psi_{K,a}}, \frac{\phi \mu_K \theta_{K,a} \sqrt{\epsilon_1 / 2}}{32 \overline{\varkappa}_a \psi_{K,a}}, \rho_{K,a} \right\} \leq \min_{l \in \{0, \cdots, L-1\}} r_{1,L_l},$$

$$\eta \leq \min\left\{ \frac{\phi^2}{32 \psi_{K,a} \overline{\varkappa}_a}, \frac{1}{2\psi_{K,a}}, \rho_{K,a} \cdot \left[ \frac{\sqrt{\mu_K}}{4(\underline{\varkappa}_a + 1)} + \frac{2\psi_{K,a}}{\phi} + l_{K,a} + \frac{\phi l_{K,a}}{2} \right]^{-1} \right\} \leq \min_{l \in \{0, \cdots, L-1\}} \eta_{L_l},$$

where $\theta_{K,a} = \min\left\{ 1/[2\psi_{K,a}], \rho_{K,a}/l_{K,a} \right\}$ and

$$\overline{\varkappa}_a := \max_{L \in \mathcal{L}_K(a)} \varkappa < \infty, \quad \underline{\varkappa}_a := \min_{L \in \mathcal{L}_K(a)} \varkappa > 0, \quad \rho_{K,a} = \min_{L \in \mathcal{L}_K(a)} \rho_{K,L} > 0, \quad l_{K,a} \geq \max_{L \in \mathcal{L}_K(a)} l_{K,L}, \quad \psi_{K,a} \geq \max_{L \in \mathcal{L}_K(a)} \psi_{K,L}.$$

Then, we can guarantee with probability at least $1 - \delta_1$ that the value of the objective function, following the ZO-NPG update (4.2), is monotonically non-decreasing. That is, we have with probability at least $1 - \delta_1$ that $L_l \in \mathcal{L}_K(a)$, for all $l \in \{0, \cdots, L\}$, when an $L_0 \in \mathcal{L}_K(a)$ is given. By Lemma B.3 and the above choices of $M_1, r_1, \eta$, we also ensure with probability at least $1 - \delta_1$ that $\Delta_l \leq \left( 1 - \frac{\eta \mu_K}{8 \overline{\varkappa}_a} \right) \cdot \Delta_{l-1} + \frac{\eta \mu_K \epsilon_1}{16 \overline{\varkappa}_a}$, for all $l \in \{1, \cdots, L-1\}$. Thus, we can show with probability at least $1 - \delta_1$ that

$$\Delta_L \leq \left( 1 - \frac{\eta \mu_K}{8 \overline{\varkappa}_a} \right) \cdot \Delta_{L-1} + \frac{\eta \mu_K \epsilon_1}{16 \overline{\varkappa}_a} \leq \left( 1 - \frac{\eta \mu_K}{8 \overline{\varkappa}_a} \right)^L \cdot \Delta_0 + \sum_{i=1}^{L-1} \left( 1 - \frac{\eta \mu_K}{8 \overline{\varkappa}_a} \right)^i \frac{\eta \mu_K \epsilon_1}{16 \overline{\varkappa}_a} \leq \left( 1 - \frac{\eta \mu_K}{8 \overline{\varkappa}_a} \right)^L \Delta_0 + \frac{\epsilon_1}{2}.$$

As a result, when $L = \frac{8 \overline{\varkappa}_a}{\eta \mu_K} \log(\frac{2}{\epsilon_1})$, the inequality $\mathcal{G}(K, L_L) \leq \mathcal{G}(K, L(K)) - \epsilon_1$ holds with probability at least $1 - \delta_1$. This proves the convergence of the generated values of the objective function for the ZO-NPG update (4.2). The convergence of the gain matrix to $L(K)$ follows from the proof of Theorem 4.1 in §B.12. This completes the proof. ∎

## B.14 Proof of Theorem 4.2

**Proof** We first present a complete version of Theorem 4.2 and a few useful lemmas, whose proofs are deferred to §C.4-§C.8.

**Theorem B.6** *(IR: Derivative-Free Setting) For any $\boldsymbol{K}_0 \in \mathcal{K}$ and defining $\zeta := \lambda_{\min}(\boldsymbol{H}_{\boldsymbol{K}_0,L(\boldsymbol{K}_0)}) > 0$, introduce the following set $\widehat{\mathcal{K}}$:*

$$\widehat{\mathcal{K}} := \left\{ \boldsymbol{K} \mid \text{(3.8)} \text{ admits a solution } \boldsymbol{P}_{\boldsymbol{K},L(\boldsymbol{K})} \geq 0, \text{ and } \boldsymbol{P}_{\boldsymbol{K},L(\boldsymbol{K})} \leq \boldsymbol{P}_{\boldsymbol{K}_0,L(\boldsymbol{K}_0)} + \frac{\zeta}{2\|\boldsymbol{D}\|^2} \cdot \boldsymbol{I} \right\} \subset \mathcal{K}. \quad \text{(B.21)}$$

*Let $\delta_2 \in (0,1)$ and other parameters in Algorithm 2 satisfy $\delta_1 \leq \delta_2 / [6 M_2 K]$ and*

$$M_2 \geq \max \left\{ \frac{128 d_2^2 \alpha^2 c_0^2 \widehat{\mathcal{G}}(\boldsymbol{K}, L(\boldsymbol{K}))^2}{r_2^2 \phi^4 \varpi^2}, \frac{512 d_2^2 \alpha^2 (\widehat{\mathcal{G}}(\boldsymbol{K}, L(\boldsymbol{K})) + r_2 \mathcal{B}_P c_0)^2}{r_2^2 \phi^2 \varpi^2}, \frac{32 \alpha^2 (\widehat{c}_{5,\boldsymbol{K}})^2 (\widehat{c}_{\Sigma_{\boldsymbol{K},L(\boldsymbol{K})}})^2}{\phi^2 \varpi^2}, \frac{8 (\widehat{c}_{\Sigma_{\boldsymbol{K},L(\boldsymbol{K})}})^2}{\phi^2} \right\}$$

$$\cdot \log \left( \frac{12 K \max\{d_2, d_\Sigma\}}{\delta_2} \right), \quad \epsilon_1 \leq \min \left\{ \frac{\phi \varpi r_2}{16 \alpha d_2}, \frac{\widehat{\mathcal{B}}_{1,L(\boldsymbol{K})}^2 \zeta}{2}, \frac{\phi^2 \varpi^2 \zeta}{128 \alpha^2 (\widehat{c}_{5,\boldsymbol{K}})^2 \widehat{\mathcal{B}}_{\Sigma,L(\boldsymbol{K})}^2}, \frac{\phi^2 \zeta}{32 \widehat{\mathcal{B}}_{\Sigma,L(\boldsymbol{K})}^2} \right\},$$

$$r_2 \leq \min \left\{ \varpi, \frac{\phi \varpi}{64 \alpha \widehat{c}_{5,\boldsymbol{K}}(\widehat{c}_{\Sigma_{\boldsymbol{K},L(\boldsymbol{K})}} + \widehat{\mathcal{B}}_{\Sigma,\boldsymbol{K}})}, \frac{\phi \varpi}{64 \alpha \widehat{c}_{2,\boldsymbol{K}} \widehat{\mathcal{B}}_{\Sigma,\boldsymbol{K}}} \right\},$$

$$\alpha \leq \frac{1}{2} \cdot \left\| \boldsymbol{R}^u + \boldsymbol{B}(\overline{\boldsymbol{P}} + \overline{\boldsymbol{P}} \boldsymbol{D}(\boldsymbol{R}^w - \boldsymbol{D}^\top \overline{\boldsymbol{P}} \boldsymbol{D})^{-1} \boldsymbol{D}^\top \overline{\boldsymbol{P}}) \boldsymbol{B} \right\|^{-1}, \qquad \overline{\boldsymbol{P}} := \boldsymbol{P}_{\boldsymbol{K}_0,L(\boldsymbol{K}_0)} + \frac{\zeta}{2\|\boldsymbol{D}\|^2} \cdot \boldsymbol{I}, \quad \text{(B.22)}$$

*where $\widehat{\mathcal{G}}(\boldsymbol{K}, L(\boldsymbol{K})), \widehat{c}_{2,\boldsymbol{K}}, \widehat{c}_{5,\boldsymbol{K}}, \widehat{c}_{\Sigma_{\boldsymbol{K},L(\boldsymbol{K})}}, \widehat{\mathcal{B}}_{\Sigma,\boldsymbol{K}}, \widehat{\mathcal{B}}_{1,L(\boldsymbol{K})}, \widehat{\mathcal{B}}_{\Sigma,L(\boldsymbol{K})}, \varpi > 0$ are uniform constants over $\widehat{\mathcal{K}}$ defined in §B.14. Then, it holds with probability at least $1 - \delta_2$ that $\boldsymbol{K}_k \in \widehat{\mathcal{K}} \subset \mathcal{K}$ for all $k \in \{1, \cdots, K\}$.*

**Lemma B.7** *For any $\boldsymbol{K}, \boldsymbol{K}' \in \mathcal{K}$, there exist some $\mathcal{B}_{1,\boldsymbol{K}}, \mathcal{B}_{P,\boldsymbol{K}}, \mathcal{B}_{L(\boldsymbol{K}),\boldsymbol{K}}, \mathcal{B}_{\Sigma,\boldsymbol{K}} > 0$ that are continuous functions of $\boldsymbol{K}$ such that all $\boldsymbol{K}'$ satisfying $\|\boldsymbol{K}' - \boldsymbol{K}\|_F \leq \mathcal{B}_{1,\boldsymbol{K}}$ satisfy*

$$\|\boldsymbol{P}_{\boldsymbol{K}',L(\boldsymbol{K}')} - \boldsymbol{P}_{\boldsymbol{K},L(\boldsymbol{K})}\|_F \leq \mathcal{B}_{P,\boldsymbol{K}} \cdot \|\boldsymbol{K}' - \boldsymbol{K}\|_F \quad \text{(B.23)}$$

$$\|L(\boldsymbol{K}') - L(\boldsymbol{K})\|_F \leq \mathcal{B}_{L(\boldsymbol{K}),\boldsymbol{K}} \cdot \|\boldsymbol{K}' - \boldsymbol{K}\|_F \quad \text{(B.24)}$$

$$\|\Sigma_{\boldsymbol{K}',L(\boldsymbol{K}')} - \Sigma_{\boldsymbol{K},L(\boldsymbol{K})}\|_F \leq \mathcal{B}_{\Sigma,\boldsymbol{K}} \cdot \|\boldsymbol{K}' - \boldsymbol{K}\|_F. \quad \text{(B.25)}$$

**Lemma B.8** *For any $\boldsymbol{K}, \boldsymbol{K}' \in \mathcal{K}$, there exist some $\mathcal{B}_{1,\boldsymbol{K}}, \mathcal{B}_{\Sigma,\boldsymbol{K}} > 0$ as defined in Lemma B.7 such that if $\boldsymbol{K}'$ satisfies*

$$\|\boldsymbol{K}' - \boldsymbol{K}\|_F \leq \left\{ \mathcal{B}_{1,\boldsymbol{K}}, \frac{\epsilon_2}{4 c_{5,\boldsymbol{K}}(c_{\Sigma_{\boldsymbol{K},L(\boldsymbol{K})}} + \mathcal{B}_{\Sigma,\boldsymbol{K}})}, \frac{\epsilon_2}{4 c_{2,\boldsymbol{K}} \mathcal{B}_{\Sigma,\boldsymbol{K}}} \right\},$$

*where $c_{\Sigma_{\boldsymbol{K},L(\boldsymbol{K})}}$ is a polynomial of $\|\boldsymbol{A}\|_F, \|\boldsymbol{B}\|_F, \|\boldsymbol{D}\|_F, \|\boldsymbol{K}\|_F$, and $c_0$, and $c_{2,\boldsymbol{K}}, c_{5,\boldsymbol{K}}$ are defined in §C.9, then it holds that $\|\nabla_{\boldsymbol{K}} \mathcal{G}(\boldsymbol{K}', L(\boldsymbol{K}')) - \nabla_{\boldsymbol{K}} \mathcal{G}(\boldsymbol{K}, L(\boldsymbol{K}))\|_F \leq \epsilon_2$.*

**Lemma B.9** *For any $\boldsymbol{K} \in \mathcal{K}$, there exists some $\mathcal{B}_{2,\boldsymbol{K}} > 0$ such that all $\boldsymbol{K}'$ satisfying $\|\boldsymbol{K}' - \boldsymbol{K}\|_F \leq \mathcal{B}_{2,\boldsymbol{K}}$ satisfy $\boldsymbol{K}' \in \mathcal{K}$.*

**Lemma B.10** *For any $\boldsymbol{K} \in \mathcal{K}$, let the batchsize $M_2$, smoothing radius $r_2$, inner-loop parameters $\epsilon_1$ and $\delta_1 \in (0,1)$ satisfy*

$$M_2 \geq \max \left\{ \frac{8 d_2^2 c_0^2 \mathcal{G}(\boldsymbol{K}, L(\boldsymbol{K}))^2}{r_2^2 \phi^2 \epsilon_2^2}, \frac{32 d_2^2 (\mathcal{G}(\boldsymbol{K}, L(\boldsymbol{K})) + r_2 \mathcal{B}_{P,\boldsymbol{K}} c_0)^2}{r_2^2 \epsilon_2^2} \right\} \cdot \log \left( \frac{6 d_2}{\delta_2} \right), \quad \epsilon_1 \leq \frac{\epsilon_2 r_2}{4 d_2}$$

$$r_2 \leq \min \left\{ \mathcal{B}_{1,\boldsymbol{K}}, \mathcal{B}_{2,\boldsymbol{K}}, \frac{\epsilon_2}{16 c_{5,\boldsymbol{K}}(c_{\Sigma_{\boldsymbol{K},L(\boldsymbol{K})}} + \mathcal{B}_{\Sigma,\boldsymbol{K}})}, \frac{\epsilon_2}{16 c_{2,\boldsymbol{K}} \mathcal{B}_{\Sigma,\boldsymbol{K}}} \right\}, \quad \delta_1 \leq \frac{\delta_2}{3 M_2}.$$

*where $\mathcal{B}_{1,\boldsymbol{K}}, \mathcal{B}_{2,\boldsymbol{K}}, \mathcal{B}_{\Sigma,\boldsymbol{K}}, c_{\Sigma_{\boldsymbol{K},L(\boldsymbol{K})}} > 0$ are defined in Lemmas B.7, B.8, and B.9, and $d_2 = dmN$. Then, we have with probability at least $1 - \delta_2$ that $\left\| \overline{\nabla}_{\boldsymbol{K}} \mathcal{G}(\boldsymbol{K}, \overline{L}(\boldsymbol{K})) - \nabla_{\boldsymbol{K}} \mathcal{G}(\boldsymbol{K}, L(\boldsymbol{K})) \right\|_F \leq \epsilon_2$, where $\overline{\nabla}_{\boldsymbol{K}} \mathcal{G}(\boldsymbol{K}, \overline{L}(\boldsymbol{K}))$ is the estimated PG from Algorithm 2.*

**Lemma B.11** *For any $\boldsymbol{K} \in \mathcal{K}$, let the batchsize $M_2$, inner-loop parameters $\epsilon_1$ and $\delta_1 \in (0,1)$ satisfy*

$$M_2 \geq \frac{2 c_{\Sigma_{\boldsymbol{K},L(\boldsymbol{K})}}^2}{\epsilon_2^2} \cdot \log \left( \frac{4 d_\Sigma}{\delta_2} \right), \quad \epsilon_1 \leq \min \left\{ \mathcal{B}_{1,L(\boldsymbol{K})}^2 \lambda_{\min}(\boldsymbol{H}_{\boldsymbol{K},L(\boldsymbol{K})}), \frac{\epsilon_2^2 \lambda_{\min}(\boldsymbol{H}_{\boldsymbol{K},L(\boldsymbol{K})})}{4 \mathcal{B}_{\Sigma,L(\boldsymbol{K})}^2} \right\}, \quad \delta_1 \leq \frac{\delta_2}{2},$$

where $\mathcal{B}_{1,L(K)}, \mathcal{B}_{\Sigma,L(K)}$ are defined in Lemma C.2 and $c_{\Sigma_{K,L(K)}}$ is a polynomial of $\|A\|_F, \|B\|_F, \|D\|_F,$ $\|K\|_F$, and $c_0$. Then, we have with probability at least $1 - \delta_2$ that $\|\overline{\Sigma}_{K,\overline{L}(K)} - \Sigma_{K,L(K)}\|_F \le \epsilon_2$, with $\overline{\Sigma}_{K,\overline{L}(K)}$ being the estimated correlation matrix from Algorithm 2. Moreover, it holds with probability at least $1 - \delta_2$ that $\lambda_{\min}(\overline{\Sigma}_{K,\overline{L}(K)}) \ge \phi/2$ if $\epsilon_2 \le \phi/2$.

Based on the above lemmas, we prove the implicit regularization property of the outer-loop ZO-NPG update (4.3). Define $\zeta := \lambda_{\min}(H_{K_0,L(K_0)}) > 0$ and the following set $\widehat{\mathcal{K}}$:

$$\widehat{\mathcal{K}} := \left\{ K \mid \text{(3.8) admits a solution } P_{K,L(K)} \ge 0, \text{ and } P_{K,L(K)} \le P_{K_0,L(K_0)} + \frac{\zeta}{2\|D\|^2} \cdot I \right\} \subset \mathcal{K}.$$

It is a strict subset of $\mathcal{K}$ as all $K \in \widehat{\mathcal{K}}$ satisfy $\lambda_{\min}(H_{K,L(K)}) \ge \frac{\zeta}{2} > 0$. Then, we prove the compactness of $\widehat{\mathcal{K}}$ by first proving its boundedness. Specifically, for any $K \in \widehat{\mathcal{K}} \subset \mathcal{K}$, we have by (3.8) that $P_{K,L(K)} \ge 0$ solves

$$P_{K,L(K)} = Q + K^\top R^u K + (A - BK)^\top \Big( P_{K,L(K)} + P_{K,L(K)} D (R^w - D^\top P_{K,L(K)} D)^{-1} D^\top P_{K,L(K)} \Big)(A - BK), \quad \text{(B.26)}$$

where the second term on the RHS of (B.26) is p.s.d. and thus $Q + K^\top R^u K \le P_{K,L(K)}$ with $Q \ge 0$ and $R^u > 0$. Since $P_{K,L(K)}, P_{K_0,L(K_0)}$ are symmetric and p.s.d., all $K \in \widehat{\mathcal{K}}$ satisfy $\|P_{K,L(K)}\|_F \le \left\| P_{K_0,L(K_0)} + \frac{\zeta}{2\|D\|^2} \cdot I \right\|_F$. These arguments together imply that for all $K \in \widehat{\mathcal{K}}$, $\|K\|_F \le \sqrt{\|P_{K_0,L(K_0)} + \frac{\zeta}{2\|D\|^2} \cdot I\|_F / \lambda_{\min}(R^u)}$, proving the boundedness of $\widehat{\mathcal{K}}$.

Next, we take an arbitrary sequence $\{K_n\} \in \widehat{\mathcal{K}}$ and note that $\|K_n\|_F$ is bounded for all $n$. Applying Bolzano-Weierstrass theorem implies that the set of limit points of $\{K_n\}$, denoted as $\widehat{\mathcal{K}}_{\lim}$, is nonempty. Then, for any $K_{\lim} \in \widehat{\mathcal{K}}_{\lim}$, we can find a subsequence $\{K_{\tau_n}\} \in \widehat{\mathcal{K}}$ that converges to $K_{\lim}$. We denote the corresponding sequence of solutions to (B.26) as $\{P_{K_{\tau_n},L(K_{\tau_n})}\}$, where $0 \le P_{K_{\tau_n},L(K_{\tau_n})} \le P_{K_0,L(K_0)} + \frac{\zeta}{2\|D\|^2} \cdot I$ for all $n$. By Bolzano-Weierstrass theorem, the boundedness of $\{P_{K_{\tau_n},L(K_{\tau_n})}\}$, and the continuity of (B.26) with respect to $K$, we have the set of limit points of $\{P_{K_{\tau_n},L(K_{\tau_n})}\}$, denoted as $\widehat{\mathcal{P}}_{\lim}$, is nonempty. Then, for any $P_{\lim} \in \widehat{\mathcal{P}}_{\lim}$, we can again find a subsequence $\{P_{K_{\kappa_{\tau_n}},L(K_{\kappa_{\tau_n}})}\}$ that converges to $P_{\lim}$. Since $P_{K_{\kappa_{\tau_n}},L(K_{\kappa_{\tau_n}})}$ is a p.s.d. solution to (B.26) satisfying $0 \le P_{K_{\kappa_{\tau_n}},L(K_{\kappa_{\tau_n}})} \le P_{K_0,L(K_0)} + \frac{\zeta}{2\|D\|^2} \cdot I$ for all $n$ and (B.26) is continuous in $K$, $P_{\lim}$ must solve (B.26) and satisfy $0 \le P_{\lim} \le P_{K_0,L(K_0)} + \frac{\zeta}{2\|D\|^2} \cdot I$, which implies $K_{\lim} \in \widehat{\mathcal{K}}$. Note that the above arguments work for any sequence $\{K_n\} \in \widehat{\mathcal{K}}$ and any limit points $K_{\lim}$ and $P_{\lim}$, which proves the closedness of $\widehat{\mathcal{K}}$. Together with the boundedness, $\widehat{\mathcal{K}}$ is thus compact.

Now, denote the iterates after one-step of the outer-loop ZO-NPG update (4.3) and the exact NPG update (3.17) from $K_k$ as $K_{k+1}$ and $\widetilde{K}_{k+1}$, respectively, for $k \in \{0, \cdots, K-1\}$. Clearly, it holds that $K_0 \in \widehat{\mathcal{K}}$. Additionally, we require the stepsize $\alpha > 0$ to satisfy

$$\alpha \le \frac{1}{2} \cdot \left\| R^u + B(\overline{P} + \overline{P}D(R^w - D^\top \overline{P}D)^{-1}D^\top \overline{P})B \right\|^{-1}, \quad \text{where} \quad \overline{P} := P_{K_0,L(K_0)} + \frac{\zeta}{2\|D\|^2} \cdot I, \quad \text{(B.27)}$$

which is stricter than the requirement in Theorem 3.7. Given our stepsize choice, Theorem 3.7 guarantees that $P_{\widetilde{K}_1,L(\widetilde{K}_1)} \ge 0$ exists and satisfies $P_{\widetilde{K}_1,L(\widetilde{K}_1)} \le P_{K_0,L(K_0)}$ almost surely. By requiring $\|K_1 - \widetilde{K}_1\|_F \le \min\left\{ \mathcal{B}_{1,\widetilde{K}_1}, \mathcal{B}_{2,\widetilde{K}_1}, \frac{\zeta}{2\mathcal{B}_{P,\widetilde{K}_1} K\|D\|^2} \right\}$ with probability at least $1 - \frac{\delta_2}{K}$, we have by Lemma B.9 that $K_1 \in \mathcal{K}$ with probability at least $1 - \frac{\delta_2}{K}$. Thus, we invoke the recursive arguments in the proof of Theorem 3.7 (see §B.10) to show that $P_{K_1,L(K_1)} \ge 0$ exists and invoke Lemma B.7 to obtain that

$$\|P_{K_1,L(K_1)} - P_{\widetilde{K}_1,L(\widetilde{K}_1)}\| \le \frac{\zeta}{2K\|D\|^2} \implies P_{K_1,L(K_1)} \le P_{\widetilde{K}_1,L(\widetilde{K}_1)} + \frac{\zeta}{2K\|D\|^2} \cdot I \le P_{K_0,L(K_0)} + \frac{\zeta}{2K\|D\|^2} \cdot I. \quad \text{(B.28)}$$

In other words, it holds that $K_1 \in \widehat{\mathcal{K}}$ with probability at least $1 - \frac{\delta_2}{K}$. Subsequently, we can invoke Theorem 3.7 again to get that taking another step of the exact outer-loop NPG update starting from $K_1$, the updated gain matrix, denoted as $\widetilde{K}_2$, satisfies $P_{\widetilde{K}_2, L(\widetilde{K}_2)} \preceq P_{K_1, L(K_1)}$. Similarly, by requiring $\|K_2 - \widetilde{K}_2\|_F \leq \min\left\{\mathcal{B}_{1,\widetilde{K}_2}, \mathcal{B}_{2,\widetilde{K}_2}, \frac{\zeta}{2\mathcal{B}_{P,\widetilde{K}_2}K\|D\|^2}\right\}$, we can guarantee with probability at least $1 - \frac{\delta_2}{K}$ that $K_2 \in \mathcal{K}$, conditioned on $K_1 \in \widehat{\mathcal{K}}$. Conditioning on (B.28), we apply Theorem 3.7 and Lemma B.7 again to get with probability at least $1 - \frac{\delta_2}{K}$ that $P_{K_2, L(K_2)} \geq 0$ exists and satisfies

$$P_{K_2, L(K_2)} \preceq P_{\widetilde{K}_2, L(\widetilde{K}_2)} + \frac{\zeta}{2K\|D\|^2} \cdot I \preceq P_{K_1, L(K_1)} + \frac{\zeta}{2K\|D\|^2} \cdot I \preceq P_{K_0, L(K_0)} + \frac{\zeta}{K\|D\|^2} \cdot I.$$

That is, $K_2 \in \widehat{\mathcal{K}}$ with probability at least $1 - \frac{\delta_2}{K}$. Applying above arguments iteratively for all iterations and taking a union bound yield that if the estimation accuracy, for all $k \in \{1, \cdots, K\}$, satisfies with probability at least $1 - \frac{\delta_2}{K}$ that

$$\|K_k - \widetilde{K}_k\|_F \leq \min\left\{\mathcal{B}_{1,\widetilde{K}_k}, \mathcal{B}_{2,\widetilde{K}_k}, \frac{\zeta}{2\mathcal{B}_{P,\widetilde{K}_k}K\|D\|^2}\right\}, \tag{B.29}$$

then it holds with probability at least $1 - \delta_2$ that $K_K \in \widehat{\mathcal{K}} \subset \mathcal{K}$, where $K_K$ is the gain matrix after $K$ steps of the ZO-NPG update (4.3) starting from $K_0$ and with a constant stepsize $\alpha$ satisfying (B.27).

Lastly, we provide precise choices of the parameters for Algorithm 2 such that (B.29) can be satisfied. Since $\widehat{\mathcal{K}}$ is compact, there exist some uniform constants over $\widehat{\mathcal{K}}$

$$\mathcal{B}_1 := \min_{K \in \widehat{\mathcal{K}}} \mathcal{B}_{1,K} > 0, \quad \mathcal{B}_P := \max_{K \in \widehat{\mathcal{K}}} \mathcal{B}_{P,K} < \infty.$$

Moreover, let us denote the closure of the complement of $\mathcal{K}$ as $\overline{\mathcal{K}^c}$. Since $\widehat{\mathcal{K}} \subset \mathcal{K}$ is compact and all $K \in \widehat{\mathcal{K}}$ satisfy: i) $P_{K,L(K)} \geq 0$ exists; and (ii) $\lambda_{\min}(H_{K,L(K)}) \geq \frac{\zeta}{2} > 0$, $\widehat{\mathcal{K}}$ is disjoint from $\overline{\mathcal{K}^c}$, i.e., $\widehat{\mathcal{K}} \cap \overline{\mathcal{K}^c} = \varnothing$. Hence, one can deduce (see for example Lemma A.1 of [75]) that, there exists a distance $\mathcal{B}_2 > 0$ between $\widehat{\mathcal{K}}$ and $\overline{\mathcal{K}^c}$ such that $\mathcal{B}_2 \leq \min_{K \in \widehat{\mathcal{K}}} \mathcal{B}_{2,K}$. As a result, we now aim to enforce for all $k \in \{1, \cdots, K\}$, that the following inequality holds with probability at least $1 - \frac{\delta_2}{K}$:

$$\|K_k - \widetilde{K}_k\|_F \leq \varpi := \min\left\{\mathcal{B}_1, \mathcal{B}_2, \frac{\zeta}{2\mathcal{B}_P K\|D\|^2}\right\}. \tag{B.30}$$

Note that compared to (B.29), (B.30) is independent of the iteration index $k$, and is more stringent. Thus, (B.30) also ensures $K_k \in \widehat{\mathcal{K}}$ for all $k \in \{1, \cdots, K\}$ with probability at least $1 - \delta_2$, as (B.29) does. Condition (B.30) can be achieved by Lemmas B.10 and B.11. Specifically, we start from any gain matrix $K$ from the set of $\{K_0, \cdots, K_{K-1}\}$ and use $K'$ and $\widetilde{K}'$ to denote the gain matrices after one step of the ZO-NPG (4.3) and the exact NPG (3.17) updates, respectively. Now, suppose that the parameters of Algorithm 2 satisfy $\delta_1 \leq \delta_2/[6M_2 K]$ and

$$M_2 \geq \max\left\{\frac{128 d_2^2 \alpha^2 c_0^2 \widehat{\mathcal{G}}(K, L(K))^2}{r_2^2 \phi^4 \varpi^2}, \frac{512 d_2^2 \alpha^2 (\widehat{\mathcal{G}}(K, L(K)) + r_2 \mathcal{B}_P c_0)^2}{r_2^2 \phi^2 \varpi^2}, \frac{32 \alpha^2 (\widehat{c}_{5,K})^2 (\widehat{c}_{\Sigma_{K,L(K)}})^2}{\phi^2 \varpi^2}, \frac{8 (\widehat{c}_{\Sigma_{K,L(K)}})^2}{\phi^2}\right\}$$

$$\cdot \log\left(\frac{12 K \max\{d_2, d_\Sigma\}}{\delta_2}\right), \quad \epsilon_1 \leq \min\left\{\frac{\phi \varpi r_2}{16 \alpha d_2}, \frac{\widehat{\mathcal{B}}_{1,L(K)}^2 \zeta}{2}, \frac{\phi^2 \varpi^2 \zeta}{128 \alpha^2 (\widehat{c}_{5,K})^2 \widehat{\mathcal{B}}_{\Sigma,L(K)}^2}, \frac{\phi^2 \zeta}{32 \widehat{\mathcal{B}}_{\Sigma,L(K)}^2}\right\},$$

$$r_2 \leq \min\left\{\varpi, \frac{\phi \varpi}{64 \alpha \widehat{c}_{5,K} (\widehat{c}_{\Sigma_{K,L(K)}} + \widehat{\mathcal{B}}_{\Sigma,K})}, \frac{\phi \varpi}{64 \alpha \widehat{c}_{2,K} \widehat{\mathcal{B}}_{\Sigma,K}}\right\},$$

with the requirement on $\epsilon_1$ uses the fact that all $K \in \widehat{\mathcal{K}}$ satisfy $\lambda_{\min}(H_{K,L(K)}) \geq \frac{\zeta}{2}$. Also, $\widehat{\mathcal{G}}(K, L(K))$, $\widehat{c}_{2,K}$, $\widehat{c}_{5,K}$, $\widehat{c}_{\Sigma_{K,L(K)}}$, $\widehat{\mathcal{B}}_{\Sigma,K}$, $\widehat{\mathcal{B}}_{1,L(K)}$, $\widehat{\mathcal{B}}_{\Sigma,L(K)} > 0$ are uniform constants over $\widehat{\mathcal{K}}$ such that

$$\widehat{\mathcal{G}}(K, L(K)) := \max_{K \in \widehat{\mathcal{K}}} \mathcal{G}(K, L(K)), \quad \widehat{\mathcal{B}}_{\Sigma,K} := \max_{K \in \widehat{\mathcal{K}}} \mathcal{B}_{\Sigma,K}, \quad \widehat{\mathcal{B}}_{1,L(K)} := \min_{K \in \widehat{\mathcal{K}}} \mathcal{B}_{1,L(K)},$$

$$\widehat{\mathcal{B}}_{\Sigma,L(K)} := \max_{K \in \widehat{\mathcal{K}}} \mathcal{B}_{\Sigma,L(K)}, \quad \widehat{c}_{2,K} := \max_{K \in \widehat{\mathcal{K}}} c_{2,K}, \quad \widehat{c}_{5,K} := \max_{K \in \widehat{\mathcal{K}}} c_{5,K}, \quad \widehat{c}_{\Sigma_{K,L(K)}} := \max_{K \in \widehat{\mathcal{K}}} c_{\Sigma_{K,L(K)}},$$

where $\mathcal{B}_{\Sigma,K}$ is from Lemma B.7, $\mathcal{B}_{1,L(K)}, \mathcal{B}_{\Sigma,L(K)}$ are from Lemma C.2, $c_{2,K}, c_{5,K}$ are defined in §C.9, and $c_{\Sigma_{K,L(K)}}$ is from Lemma B.8. Then, Lemma B.10 proves that the following inequality holds with probability at least $1 - \frac{\delta_2}{2K}$:

$$\left\| \overline{\nabla}_K \mathcal{G}(K, \overline{L}(K)) - \nabla_K \mathcal{G}(K, L(K)) \right\|_F \leq \frac{\phi\varpi}{4\alpha}. \tag{B.31}$$

Moreover, we invoke Lemma B.11 to get with probability at least $1 - \frac{\delta_2}{2K}$ that

$$\left\| \overline{\Sigma}_{K,\overline{L}(K)} - \Sigma_{K,L(K)} \right\|_F \leq \min\left\{ \frac{\phi\varpi}{4\alpha\widehat{c}_{5,K}}, \frac{\phi}{2} \right\} \leq \min\left\{ \frac{\phi\varpi}{4\alpha c_{5,K}}, \frac{\phi}{2} \right\}.$$

By matrix perturbation theory (see for example Theorem 35 of [11]) and noting that $\overline{\Sigma}_{K,\overline{L}(K)} = \Sigma_{K,L(K)} + (\overline{\Sigma}_{K,\overline{L}(K)} - \Sigma_{K,L(K)})$, since $\Sigma_{K,L(K)} \geq \Sigma_0$ and $\|\overline{\Sigma}_{K,\overline{L}(K)} - \Sigma_{K,L(K)}\|_F \leq \frac{\phi}{2}$, we have with probability at least $1 - \frac{\delta_2}{2K}$ that

$$\left\| \overline{\Sigma}_{K,\overline{L}(K)}^{-1} - \Sigma_{K,L(K)}^{-1} \right\|_F \leq \frac{2\|\overline{\Sigma}_{K,\overline{L}(K)} - \Sigma_{K,L(K)}\|_F}{\phi} \leq \frac{\varpi}{2\alpha c_{5,K}}. \tag{B.32}$$

Lastly, by Lemma B.11 and $\|\overline{\Sigma}_{K,\overline{L}(K)} - \Sigma_{K,L(K)}\|_F \leq \frac{\phi}{2}$, we can ensure that $\lambda_{\min}(\overline{\Sigma}_{K,\overline{L}(K)}) \geq \frac{\phi}{2}$ and thus $\left\| \overline{\Sigma}_{K,\overline{L}(K)}^{-1} \right\|_F \leq \frac{2}{\phi}$, all with probability at least $1 - \frac{\delta_2}{2K}$. Then, we combine (B.31) and (B.32) to show that with probability at least $1 - \frac{\delta_2}{K}$,

$$\begin{aligned}
\|\widetilde{K}' - K'\|_F &= \alpha \left\| \overline{\nabla}_K \mathcal{G}(K, \overline{L}(K))\overline{\Sigma}_{K,\overline{L}(K)}^{-1} - \nabla_K \mathcal{G}(K, L(K))\Sigma_{K,L(K)}^{-1} \right\|_F \\
&\leq \alpha \left\| \overline{\nabla}_K \mathcal{G}(K, \overline{L}(K)) - \nabla_K \mathcal{G}(K, L(K)) \right\|_F \left\| \overline{\Sigma}_{K,\overline{L}(K)}^{-1} \right\|_F + \alpha \left\| \nabla_K \mathcal{G}(K, L(K)) \right\|_F \left\| \overline{\Sigma}_{K,\overline{L}(K)}^{-1} - \Sigma_{K,L(K)}^{-1} \right\|_F \\
&\leq \frac{2\alpha}{\phi} \cdot \left\| \overline{\nabla}_K \mathcal{G}(K, \overline{L}(K)) - \nabla_K \mathcal{G}(K, L(K)) \right\|_F + \alpha c_{5,K} \cdot \left\| \overline{\Sigma}_{K,\overline{L}(K)}^{-1} - \Sigma_{K,L(K)}^{-1} \right\|_F \leq \varpi,
\end{aligned}$$

where the second inequality uses (C.16). Therefore, the above choices of parameters fulfill the requirements in (B.30), and thus guarantee $K_k \in \widehat{\mathcal{K}}$ for all $k \in \{1, \cdots, K\}$ with probability at least $1 - \delta_2$. This completes the proof. ∎

## B.15 Proof of Theorem 4.3

**Proof** We provide a complete version of Theorem 4.3 as follows.

**Theorem B.12** *(Outer-Loop Sample Complexity) For any $K_0 \in \mathcal{K}$, let $\epsilon_2 \leq \phi/2$, $\delta_2, \epsilon_1, \delta_1, M_2, r_2, \alpha$ in Algorithm 2 satisfy the requirements in Theorem B.6, but with $\varpi$ therein replaced by $\varphi := \min\{\varpi, \alpha\phi\epsilon_2/[\sqrt{s}c_0\mathcal{B}_P]\}$. Then, it holds with probability at least $1 - \delta_2$ that the sequence $\{K_k\}$, $k \in \{0, \cdots, K\}$, converges with $\mathcal{O}(1/K)$ rate in the sense that $K^{-1}\sum_{k=0}^{K-1} \|F_{K_k,L(K_k)}\|_F^2 \leq \epsilon_2$ with $K = \mathrm{Tr}(P_{K_0,L(K_0)} - P_{K^*,L(K^*)})/[\alpha\phi\epsilon_2]$, which is also towards the unique Nash equilibrium.*

Then, we focus on proving Theorem B.12. First, we require the parameters in Algorithm 2 to satisfy (B.22). Then, Theorem 4.2 proves with probability at least $1 - \delta_2$ that $K_k \in \widehat{\mathcal{K}}$, for all $k \in \{1, \cdots, K\}$. Subsequently, we characterize the convergence rate of the outer-loop ZO-NPG update (4.3), conditioned on $K_k \in \widehat{\mathcal{K}}$, for all $k \in \{1, \cdots, K\}$. Once again, we start from $K_0$ and characterize the one-step progress of (4.3) from $K_0$ to $K_1$. In addition to the requirement in (B.29), we now require $\|K_1 - \widetilde{K}_1\|_F \leq \frac{\alpha\phi\epsilon_2}{\sqrt{s}c_0\mathcal{B}_{P,\widetilde{K}_1}}$ with probability at least $1 - \frac{\delta_2}{K}$, where $\widetilde{K}_1$ is the gain matrix after one step of (3.17) starting from $K_0$ and with a constant stepsize $\alpha$ satisfying (B.27). Invoking Lemma B.7 yields with probability at least $1 - \frac{\delta_2}{K}$ that

$$\mathrm{Tr}(P_{K_1,L(K_1)} - P_{\widetilde{K}_1,L(\widetilde{K}_1)}) \leq \sqrt{s}\|P_{K_1,L(K_1)} - P_{\widetilde{K}_1,L(\widetilde{K}_1)}\|_F \leq \sqrt{s}\mathcal{B}_{P,\widetilde{K}_1} \cdot \|K_1 - \widetilde{K}_1\|_F \leq \frac{\alpha\phi\epsilon_2}{c_0}, \tag{B.33}$$

where $P_{K_1,L(K_1)} - P_{\widetilde{K_1},L(\widetilde{K_1})} \in \mathbb{R}^{s \times s}$ is symmetric. By (B.17), we have with probability at least $1 - \frac{\delta_2}{K}$ that

$$
\begin{aligned}
\mathrm{Tr}\Big((P_{K_1,L(K_1)} - P_{K_0,L(K_0)})\Sigma_0\Big) &= \mathrm{Tr}\Big((P_{K_1,L(K_1)} - P_{\widetilde{K_1},L(\widetilde{K_1})})\Sigma_0\Big) + \mathrm{Tr}\Big((P_{\widetilde{K_1},L(\widetilde{K_1})} - P_{K_0,L(K_0)})\Sigma_0\Big) \\
&\leq \frac{\alpha\phi\epsilon_2}{c_0} \cdot c_0 - 2\alpha\phi\mathrm{Tr}(F_{K_0,L(K_0)}^\top F_{K_0,L(K_0)}) \leq -\alpha\phi\mathrm{Tr}(F_{K_0,L(K_0)}^\top F_{K_0,L(K_0)}),
\end{aligned}
$$

where the first inequality is due to $\|\Sigma_0\|_F \leq c_0$ almost surely and the last inequality follows from our assumption that $\mathrm{Tr}(F_{K_0,L(K_0)}^\top F_{K_0,L(K_0)}) \geq \epsilon_2$. Similarly, for all future iterations $k \in \{2,\cdots,K\}$, we can require $\|K_k - \widetilde{K}_k\|_F \leq \frac{\alpha\phi\epsilon_2}{\sqrt{s}c_0\mathcal{B}_{P,\widetilde{K}_k}}$ with probability at least $1 - \frac{\delta_2}{K}$ to obtain $\mathrm{Tr}\Big((P_{K_k,L(K_k)} - P_{K_{k-1},L(K_{k-1})})\Sigma_0\Big) \leq -\alpha\phi\mathrm{Tr}(F_{K_{k-1},L(K_{k-1})}^\top F_{K_{k-1},L(K_{k-1})})$ until the update converges to the $\epsilon_2$-stationary point of the outer-loop, which is also the unique Nash equilibrium. Summing up all iterations and taking a union bound yield with probability at least $1 - \delta_2$ the sublinear convergence rate, in that

$$
\frac{1}{K}\sum_{k=0}^{K-1}\mathrm{Tr}(F_{K_k,L(K_k)}^\top F_{K_k,L(K_k)}) \leq \frac{\mathrm{Tr}(P_{K_0,L(K_0)} - P_{K^*,L(K^*)})}{\alpha\phi \cdot K}. \tag{B.34}
$$

That is, when $K = \mathrm{Tr}(P_{K_0,L(K_0)} - P_{K^*,L(K^*)})/[\alpha\phi\epsilon_2]$, it satisfies with probability at least $1 - \delta_2$ that $K^{-1}\sum_{k=0}^{K-1}\|F_{K_k,L(K_k)}\|_F^2 \leq \epsilon_2$.

Lastly, the precise choices of the parameters for Algorithm 2 such that the above requirements on the estimation accuracy can be satisfied follow from the proof of Theorem 4.2 in §B.14, with additionally $\epsilon_2 \leq \phi/2$, and $\varpi$ in (B.30) replaced by

$$
\varphi := \min\left\{\varpi, \frac{\alpha\phi\epsilon_2}{\sqrt{s}c_0\mathcal{B}_P}\right\}, \tag{B.35}
$$

where $\mathcal{B}_P = \max_{K \in \widehat{\mathcal{K}}} \mathcal{B}_{P,K}$ is a uniform constant. Note that (B.35) is again independent of the iteration index $k$, and is more stringent than (B.30). Thus, (B.35) also ensures $K_k \in \widehat{\mathcal{K}}$ for all $k \in \{1,\cdots,K\}$ with probability at least $1 - \delta_2$, as (B.30) does. By Lemmas B.10 and B.11, requirements on the parameters in (B.22) with $\varpi$ replaced by $\varphi$ suffice to guarantee that $K_k \in \widehat{\mathcal{K}}$ for all $k \in \{1,\cdots,K\}$ with probability at least $1 - \delta_2$. Also, it ensures with probability at least $1 - \delta_2$ that the convergence in (B.34) holds. This completes the proof. ∎

# C  Supplementary Results

## C.1  Proof of Lemma B.1

**Proof** Firstly, we have $R_t^w - D_t^\top P_{K_{t+1},L(K_{t+1})}D_t$, $R_t^w - D_t^\top P_{K_{t+1}',L(K_{t+1}')}D_t$ invertible since the conditions

$$
R_t^w - D_t^\top P_{K_{t+1},L(K_{t+1})}D_t > 0, \quad R_t^w - D_t^\top P_{K_{t+1}',L(K_{t+1}')}D_t > 0
$$

are satisfied. Then, by the definition of $P_{K,L}$ in (3.3) and the definition of $L(K)$, we can derive

$$P_{K_t',L(K_t')} - P_{K_t,L(K_t)} - A_{K_t',L(K_t')}^\top (P_{K_{t+1}',L(K_{t+1}')} - P_{K_{t+1},L(K_{t+1})}) A_{K_t',L(K_t')}$$

$$= A_{K_t',L(K_t')}^\top P_{K_{t+1},L(K_{t+1})} A_{K_t',L(K_t')} - P_{K_t,L(K_t)} + P_{K_t',L(K_t')} - A_{K_t',L(K_t')}^\top P_{K_{t+1}',L(K_{t+1}')} A_{K_t',L(K_t')}$$

$$= (A_t - B_t K_t')^\top P_{K_{t+1},L(K_{t+1})}(A_t - B_t K_t') - L(K_t')^\top D_t^\top P_{K_{t+1},L(K_{t+1})}(A_t - B_t K_t') - (A_t - B_t K_t')^\top P_{K_{t+1},L(K_{t+1})} D_t L(K_t')$$

$$\quad - P_{K_t,L(K_t)} + Q_t + (K_t')^\top R_t^u (K_t') - L(K_t')^\top (R_t^w - D_t^\top P_{K_{t+1},L(K_{t+1})} D_t) L(K_t')$$

$$= (A_t - B_t K_t')^\top \widetilde{P}_{K_{t+1},L(K_{t+1})}(A_t - B_t K_t')$$

$$\quad - (A_t - B_t K_t')^\top P_{K_{t+1},L(K_{t+1})} D_t (R_t^w - D_t^\top P_{K_{t+1},L(K_{t+1})} D_t)^{-1} D_t^\top P_{K_{t+1},L(K_{t+1})}(A_t - B_t K_t') - P_{K_t,L(K_t)} + Q_t$$

$$\quad + (K_t')^\top R_t^u (K_t') - L(K_t')^\top D_t^\top P_{K_{t+1},L(K_{t+1})}(A_t - B_t K_t') - (A_t - B_t K_t')^\top P_{K_{t+1},L(K_{t+1})} D_t L(K_t')$$

$$\quad - L(K_t')^\top (R_t^w - D_t^\top P_{K_{t+1},L(K_{t+1})} D_t) L(K_t')$$

$$= \mathcal{R}_{K_t,K_t'} - (A_t - B_t K_t')^\top P_{K_{t+1},L(K_{t+1})} D_t (R_t^w - D_t^\top P_{K_{t+1},L(K_{t+1})} D_t)^{-1} D_t^\top P_{K_{t+1},L(K_{t+1})}(A_t - B_t K_t')$$

$$\quad - L(K_t')^\top D_t^\top P_{K_{t+1},L(K_{t+1})}(A_t - B_t K_t') - (A_t - B_t K_t')^\top P_{K_{t+1},L(K_{t+1})} D_t L(K_t')$$

$$\quad - L(K_t')^\top (R_t^w - D_t^\top P_{K_{t+1},L(K_{t+1})} D_t) L(K_t')$$

$$= \mathcal{R}_{K_t,K_t'} - \left[ -(R_t^w - D_t^\top P_{K_{t+1},L(K_{t+1})} D_t) L(K_t') - D_t^\top P_{K_{t+1},L(K_{t+1})}(A_t - B_t K_t') \right]^\top (R_t^w - D_t^\top P_{K_{t+1},L(K_{t+1})} D_t)^{-1}$$

$$\quad \cdot \left[ -(R_t^w - D_t^\top P_{K_{t+1},L(K_{t+1})} D_t) L(K_t') - D_t^\top P_{K_{t+1},L(K_{t+1})}(A_t - B_t K_t') \right]$$

$$= \mathcal{R}_{K_t,K_t'} - \Xi_{K_t,K_t'}^\top (R_t^w - D_t^\top P_{K_{t+1},L(K_{t+1})} D_t)^{-1} \Xi_{K_t,K_t'}.$$

This completes the proof of (B.11). Further, we invoke (3.3) to represent $\mathcal{R}_{K_t,K_t'}$ as

$$\mathcal{R}_{K_t,K_t'} = (A_t - B_t K_t')^\top \widetilde{P}_{K_{t+1},L(K_{t+1})}(A_t - B_t K_t') - P_{K_t,L(K_t)} + Q_t + (K_t')^\top R_t^u (K_t')$$

$$= (K_t' - K_t)^\top \left( (R_t^u + B_t^\top \widetilde{P}_{K_{t+1},L(K_{t+1})} B_t) K_t - B_t^\top \widetilde{P}_{K_{t+1},L(K_{t+1})} A_t \right)$$

$$\quad + (K_t' - K_t)^\top (R_t^u + B_t^\top \widetilde{P}_{K_{t+1},L(K_{t+1})} B_t)(K_t' - K_t)$$

$$\quad + \left( (R_t^u + B_t^\top \widetilde{P}_{K_{t+1},L(K_{t+1})} B_t) K_t - B_t^\top \widetilde{P}_{K_{t+1},L(K_{t+1})} A_t \right)^\top (K_t' - K_t)$$

$$= (K_t' - K_t)^\top F_{K_t,L(K_t)} + F_{K_t,L(K_t)}^\top (K_t' - K_t) + (K_t' - K_t)^\top (R_t^u + B_t^\top \widetilde{P}_{K_{t+1},L(K_{t+1})} B_t)(K_t' - K_t).$$

This completes the proof.  ∎

## C.2  Proof of Lemma B.3

**Proof** Firstly, for a fixed $K \in \mathcal{K}$ and a given $L$, we require $r_{1,L} \leq \rho_{K,L}$ to ensure that the "size" of the perturbation added to the gain matrix $L$ is smaller than the radius within which the local Lipschitz continuity and the local smoothness properties in Lemma 3.4 hold. In other words, we have $\|r_{1,L} U\|_F \leq \rho_{K,L}$, where $U$ is drawn uniformly from $\mathcal{S}(n, m, N)$ and has $\|U\|_F = 1$. Then, we invoke Lemma 23 of [19] to obtain that with probability at least $1 - \delta_1$, the concentration of gradient estimates around its mean is $\left\| \overline{\nabla}_L \mathcal{G}(K, L) - \nabla_L \mathcal{G}_{r_{1,L}}(K, L) \right\|_F \leq \frac{d_1}{r_{1,L} \sqrt{M_{1,L}}} \left( \mathcal{G}(K, L) + \frac{l_{K,L}}{\rho_{K,L}} \right) \sqrt{\log\left(\frac{2d_1}{\delta_1}\right)}$, where $\overline{\nabla}_L \mathcal{G}(K, L)$ is the gradient estimate obtained from an $M_{1,L}$-sample one-point minibatch estimator, $\nabla_L \mathcal{G}_{r_{1,L}}(K, L) := \nabla_L \mathbb{E}[\mathcal{G}(K, L + r_{1,L} U)]$ is the gradient of the smoothed version of $\mathcal{G}(K, L)$, and $d_1 = nmN$ is the degrees of freedom of $\mathcal{S}(n, m, N)$. Therefore, it suffices to require the batchsize $M_{1,L} > 0$ to satisfy $M_{1,L} \geq \left( \frac{d_1}{r_{1,L}} \left( \mathcal{G}(K, L) + \frac{l_{K,L}}{\rho_{K,L}} \right) \sqrt{\log\left(\frac{2d_1}{\delta_1}\right)} \right)^2 \frac{1024}{\mu_K \epsilon_1}$ in order to ensure that with probability at least $1 - \delta_1$ that

$$\left\| \overline{\nabla}_L \mathcal{G}(K, L) - \nabla_L \mathcal{G}_{r_{1,L}}(K, L) \right\|_F \leq \frac{\sqrt{\mu_K \epsilon_1}}{32}. \tag{C.1}$$

Then, we can bound, with probability at least $1 - \delta_1$, the "size" of the one-step ZO-PG update (4.1) by

$$
\begin{aligned}
\|\boldsymbol{L}' - \boldsymbol{L}\|_F &= \|\eta_L \overline{\nabla}_L \mathcal{G}(\boldsymbol{K}, \boldsymbol{L})\|_F \\
&\leq \eta_L \|\overline{\nabla}_L \mathcal{G}(\boldsymbol{K}, \boldsymbol{L}) - \nabla_L \mathcal{G}_{r_{1,L}}(\boldsymbol{K}, \boldsymbol{L})\|_F + \eta_L \|\nabla_L \mathcal{G}_{r_{1,L}}(\boldsymbol{K}, \boldsymbol{L}) - \nabla_L \mathcal{G}(\boldsymbol{K}, \boldsymbol{L})\|_F + \eta_L \|\nabla_L \mathcal{G}(\boldsymbol{K}, \boldsymbol{L})\|_F \\
&\leq \eta_L \left( \frac{\sqrt{\mu_K \epsilon_1}}{32} + \psi_{K,L} \sqrt{\epsilon_1} + l_{K,L} \right) \leq \eta_L \left( \frac{\sqrt{\mu_K}}{32} + \psi_{K,L} + l_{K,L} \right),
\end{aligned}
$$

where the second to last inequality follows from (C.1), Lemma 14(b) of [19], Lemma 3.4, and $r_{1,L} \sim \Theta(\sqrt{\epsilon_1})$. The last inequality follows from our requirement that $\epsilon_1 < 1$. Therefore, it suffices to require $\eta_L \leq \rho_{K,L} \cdot \left( \frac{\sqrt{\mu_K}}{32} + \psi_{K,L} + l_{K,L} \right)^{-1}$ to ensure that $\boldsymbol{L}'$ lies within the radius that the local Lipschitz continuity and the local smoothness properties hold with probability at least $1 - \delta_1$. Now, we exploit the local smoothness property to derive that

$$
\begin{aligned}
\mathcal{G}(\boldsymbol{K}, \boldsymbol{L}) - \mathcal{G}(\boldsymbol{K}, \boldsymbol{L}') &\leq -\eta_L \langle \nabla_L \mathcal{G}(\boldsymbol{K}, \boldsymbol{L}), \overline{\nabla}_L \mathcal{G}(\boldsymbol{K}, \boldsymbol{L}) \rangle + \frac{\psi_{K,L} \eta_L^2}{2} \|\overline{\nabla}_L \mathcal{G}(\boldsymbol{K}, \boldsymbol{L})\|_F^2 \\
&= -\eta_L \langle \nabla_L \mathcal{G}(\boldsymbol{K}, \boldsymbol{L}), \overline{\nabla}_L \mathcal{G}(\boldsymbol{K}, \boldsymbol{L}) - \nabla_L \mathcal{G}_{r_{1,L}}(\boldsymbol{K}, \boldsymbol{L}) \rangle - \eta_L \langle \nabla_L \mathcal{G}(\boldsymbol{K}, \boldsymbol{L}), \nabla_L \mathcal{G}_{r_{1,L}}(\boldsymbol{K}, \boldsymbol{L}) \rangle + \frac{\psi_{K,L} \eta_L^2}{2} \|\overline{\nabla}_L \mathcal{G}(\boldsymbol{K}, \boldsymbol{L})\|_F^2 \\
&\leq \eta_L \|\nabla_L \mathcal{G}(\boldsymbol{K}, \boldsymbol{L})\|_F \|\overline{\nabla}_L \mathcal{G}(\boldsymbol{K}, \boldsymbol{L}) - \nabla_L \mathcal{G}_{r_{1,L}}(\boldsymbol{K}, \boldsymbol{L})\|_F - \eta_L \|\nabla_L \mathcal{G}(\boldsymbol{K}, \boldsymbol{L})\|_F^2 \\
&\quad + \eta_L \psi_{K,L} r_{1,L} \|\nabla_L \mathcal{G}(\boldsymbol{K}, \boldsymbol{L})\|_F + \frac{\psi_{K,L} \eta_L^2}{2} \|\overline{\nabla}_L \mathcal{G}(\boldsymbol{K}, \boldsymbol{L})\|_F^2 \\
&\leq -\frac{\eta_L}{2} \|\nabla_L \mathcal{G}(\boldsymbol{K}, \boldsymbol{L})\|_F^2 + \frac{\eta_L}{2} \|\overline{\nabla}_L \mathcal{G}(\boldsymbol{K}, \boldsymbol{L}) - \nabla_L \mathcal{G}_{r_{1,L}}(\boldsymbol{K}, \boldsymbol{L})\|_F^2 \\
&\quad + \eta_L \psi_{K,L} r_{1,L} \|\nabla_L \mathcal{G}(\boldsymbol{K}, \boldsymbol{L})\|_F + \frac{\psi_{K,L} \eta_L^2}{2} \|\overline{\nabla}_L \mathcal{G}(\boldsymbol{K}, \boldsymbol{L})\|_F^2. \tag{C.2}
\end{aligned}
$$

Moreover, we can bound the last term of (C.2) as

$$
\frac{\psi_{K,L} \eta_L^2}{2} \|\overline{\nabla}_L \mathcal{G}(\boldsymbol{K}, \boldsymbol{L})\|_F^2 \leq \psi_{K,L} \eta_L^2 \|\overline{\nabla}_L \mathcal{G}(\boldsymbol{K}, \boldsymbol{L}) - \nabla_L \mathcal{G}_{r_{1,L}}(\boldsymbol{K}, \boldsymbol{L})\|_F^2 + 2\psi_{K,L} \eta_L^2 (\psi_{K,L}^2 r_1^2 + \|\nabla_L \mathcal{G}(\boldsymbol{K}, \boldsymbol{L})\|_F^2). \tag{C.3}
$$

Substituting (C.3) into (C.2) yields that the following inequality holds almost surely:

$$
\begin{aligned}
\mathcal{G}(\boldsymbol{K}, \boldsymbol{L}) - \mathcal{G}(\boldsymbol{K}, \boldsymbol{L}') &\leq \left( -\frac{\eta_L}{2} + 2\psi_{K,L} \eta_L^2 \right) \|\nabla_L \mathcal{G}(\boldsymbol{K}, \boldsymbol{L})\|_F^2 + \left( \frac{\eta_L}{2} + \psi_{K,L} \eta_L^2 \right) \|\overline{\nabla}_L \mathcal{G}(\boldsymbol{K}, \boldsymbol{L}) - \nabla_L \mathcal{G}_{r_{1,L}}(\boldsymbol{K}, \boldsymbol{L})\|_F^2 \\
&\quad + \eta_L \psi_{K,L} r_{1,L} \|\nabla_L \mathcal{G}(\boldsymbol{K}, \boldsymbol{L})\|_F + 2\psi_{K,L}^3 \eta_L^2 r_{1,L}^2.
\end{aligned}
$$

Next, recalling the definition that $\theta_{K,L} = \min\left\{ \frac{1}{2\psi_{K,L}}, \frac{\rho_{K,L}}{l_{K,L}} \right\}$, we invoke the local smoothness property to obtain

$$
\left( \theta_{K,L} - \frac{\theta_{K,L}^2 \psi_{K,L}}{2} \right) \|\nabla_L \mathcal{G}(\boldsymbol{K}, \boldsymbol{L})\|_F^2 \leq \mathcal{G}\left( \boldsymbol{K}, \boldsymbol{L} + \theta_{K,L} \nabla_L \mathcal{G}(\boldsymbol{K}, \boldsymbol{L}) \right) - \mathcal{G}(\boldsymbol{K}, \boldsymbol{L}) \leq \mathcal{G}(\boldsymbol{K}, \boldsymbol{L}(\boldsymbol{K})) - \mathcal{G}(\boldsymbol{K}, \boldsymbol{L}).
$$

Define $\Delta := \mathcal{G}(\boldsymbol{K}, \boldsymbol{L}(\boldsymbol{K})) - \mathcal{G}(\boldsymbol{K}, \boldsymbol{L})$ and $\Delta' := \mathcal{G}(\boldsymbol{K}, \boldsymbol{L}(\boldsymbol{K})) - \mathcal{G}(\boldsymbol{K}, \boldsymbol{L}')$, then it holds almost surely that

$$
\begin{aligned}
\Delta' - \Delta &\leq \left( -\frac{\eta_L}{2} + 2\psi_{K,L} \eta_L^2 \right) \|\nabla_L \mathcal{G}(\boldsymbol{K}, \boldsymbol{L})\|_F^2 + \frac{2\eta_L \psi_{K,L} r_{1,L}}{\theta_{K,L}} \Delta^{1/2} \\
&\quad + \left( \frac{\eta_L}{2} + \psi_{K,L} \eta_L^2 \right) \|\overline{\nabla}_L \mathcal{G}(\boldsymbol{K}, \boldsymbol{L}) - \nabla_L \mathcal{G}_{r_{1,L}}(\boldsymbol{K}, \boldsymbol{L})\|_F^2 + 2\psi_{K,L}^3 \eta_L^2 r_{1,L}^2 \\
&\leq -\frac{\eta_L \mu_K}{4} \Delta + \frac{2\eta_L \psi_{K,L} r_{1,L}}{\theta_{K,L}} \Delta^{1/2} + \eta_L \|\overline{\nabla}_L \mathcal{G}(\boldsymbol{K}, \boldsymbol{L}) - \nabla_L \mathcal{G}_{r_{1,L}}(\boldsymbol{K}, \boldsymbol{L})\|_F^2 + 2\psi_{K,L}^3 \eta_L^2 r_{1,L}^2 \\
&\leq -\frac{\eta_L \mu_K}{8} \Delta + \frac{8\eta_L \psi_{K,L}^2 r_{1,L}^2}{\mu_K \theta_{K,L}^2} + \eta_L \|\overline{\nabla}_L \mathcal{G}(\boldsymbol{K}, \boldsymbol{L}) - \nabla_L \mathcal{G}_{r_{1,L}}(\boldsymbol{K}, \boldsymbol{L})\|_F^2 + 2\psi_{K,L}^3 \eta_L^2 r_{1,L}^2,
\end{aligned}
$$

where the last two inequalities utilize $\eta_L \leq \frac{1}{8\psi_{K,L}}$ and the $\mu_K$-PL condition in Lemma 3.4. Thus, if the stepsize $\eta_L$ of the one-step ZO-PG update (4.1) and the smoothing radius $r_{1,L}$ for the minibatch

estimator further satisfy

$$\eta_L \le \min\left\{1, \frac{1}{8\psi_{K,L}}\right\}, \quad r_{1,L} \le \frac{1}{8\psi_{K,L}} \min\left\{\theta_{K,L}\mu_K\sqrt{\frac{\epsilon_1}{240}}, \frac{1}{\psi_{K,L}}\sqrt{\frac{\epsilon_1\mu_K}{30}}\right\},$$

then with probability at least $1 - \delta_1$, we can bound the one-step ascent as

$$\Delta' - \Delta \le -\frac{\eta_L\mu_K}{8}\Delta + \eta_L\|\overline{\nabla}_L\mathcal{G}(K,L) - \nabla_L\mathcal{G}_{r_{1,L}}(K,L)\|_F^2 + \frac{\eta_L\mu_K\epsilon_1}{60} \Rightarrow \Delta' \le \left(1 - \frac{\eta_L\mu_K}{8}\right)\Delta + \eta_L\frac{\mu_K\epsilon_1}{16}, \tag{C.4}$$

where the last inequality follows from (C.1). Lastly, by (C.4), we have $\mathcal{G}(K,L) - \mathcal{G}(K,L') = \Delta' - \Delta \le -\frac{\eta_L\mu_K}{8}\Delta + \eta_L\frac{\mu_K\epsilon_1}{16}$. Therefore, it holds that $\mathcal{G}(K,L) - \mathcal{G}(K,L') \le 0$ with probability at least $1 - \delta_1$ since by the implicit assumption that $\epsilon_1 \le \Delta$. This completes the proof. ∎

## C.3 Proof of Lemma B.5

**Proof** We first introduce the following lemma and defer its proof to the end of this subsection.

**Lemma C.1** ($\Sigma_{K,L}$ *Estimation) For any $K \in \mathcal{K}$ and any $L \in \mathcal{S}(n,m,N)$, let the batchsize $M_1$ in Algorithm 1 satisfy*

$$M_1 \ge \frac{1}{2}c_{\Sigma_{K,L}}^2\epsilon_1^{-2}\log\left(\frac{2d_\Sigma}{\delta_1}\right),$$

*where $c_{\Sigma_{K,L}}$ is a polynomial of $\|A\|_F$, $\|B\|_F$, $\|D\|_F$, $\|K\|_F$, $\|L\|_F$, and is linear in $c_0$. Then, it holds with probability at least $1 - \delta_1$ that $\|\overline{\Sigma}_{K,L} - \Sigma_{K,L}\|_F \le \epsilon_1$, where $\overline{\Sigma}_{K,L}$ is the estimated state correlation matrix obtained from Algorithm 1. Moreover, if $\epsilon_1 \le \phi/2$, then it satisfies with probability at least $1 - \delta_1$ that $\lambda_{\min}(\overline{\Sigma}_{K,L}) \ge \phi/2$.*

Similar to the proof of Lemma B.3, we first require $r_{1,L} \le \rho_{K,L}$ to ensure that perturbing $L$ preserves the local Lipschitz and smoothness properties. Then, Lemma 23 of [19] suggests that if

$$M_{1,L} \ge \left(\frac{d_1}{r_{1,L}}\left(\mathcal{G}(K,L(K)) + \frac{l_{K,L}}{\rho_{K,L}}\right)\sqrt{\log\left(\frac{4d_1}{\delta_1}\right)}\right)^2\frac{64(\varkappa+1)^2}{\phi^2\mu_K\epsilon_1}, \text{ where } \varkappa := c_{\Sigma_{K,L}} + \frac{\phi}{2} \text{ and } c_{\Sigma_{K,L}} \text{ follows the}$$

definition in Lemma C.1, then it holds with probability at least $1 - \frac{\delta_1}{2}$ that

$$\left\|\overline{\nabla}_L\mathcal{G}(K,L) - \nabla_L\mathcal{G}_{r_1}(K,L)\right\|_F \le \frac{\phi\sqrt{\mu_K\epsilon_1}}{8(\varkappa+1)}. \tag{C.5}$$

Moreover, by Lemma C.1 and if we require $M_{1,L} \ge 2\varkappa^2/\phi^2 \cdot \log(4d_\Sigma/\delta_1)$, then it holds with probability at least $1 - \frac{\delta_1}{2}$ that $\left\|\overline{\Sigma}_{K,L} - \Sigma_{K,L}\right\|_F \le \frac{\phi}{2}$. By the standard matrix perturbation theory (see for example Theorem 35 of [11]), we have

$$\left\|\overline{\Sigma}_{K,L}^{-1} - \Sigma_{K,L}^{-1}\right\|_F \le \frac{2\|\overline{\Sigma}_{K,L} - \Sigma_{K,L}\|_F}{\phi} \le 1. \tag{C.6}$$

Furthermore, by Lemma C.1 and the condition that $\|\overline{\Sigma}_{K,L} - \Sigma_{K,L}\| \le \frac{\phi}{2}$, we can ensure that $\lambda_{\min}(\overline{\Sigma}_{K,L}) \ge \frac{\phi}{2}$ and thus $\left\|\overline{\Sigma}_{K,L}^{-1}\right\| \le \frac{2}{\phi}$. Combining (C.5) and (C.6) yields that with probability at least $1 - \delta_1$, the "size" of the one-step inner-loop ZO-NPG update (4.2) can be bounded by

$$\|L' - L\|_F = \eta_L\|\overline{\nabla}_L\mathcal{G}(K,L)\overline{\Sigma}_{K,L}^{-1}\|_F \le \eta_L\left\|\overline{\nabla}_L\mathcal{G}(K,L)\overline{\Sigma}_{K,L}^{-1} - \nabla_L\mathcal{G}(K,L)\Sigma_{K,L}^{-1}\right\|_F + \eta_L\left\|\nabla_L\mathcal{G}(K,L)\Sigma_{K,L}^{-1}\right\|_F$$

$$\le \eta_L\left[\left\|\overline{\nabla}_L\mathcal{G}(K,L) - \nabla_L\mathcal{G}(K,L)\right\|_F\left\|\overline{\Sigma}_{K,L}^{-1}\right\|_F + \left\|\nabla_L\mathcal{G}(K,L)\right\|_F\left\|\overline{\Sigma}_{K,L}^{-1} - \Sigma_{K,L}^{-1}\right\|_F + \left\|\nabla_L\mathcal{G}(K,L)\right\|_F\left\|\Sigma_{K,L}^{-1}\right\|_F\right]$$

$$\le \eta_L\left[\frac{\sqrt{\mu_K}}{4(\varkappa+1)} + \frac{2\psi_{K,L}}{\phi} + l_{K,L} + \frac{\phi l_{K,L}}{2}\right],$$

where the last inequality follows from (C.5), Lemma 14 of [19], $\epsilon_1 < 1$ and $r_{1,L} \sim \Theta(\sqrt{\epsilon_1})$. Therefore, it suffices to require $\eta_L \le \rho_{K,L} \cdot \left[\frac{\sqrt{\mu_K}}{4(\varkappa+1)} + \frac{2\psi_{K,L}}{\phi} + l_{K,L} + \frac{\phi l_{K,L}}{2}\right]^{-1}$ to ensure that $L'$ lies within the

radius that the Lipschitz and smoothness properties hold. Now, we exploit the smoothness property to derive that

$$
\mathcal{G}(K,L) - \mathcal{G}(K,L') \leq -\eta_L \langle \nabla_L \mathcal{G}(K,L), \overline{\nabla}_L \mathcal{G}(K,L) \overline{\Sigma}_{K,L}^{-1} \rangle + \frac{\psi_{K,L} \eta_L^2}{2} \left\| \overline{\nabla}_L \mathcal{G}(K,L) \overline{\Sigma}_{K,L}^{-1} \right\|_F^2
$$

$$
\leq -\eta_L \langle \nabla_L \mathcal{G}(K,L), \nabla_L \mathcal{G}(K,L) \overline{\Sigma}_{K,L}^{-1} \rangle - \eta_L \langle \nabla_L \mathcal{G}(K,L), \overline{\nabla}_L \mathcal{G}(K,L) \overline{\Sigma}_{K,L}^{-1} - \nabla_L \mathcal{G}(K,L) \overline{\Sigma}_{K,L}^{-1} \rangle
$$

$$
+ \frac{\psi_{K,L} \eta_L^2}{2} \left\| \overline{\nabla}_L \mathcal{G}(K,L) \overline{\Sigma}_{K,L}^{-1} \right\|_F^2
$$

$$
\leq -\eta_L \lambda_{\min}(\overline{\Sigma}_{K,L}^{-1}) \left\| \nabla_L \mathcal{G}(K,L) \right\|_F^2 + \eta_L \left\| \nabla_L \mathcal{G}(K,L) \right\|_F \left\| \overline{\nabla}_L \mathcal{G}(K,L) - \nabla_L \mathcal{G}(K,L) \right\|_F \left\| \overline{\Sigma}_{K,L}^{-1} \right\|_F
$$

$$
+ \frac{\psi_{K,L} \eta_L^2 \left\| \overline{\Sigma}_{K,L}^{-1} \right\|_F^2}{2} \left\| \overline{\nabla}_L \mathcal{G}(K,L) \right\|_F^2
$$

$$
\leq -\eta_L \lambda_{\min}(\overline{\Sigma}_{K,L}^{-1}) \left\| \nabla_L \mathcal{G}(K,L) \right\|_F^2 + \frac{\eta_L}{2} \lambda_{\min}(\overline{\Sigma}_{K,L}^{-1}) \left\| \nabla_L \mathcal{G}(K,L) \right\|_F^2 + \frac{\eta_L \left\| \overline{\Sigma}_{K,L}^{-1} \right\|_F^2}{2 \lambda_{\min}(\overline{\Sigma}_{K,L}^{-1})} \left\| \overline{\nabla}_L \mathcal{G}(K,L) - \nabla_L \mathcal{G}_{r_1}(K,L) \right\|_F^2
$$

$$
+ \psi_{K,L} r_{1,L} \left\| \nabla_L \mathcal{G}(K,L) \right\|_F \left\| \overline{\Sigma}_{K,L}^{-1} \right\|_F + \frac{\psi_{K,L} \eta_L^2 \left\| \overline{\Sigma}_{K,L}^{-1} \right\|_F^2}{2} \left\| \overline{\nabla}_L \mathcal{G}(K,L) \right\|_F^2
$$

$$
\leq -\frac{\eta_L}{2\varkappa} \left\| \nabla_L \mathcal{G}(K,L) \right\|_F^2 + \frac{2\eta_L \varkappa}{\phi^2} \left\| \overline{\nabla}_L \mathcal{G}(K,L) - \nabla_L \mathcal{G}_{r_1}(K,L) \right\|_F^2 + \frac{2\eta_L \psi_{K,L} r_{1,L}}{\phi} \left\| \nabla_L \mathcal{G}(K,L) \right\|_F
$$

$$
+ \frac{2\psi_{K,L} \eta_L^2}{\phi^2} \left\| \overline{\nabla}_L \mathcal{G}(K,L) \right\|_F^2, \tag{C.7}
$$

where the second to last inequality utilizes Lemma 14 of [19] and

$$
\eta_L \left\| \nabla_L \mathcal{G}(K,L) \right\|_F \left\| \overline{\nabla}_L \mathcal{G}(K,L) - \nabla_L \mathcal{G}(K,L) \right\|_F \left\| \overline{\Sigma}_{K,L}^{-1} \right\|_F \leq \frac{\eta_L \lambda_{\min}(\overline{\Sigma}_{K,L}^{-1})}{2} \left\| \nabla_L \mathcal{G}(K,L) \right\|_F^2
$$

$$
+ \frac{\eta_L \left\| \overline{\Sigma}_{K,L}^{-1} \right\|_F^2}{2 \lambda_{\min}(\overline{\Sigma}_{K,L}^{-1})} \left\| \overline{\nabla}_L \mathcal{G}(K,L) - \nabla_L \mathcal{G}_{r_1}(K,L) \right\|_F^2.
$$

The last inequality in (C.7) uses $\lambda_{\min}(\overline{\Sigma}_{K,L}^{-1}) = \|\overline{\Sigma}_{K,L}\|^{-1} \geq \left[ \|\Sigma_{K,L}\|_F + \|\overline{\Sigma}_{K,L} - \Sigma_{K,L}\|_F \right]^{-1}$ and $\left\| \overline{\Sigma}_{K,L}^{-1} \right\| \leq \frac{2}{\phi}$. Moreover, we can bound the last term in (C.7) as

$$
\frac{2\psi_{K,L} \eta_L^2}{\phi^2} \left\| \overline{\nabla}_L \mathcal{G}(K,L) \right\|_F^2 \leq \frac{4\psi_{K,L} \eta_L^2}{\phi^2} \left\| \overline{\nabla}_L \mathcal{G}(K,L) - \nabla_L \mathcal{G}_{r_1}(K,L) \right\|_F^2 + \frac{8\psi_{K,L} \eta_L^2}{\phi^2} \left( \psi_{K,L}^2 r_{1,L}^2 + \|\nabla_L \mathcal{G}(K,L)\|_F^2 \right). \tag{C.8}
$$

Further, recalling the definition that $\theta_{K,L} = \min\left\{ \frac{1}{2\psi_{K,L}}, \frac{\rho_{K,L}}{l_{K,L}} \right\}$, we invoke the local smoothness property to obtain

$$
\left( \theta_{K,L} - \frac{\theta_{K,L}^2 \psi_{K,L}}{2} \right) \|\nabla_L \mathcal{G}(K,L)\|_F^2 \leq \mathcal{G}\left( K, L + \theta_{K,L} \nabla_L \mathcal{G}(K,L) \right) - \mathcal{G}(K,L) \leq \mathcal{G}(K, L(K)) - \mathcal{G}(K,L).
$$

Let $\Delta := \mathcal{G}(K, L(K)) - \mathcal{G}(K,L)$ and $\Delta' := \mathcal{G}(K, L(K)) - \mathcal{G}(K,L')$. Substituting (C.8) into (C.7) implies that the following inequality holds almost surely:

$$
\Delta' - \Delta \leq \left( -\frac{\eta_L}{2\varkappa} + \frac{8\psi_{K,L} \eta_L^2}{\phi^2} \right) \left\| \nabla_L \mathcal{G}(K,L) \right\|_F^2 + \frac{2\eta_L \psi_{K,L} r_{1,L}}{\phi} \left\| \nabla_L \mathcal{G}(K,L) \right\|_F
$$

$$
+ \left( \frac{2\eta_L \varkappa}{\phi^2} + \frac{4\psi_{K,L} \eta_L^2}{\phi^2} \right) \left\| \overline{\nabla}_L \mathcal{G}(K,L) - \nabla_L \mathcal{G}_{r_{1,L}}(K,L) \right\|_F^2 + \frac{8\psi_{K,L}^3 \eta_L^2 r_{1,L}^2}{\phi^2}
$$

$$
\leq -\frac{\eta_L \mu_K}{4\varkappa} \Delta + \frac{4\eta_L \psi_{K,L} r_{1,L}}{\phi \theta_{K,L}} \Delta^{1/2} + \left( \frac{2\eta_L \varkappa}{\phi^2} + \frac{4\psi_{K,L} \eta_L^2}{\phi^2} \right) \left\| \overline{\nabla}_L \mathcal{G}(K,L) - \nabla_L \mathcal{G}_{r_{1,L}}(K,L) \right\|_F^2 + \frac{8\psi_{K,L}^3 \eta_L^2 r_{1,L}^2}{\phi^2}
$$

$$
\leq -\frac{\eta_L \mu_K}{8\varkappa} \Delta + \frac{32\eta_L \varkappa \psi_{K,L}^2 r_{1,L}^2}{\mu_K \phi^2 \theta_{K,L}^2} + \left( \frac{2\eta_L \varkappa}{\phi^2} + \frac{4\psi_{K,L} \eta_L^2}{\phi^2} \right) \left\| \overline{\nabla}_L \mathcal{G}(K,L) - \nabla_L \mathcal{G}_{r_{1,L}}(K,L) \right\|_F^2 + \frac{8\psi_{K,L}^3 \eta_L^2 r_{1,L}^2}{\phi^2}
$$

where the second to last inequality uses $\eta_L \leq \phi^2/[32\psi_{K,L}\varkappa]$ and the PL condition. The last inequality uses $2ab \leq a^2 + b^2$. Thus, if the stepsize $\eta_L$ of the one-step ZO-NPG update (4.2) and the smoothing radius $r_{1,L}$ for the minibatch estimator satisfy

$$\eta_L \leq \min\left\{\frac{\phi^2}{32\psi_{K,L}\varkappa}, \frac{1}{2\psi_{K,L}}\right\}, \quad r_{1,L} \leq \frac{\phi}{16\varkappa\psi_{K,L}}\min\left\{\sqrt{\mu_K\epsilon_1}, \mu_K\theta_{K,L}\sqrt{\frac{\epsilon_1}{8}}\right\},$$

then with probability at least $1 - \delta_1$, we can bound the one-step ascent as

$$\Delta' - \Delta \leq -\frac{\eta_L\mu_K}{8\varkappa}\Delta + \frac{2\eta_L(\varkappa+1)}{\phi^2}\left\|\overline{\nabla}_L\mathcal{G}(K,L) - \nabla_L\mathcal{G}_{r_{1,L}}(K,L)\right\|_F^2 + \frac{32\eta_L\varkappa\psi_{K,L}^2 r_{1,L}^2}{\mu_K\phi^2\theta_{K,L}^2} + \frac{4\eta_L\varkappa\psi_{K,L}^2 r_{1,L}^2}{\phi^2}$$

$$\leq -\frac{\eta_L\mu_K}{8\varkappa}\Delta + \frac{\eta_L\mu_K\epsilon_1}{16\varkappa} \implies \Delta' \leq \left(1 - \frac{\eta_L\mu_K}{8\varkappa}\right)\Delta + \frac{\eta_L\mu_K\epsilon_1}{16\varkappa}, \tag{C.9}$$

where the last inequality uses (C.5). By (C.9), we have $\mathcal{G}(K,L) - \mathcal{G}(K,L') = \Delta' - \Delta \leq -\frac{\eta_L\mu_K}{8\varkappa}\Delta + \frac{\eta_L\mu_K\epsilon_1}{16\varkappa}$. Therefore, it holds that $\mathcal{G}(K,L) - \mathcal{G}(K,L') \leq 0$ with probability at least $1 - \delta_1$ because $\epsilon_1 \leq \Delta$.

Lastly, we prove Lemma C.1. Due to the modification to sample state trajectories using unperturbed gain matrix, we avoid the bias induced by the perturbation on the gain matrix compared to the results presented in [11]. As a result, the only bias between the estimated correlation matrix and the exact one, denoted as $\|\overline{\Sigma}_{K,L} - \Sigma_{K,L}\|_F$, where $\overline{\Sigma}_{K,L(K)} = \frac{1}{M_1}\sum_{i=0}^{M_1-1} diag\left(x_0^i(x_0^i)^\top, \cdots, x_N^i(x_N^i)^\top\right)$ and $\Sigma_{K,L} = \mathbb{E}_\xi\left[\overline{\Sigma}_{K,L}\right]$, is induced by $\xi$. Since

$$\left\|diag\left(x_0^i(x_0^i)^\top, \xi_0^i(\xi_0^i)^\top, \cdots, \xi_{N-1}^i(\xi_{N-1}^i)^\top\right)\right\|_F \leq c_0$$

holds almost surely, and

$$diag\left(x_0^i(x_0^i)^\top, \cdots, x_N^i(x_N^i)^\top\right) = \sum_{t=0}^{N-1}(A_{K,L})^t \cdot diag\left(x_0^i(x_0^i)^\top, \xi_0^i(\xi_0^i)^\top, \cdots, \xi_{N-1}^i(\xi_{N-1}^i)^\top\right) \cdot (A_{K,L}^\top)^t,$$

there exists a $c_{\Sigma_{K,L}}$ that is a polynomial of $\|A\|_F, \|B\|_F, \|D\|_F, \|K\|_F, \|L\|_F$, and is linear in $c_0$ such that $\left\|diag\left(x_0^i(x_0^i)^\top, \cdots, x_N^i(x_N^i)^\top\right)\right\|_F \leq c_{\Sigma_{K,L}}$ almost surely. Then, we apply Hoeffding's inequality to get with probability at least $1 - 2\delta_1$ that $\left\|\overline{\Sigma}_{K,L} - \Sigma_{K,L}\right\|_F \leq \sqrt{c_{\Sigma_{K,L}}^2 \log(d_\Sigma/\delta_1)/[2M_1]}$. Therefore, it suffices to choose $M_1 \geq \frac{1}{2}c_{\Sigma_{K,L}}^2\epsilon_1^{-2}\log(2d_\Sigma/\delta_1)$ to ensure that $\left\|\overline{\Sigma}_{K,L} - \Sigma_{K,L}\right\|_F \leq \epsilon_1$ with probability at least $1 - \delta_1$. Lastly, by Weyl's theorem, we can bound $\lambda_{\min}(\overline{\Sigma}_{K,L}) \geq \phi/2$ by requiring $\epsilon_1 \leq \phi/2$. This completes the proof. ∎

## C.4 Proof of Lemma B.7

**Proof** We note that (B.23) follows from Proposition B.1 of [15], but with the inner- and outer-loop players being interchanged. That is, our $B, D, K, L, R^u, R^w$ correspond to $C, B, L, K, R^v, R^u$ in [15]. Following (B.31) and (B.32) of [15], we can prove (B.23). We note that the constants in Proposition B.1 of [15] are uniform over the compact set $\Omega$ therein. However, it is hard to construct such a compact set in our setting without posing additional constraints. Therefore, $\mathcal{B}_{1,K}$ and $\mathcal{B}_{P,K}$ are only continuous functions of $K$ rather than being absolute constants, due to the quotient of two continuous functions being a continuous function provided that the denominator is not zero. Equations (B.24) and (B.25) follow from Lemmas C.2 and B.8, respectively, in the supplementary material of [15], with the same notational correspondences introduced above. Similarly, we do not have uniform constants but instead $\mathcal{B}_{L(K),K}, \mathcal{B}_{\Sigma,K}$ are continuous functions of $K$. This completes the proof. ∎

## C.5 Proof of Lemma B.8

**Proof** By the definition that $\nabla_K\mathcal{G}(K,L(K)) = 2F_{K,L(K)}\Sigma_{K,L(K)}$, we have

$$\left\|\nabla_K\mathcal{G}(K',L(K')) - \nabla_K\mathcal{G}(K,L(K))\right\|_F \leq 2\left\|F_{K',L(K')} - F_{K,L(K)}\right\|_F\left\|\Sigma_{K',L(K')}\right\|_F$$
$$+ 2\left\|F_{K,L(K)}\right\|_F\left\|\Sigma_{K',L(K')} - \Sigma_{K,L(K)}\right\|_F.$$

From Lemma C.4 and let $K, K' \in \mathcal{K}$ be sufficiently close to each other such that $\|K' - K\|_F \leq \mathcal{B}_{1,K}$, we have $\|F_{K',L(K')} - F_{K,L(K)}\|_F \leq c_{5,K} \cdot \|K' - K\|_F$, where $c_{5,K}$ is defined in §C.9. Next, the term $\|\Sigma_{K',L(K')}\|_F$ can be separated into $\|\Sigma_{K',L(K')}\|_F \leq \|\Sigma_{K,L(K)}\|_F + \|\Sigma_{K',L(K')} - \Sigma_{K,L(K)}\|_F$. To bound $\|\Sigma_{K,L(K)}\|_F$, we note that because $\|x_0\|, \|\xi_t\| \leq \vartheta$ almost surely for all $t$, there exists a finite $c_{\Sigma_{K,L(K)}}$ that is a polynomial of $\|A\|_F, \|B\|_F, \|D\|_F, \|K\|_F$, and $c_0$ such that $\|\Sigma_{K,L(K)}\|_F \leq c_{\Sigma_{K,L(K)}}$ almost surely. By Lemma B.7 and the condition that $\|K' - K\|_F \leq \mathcal{B}_{1,K}$, we have $\|\Sigma_{K',L(K')}\|_F \leq c_{\Sigma_{K,L(K)}} + \mathcal{B}_{\Sigma,K}$, which leads to

$$2\|F_{K',L(K')} - F_{K,L(K)}\|_F \|\Sigma_{K',L(K')}\|_F \leq 2c_{5,K}(c_{\Sigma_{K,L(K)}} + \mathcal{B}_{\Sigma,K}) \cdot \|K' - K\|_F.$$

Therefore, if we require $\|K' - K\|_F \leq \epsilon_2 / [4c_{5,K}(c_{\Sigma_{K,L(K)}} + \mathcal{B}_{\Sigma,K})]$, then $2\|F_{K',L(K')} - F_{K,L(K)}\|_F \|\Sigma_{K',L(K')}\|_F \leq \frac{\epsilon_2}{2}$ holds almost surely. Subsequently, we combine (C.15) with (B.25) to get

$$2\|F_{K,L(K)}\|_F \|\Sigma_{K',L(K')} - \Sigma_{K,L(K)}\|_F \leq 2c_{2,K}\mathcal{B}_{\Sigma,K} \cdot \|K' - K\|_F.$$

By requiring $\|K' - K\|_F \leq \frac{\epsilon_2}{4c_{2,K}\mathcal{B}_{\Sigma,K}}$, it holds almost surely that $2\|F_{K,L(K)}\|_F \|\Sigma_{K',L(K')} - \Sigma_{K,L(K)}\|_F \leq \frac{\epsilon_2}{2}$. Therefore, we can almost surely bound $\|\nabla_K \mathcal{G}(K', L(K')) - \nabla_K \mathcal{G}(K, L(K))\|_F \leq \epsilon_2$ by enforcing the above requirements on $\|K' - K\|_F$. This completes the proof. ∎

## C.6 Proof of Lemma B.9

**Proof** For any $\underline{K} \in \mathcal{K}$, we can define the following set:

$$\underline{\mathcal{K}} := \left\{ K \mid (3.8) \text{ admits a solution } P_{K,L(K)} \geq 0, \text{ and } P_{K,L(K)} \leq P_{\underline{K},L(\underline{K})} \right\}. \tag{C.10}$$

Clearly, it holds that $\underline{K} \in \underline{\mathcal{K}}$. Then, we first prove that $\underline{\mathcal{K}}$ is bounded. Recall that $P_{K,L(K)} \geq 0$ is the solution to (3.8), such that

$$P_{K,L(K)} = Q + K^\top R^u K + (A - BK)^\top \left( P_{K,L(K)} + P_{K,L(K)} D(R^w - D^\top P_{K,L(K)} D)^{-1} D^\top P_{K,L(K)} \right)(A - BK). \tag{C.11}$$

Since $\underline{K} \in \mathcal{K}$, any $K \in \underline{\mathcal{K}}$ also satisfies $K \in \mathcal{K}$, due to that $P_{K,L(K)} \leq P_{\underline{K},L(\underline{K})}$ implies $R^w - D^\top P_{K,L(K)}D > 0$. As a result, the second term on the RHS of (C.11) is p.s.d. and we can obtain $Q + K^\top R^u K \leq P_{K,L(K)}$, where $Q \geq 0$ and $R^u > 0$. Then, from the definition of $\underline{\mathcal{K}}$ and $P_{\underline{K},L(\underline{K})}, P_{K,L(K)}$ are symmetric and p.s.d., we have $\|P_{K,L(K)}\|_F \leq \|P_{\underline{K},L(\underline{K})}\|_F$. These arguments together imply that for any $K \in \underline{\mathcal{K}}, \|K\|_F \leq \sqrt{\|P_{\underline{K},L(\underline{K})}\|_F / \lambda_{\min}(R^u)}$, proving the boundedness of $\underline{\mathcal{K}}$.

Next, we take an arbitrary sequence $\{K_n\} \in \underline{\mathcal{K}}$ and note that $\|K_n\|_F$ is bounded for all $n$. Applying Bolzano-Weierstrass theorem implies that the set of limit points of $\{K_n\}$, denoted as $\underline{\mathcal{K}}_{\lim}$, is nonempty. Then, for any $K_{\lim} \in \underline{\mathcal{K}}_{\lim}$, we can find a subsequence $\{K_{\tau_n}\} \in \underline{\mathcal{K}}$ that converges to $K_{\lim}$. We denote the corresponding sequence of solutions to (C.11) as $\{P_{K_{\tau_n},L(K_{\tau_n})}\}$, where $0 \leq P_{K_{\tau_n},L(K_{\tau_n})} \leq P_{\underline{K},L(\underline{K})}$ for all $n$. By Bolzano-Weierstrass theorem, the boundedness of $\{P_{K_{\tau_n},L(K_{\tau_n})}\}$, and the continuity of (C.11) with respect to $K$, we have the set of limit points of $\{P_{K_{\tau_n},L(K_{\tau_n})}\}$, denoted as $\underline{\mathcal{P}}_{\lim}$, is nonempty. Then, for any $P_{\lim} \in \underline{\mathcal{P}}_{\lim}$, we can again find a subsequence $\{P_{K_{\kappa_{\tau_n}},L(K_{\kappa_{\tau_n}})}\}$ that converges to $P_{\lim}$. Since $P_{K_{\kappa_{\tau_n}},L(K_{\kappa_{\tau_n}})}$ is a p.s.d. solution to (C.11) satisfying $0 \leq P_{K_{\kappa_{\tau_n}},L(K_{\kappa_{\tau_n}})} \leq P_{\underline{K},L(\underline{K})}$ for all $n$ and (C.11) is continuous in $K$, $P_{\lim}$ must solve (C.11) and satisfy $0 \leq P_{\lim} \leq P_{\underline{K},L(\underline{K})}$, which implies $K_{\lim} \in \underline{\mathcal{K}}$. Note that the above arguments work for any sequence $\{K_n\} \in \underline{\mathcal{K}}$ and any limit points $K_{\lim}$ and $P_{\lim}$, which proves the closedness of $\underline{\mathcal{K}}$. Together with the boundedness, $\underline{\mathcal{K}}$ is thus compact.

Finally, let us denote the closure of the complement of $\mathcal{K}$ as $\overline{\mathcal{K}^c}$. By $\underline{K} \in \underline{\mathcal{K}} \subset \mathcal{K}$ and (C.10), any $K \in \underline{\mathcal{K}}$ satisfies: i) $P_{K,L(K)} \geq 0$ exists; ii) $R^w - D^\top P_{K,L(K)}D \geq R^w - D^\top P_{\underline{K},L(\underline{K})}D > 0$. This implies that $\underline{\mathcal{K}}$ is disjoint with $\overline{\mathcal{K}^c}$, i.e., $\underline{\mathcal{K}} \cap \overline{\mathcal{K}^c} = \varnothing$. Then, there exists a distance $\mathcal{B}_{2,K} > 0$ between $\underline{\mathcal{K}}$ and $\overline{\mathcal{K}^c}$ such that for a given $\underline{K} \in \underline{\mathcal{K}}$, all $K'$ satisfying $\|K' - \underline{K}\| \leq \mathcal{B}_{2,K}$ also satisfy $K' \in \mathcal{K}$ (see for example Lemma A.1 of [75]). This completes the proof. ∎

## C.7 Proof of Lemma B.10

**Proof** To simplify the notations, we first define

$$\check{\nabla} := \overline{\nabla}_K \mathcal{G}(K, \overline{L}(K)) = \frac{1}{M_2} \sum_{j=0}^{M_2-1} \frac{d_2}{r_2} \Big[ \sum_{t=0}^{N} c_t^j \Big] V^j, \quad \overline{\nabla} := \overline{\nabla}_K \mathcal{G}(K, L(K)), \quad \widehat{\nabla} := \mathbb{E}_\xi[\overline{\nabla}],$$

$$\widetilde{\nabla} := \nabla_K \mathcal{G}_{r_2}(K, L(K)), \quad \nabla := \nabla_K \mathcal{G}(K, L(K)),$$

where $\overline{L}(K)$ is the approximate inner-loop solution obtained from Algorithm 1. Our goal is to quantify $\|\check{\nabla} - \nabla\|_F$, which can be separated into four terms $\|\check{\nabla} - \nabla\|_F \leq \|\check{\nabla} - \overline{\nabla}\|_F + \|\overline{\nabla} - \widehat{\nabla}\|_F + \|\widehat{\nabla} - \widetilde{\nabla}\|_F + \|\widetilde{\nabla} - \nabla\|_F$.

According to Theorem 4.1, $\mathcal{G}(K + r_2 V, \overline{L}(K + r_2 V)) \geq \mathcal{G}(K + r_2 V, L(K + r_2 V)) - \epsilon_1$ holds with probability at least $1 - \delta_1$. Applying union bound and triangle inequality yield that $\|\check{\nabla} - \overline{\nabla}\|_F \leq d_2 \epsilon_1 / r_2$ holds with probability at least $1 - M_2 \delta_1$. Therefore, we can bound $\|\check{\nabla} - \overline{\nabla}\|_F$ by $\epsilon_2/4$ with probability at least $1 - \delta_2/3$ by requiring $\epsilon_1 \leq \epsilon_2 r_2 / [4 d_2]$, and $\delta_1 \leq \delta_2 / [3 M_2]$.

Subsequently, we consider the bias induced by $\xi$. By definition in (3.2), we have $\sum_{t=0}^{N} c_t^j$ driven by $(K, L(K))$ and $x_0, \xi_t \sim \mathcal{D}$ for all $t$ can also be represented as $\text{Tr}\Big[ P_{K, L(K)} \cdot diag(x_0 x_0^\top, \xi_0 \xi_0^\top, \cdots, \xi_{N-1} \xi_{N-1}^\top) \Big]$, where $\big\| diag(x_0 x_0^\top, \xi_0 \xi_0^\top, \cdots, \xi_{N-1} \xi_{N-1}^\top) \big\|_F \leq c_0$ almost surely. Then, we apply Hoeffding's inequality and (C.13) to get with probability at least $1 - 2\delta_2$ that $\|\overline{\nabla} - \widehat{\nabla}\|_F \leq d_2 c_0 \mathcal{G}(K, L(K)) / [r_2 \phi] \cdot \sqrt{\log(d_2/\delta_2)/[2 M_2]}$. Then, it suffices to choose $M_2 \geq 8 d_2^2 c_0^2 \mathcal{G}(K, L(K))^2 / [r_2^2 \phi^2 \epsilon_2^2] \cdot \log(6 d_2/\delta_2)$ to ensure that $\|\overline{\nabla} - \widehat{\nabla}\|_F \leq \epsilon_2/4$ with probability at least $1 - \delta_2/3$.

For the third term, we can apply standard results in zeroth-order optimization [77], which leads to $\widetilde{\nabla} = d_2/r_2 \cdot \mathbb{E}_V \Big[ \mathcal{G}(K + r_2 V, L(K + r_2 V)) V \Big]$. Then, we invoke Lemma B.7 and Hoeffding's inequality to get with probability at least $1 - 2\delta_2$ that $\|\widehat{\nabla} - \widetilde{\nabla}\|_F \leq d_2/r_2 \cdot (\mathcal{G}(K, L(K)) + r_2 \mathcal{B}_{P,K} c_0) \cdot \sqrt{2 \log(d_2/\delta_2)/M_2}$, where we require $r_2 \leq \mathcal{B}_{1,K}$. Thus, it suffices to choose $M_2 \geq 32 d_2^2 (\mathcal{G}(K, L(K)) + r_2 \mathcal{B}_{P,K} c_0)^2 / [r_2^2 \epsilon_2^2] \cdot \log(6 d_2/\delta_2)$ to ensure that $\|\widehat{\nabla} - \widetilde{\nabla}\|_F \leq \epsilon_2/4$ with probability at least $1 - \delta_2/3$.

Lastly, by Lemma B.8, we can bound the last term $\|\widetilde{\nabla} - \nabla\|_F$ almost surely by $\epsilon_2/4$ if requiring

$$r_2 = \|K' - K\|_F \leq \min\Big\{ \mathcal{B}_{1,K}, \mathcal{B}_{2,K}, \frac{\epsilon_2}{16 c_{5,K}(c_{\Sigma_{K,L(K)}} + \mathcal{B}_{\Sigma,K})}, \frac{\epsilon_2}{16 c_{2,K} \mathcal{B}_{\Sigma,K}} \Big\}$$

where $\mathcal{B}_{1,K}, \mathcal{B}_{\Sigma,K}$ are defined in Lemma B.7, $c_{\Sigma_{K,L(K)}}$ is defined in §C.5, and $r_2 \leq \mathcal{B}_{2,K}$ ensures the perturbed gain matrices $K + r_2 V^j$, for all $j$, lie within $\mathcal{K}$, as proved in Lemma B.9. Thus, we have $\|\check{\nabla} - \nabla\|_F \leq \epsilon_2$ with probability at least $1 - \delta_2$, which completes the proof. ∎

## C.8 Proof of Lemma B.11

**Proof** We first introduce a lemma similar to Lemma 16 of [11] and defer its proof to the end of this subsection.

**Lemma C.2** ($\Sigma_{K,L}$ *Perturbation*) *For a fixed $K \in \mathcal{K}$, there exist some continuous functions $\mathcal{B}_{1,L}, \mathcal{B}_{\Sigma,L} > 0$ such that if $\|L' - L\|_F \leq \mathcal{B}_{1,L}$, then it holds that $\|\Sigma_{K,L'} - \Sigma_{K,L}\|_F \leq \mathcal{B}_{\Sigma,L} \cdot \|L' - L\|_F$.*

The estimation bias, $\|\overline{\Sigma}_{K, \overline{L}(K)} - \Sigma_{K, L(K)}\|_F$, can be separated into two terms, such that

$$\|\overline{\Sigma}_{K, \overline{L}(K)} - \Sigma_{K, L(K)}\|_F \leq \|\overline{\Sigma}_{K, \overline{L}(K)} - \Sigma_{K, \overline{L}(K)}\|_F + \|\Sigma_{K, \overline{L}(K)} - \Sigma_{K, L(K)}\|_F, \tag{C.12}$$

where $\overline{\Sigma}_{K, \overline{L}(K)} = \frac{1}{M_2} \sum_{j=0}^{M_2-1} diag\big( x_0^j (x_0^j)^\top, \cdots, x_N^j (x_N^j)^\top \big)$, $\Sigma_{K, \overline{L}(K)} = \mathbb{E}_\xi \Big[ \overline{\Sigma}_{K, \overline{L}(K)} \Big]$, and $\{x_t\}$ is the sequence of noisy states driven by the independently sampled noises $x_0, \xi_t \sim \mathcal{D}$, for all $t$, and the pair of control gain matrix $(K, \overline{L}(K))$, where $\overline{L}(K)$ is the approximate inner-loop solution. Recall that

$$\left\|diag\left(x_0^j(x_0^j)^\top, \xi_0^j(\xi_0^j)^\top, \cdots, \xi_{N-1}^j(\xi_{N-1}^j)^\top\right)\right\|_F \leq c_0 \text{ almost surely and}$$

$$diag\left(x_0^j(x_0^j)^\top, \cdots, x_N^j(x_N^j)^\top\right) = \sum_{t=0}^{N-1} (A_{K,\overline{L}(K)})^t \cdot diag\left(x_0^j(x_0^j)^\top, \xi_0^j(\xi_0^j)^\top, \cdots, \xi_{N-1}^j(\xi_{N-1}^j)^\top\right) \cdot (A_{K,\overline{L}(K)}^\top)^t,$$

then there exists a $c_{\Sigma_{K,L(K)}}$ that is a polynomial of $\|A\|_F, \|B\|_F, \|D\|_F, \|K\|_F$, and is linear in $c_0$ such that $\left\|diag\left(x_0^j(x_0^j)^\top, \cdots, x_N^j(x_N^j)^\top\right)\right\|_F \leq c_{\Sigma_{K,L(K)}}$ almost surely. Then, we apply Hoeffding's inequality to get with probability at least $1 - 2\delta_2$ that $\left\|\overline{\Sigma}_{K,\overline{L}(K)} - \Sigma_{K,\overline{L}(K)}\right\|_F \leq \sqrt{c_{\Sigma_{K,L(K)}}^2 \log(d_\Sigma/\delta_2)/[2M_2]}$. Therefore, it suffices to choose $M_2 \geq 2c_{\Sigma_{K,L(K)}}^2 \epsilon_2^{-2} \log\left(\frac{4d_\Sigma}{\delta_2}\right)$ to ensure that $\|\overline{\Sigma}_{K,\overline{L}(K)} - \Sigma_{K,\overline{L}(K)}\|_F \leq \epsilon_2/2$ with probability at least $1 - \delta_2/2$.

For the second term on the RHS of (C.12), we have by Theorems 4.1 or B.4, depending on whether the inner-loop oracle is implemented with the ZO-PG (4.1) or the ZO-NPG (4.2) updates, that with probability at least $1 - \delta_1$: $\|\overline{L}(K) - L(K)\|_F^2 \leq \lambda_{\min}^{-1}(H_{K,L(K)}) \cdot \epsilon_1$. Then, we apply Lemma C.2 to get when $\|\overline{L}(K) - L(K)\|_F \leq \mathcal{B}_{1,L(K)}$, it holds that $\|\Sigma_{K,\overline{L}(K)} - \Sigma_{K,L(K)}\|_F \leq \mathcal{B}_{\Sigma,L(K)}\|\overline{L}(K) - L(K)\|_F$ with probability at least $1 - \delta_1$. Thus, if

$$\epsilon_1 \leq \min\left\{\mathcal{B}_{1,L(K)}^2 \lambda_{\min}(H_{K,L(K)}), \frac{\epsilon_2^2 \lambda_{\min}(H_{K,L(K)})}{4\mathcal{B}_{\Sigma,L(K)}^2}\right\}, \quad \delta_1 \leq \frac{\delta_2}{2},$$

then we can bound $\|\Sigma_{K,\overline{L}(K)} - \Sigma_{K,L(K)}\|_F \leq \epsilon_2/2$ with probability at least $1 - \delta_2/2$. Combining two terms together, we can conclude that $\|\overline{\Sigma}_{K,\overline{L}(K)} - \Sigma_{K,L(K)}\|_F \leq \epsilon_2$ with probability at least $1 - \delta_2$. Then, by the Weyl's Theorem, we can bound $\lambda_{\min}(\overline{\Sigma}_{K,\overline{L}(K)}) \geq \phi/2$ if $\epsilon_2 \leq \phi/2$. Lastly, we prove Lemma C.2, which mostly follows from Lemma 16 of [11], with our $A - BK, D, Q + K^\top R^u K, L$ matrices being replaced by the $A, B, Q, K$ therein, respectively, except that the upper bound for $\|\Sigma_K\|$ therein does not hold in our setting since we only require $Q \geq 0$ and thus $Q + K^\top R^u K$ may not be full-rank. Instead, we utilize that the value of the objective function following (4.1) or (4.2) is monotonically non-decreasing, as shown in Theorems 4.1 and B.4. Moreover, the superlevel set $\mathcal{L}_K(\mathcal{G}(K, L_0))$ following the definition in (3.12) is compact. Thus, there exists a constant $c_{\Sigma_{\mathcal{G}(K,L_0)}} := \max_{L \in \mathcal{L}_K(\mathcal{G}(K,L_0))} \|\Sigma_{K,L}\|$ depending on $K$ such that for a fixed $K \in \mathcal{K}$, $\|\Sigma_{K,L}\|_F \leq c_{\Sigma_{\mathcal{G}(K,L_0)}}$ holds for all iterates of $L$ following (4.1) or (4.2) until convergence of the inner loop. This implies that for a fixed $K \in \mathcal{K}$, if $\|L' - L\|_F \leq \mathcal{B}_{1,L} := \phi/[4c_{\Sigma_{\mathcal{G}(K,L_0)}}\|D\|_F(\|A - BK - DL\|_F + 1)]$, then it holds that $\|\Sigma_{K,L'} - \Sigma_{K,L}\|_F \leq \mathcal{B}_{\Sigma,L} \cdot \|L' - L\|_F$, where $\mathcal{B}_{\Sigma,L} := 4c_{\Sigma_{\mathcal{G}(K,L_0)}}^2 \|D\|_F(\|A - BK - DL\|_F + 1)/\phi$. This completes the proof. ∎

## C.9 Auxiliary Bounds

Define the following polynomials of $\mathcal{G}(K, L(K))$:

$$c_{1,K} := \frac{\mathcal{G}(K, L(K))}{\phi} + \frac{\|D\|_F^2 \mathcal{G}(K, L(K))^2}{\phi^2 \cdot \lambda_{\min}(H_{K,L(K)})}, \qquad c_{2,K} := \sqrt{\frac{\|G_{K,L(K)}\|}{\phi} \cdot \left(\mathcal{G}(K, L(K)) - \mathcal{G}(K^*, L^*)\right)},$$

$$c_{3,K} := \frac{c_{2,K} + \|B\|_F\|A\|_F c_{1,K}}{\lambda_{\min}(R^u)}, \qquad c_{4,K} := \frac{\|D\|_F \mathcal{G}(K, L(K))}{\phi \cdot \lambda_{\min}(H_{K,L(K)})}\left(\|A\|_F + \|B\|_F c_{3,K}\right),$$

$$c_{5,K} := 2\Bigg[\|R^u\|_F + \|B\|_F^2 \mathcal{G}(K, L(K)) + \|B\|_F\|D\|_F \mathcal{G}(K, L(K))\mathcal{B}_{L(K),K}$$

$$+ \mathcal{B}_{P,K}\|B\|_F\left(\|B\|_F\left(c_{3,K} + \mathcal{B}_{1,K}\right) + \|A\|_F + \|D\|_F(c_{4,K} + \mathcal{B}_{L(K),K})\right)\Bigg],$$

where $\mathcal{B}_{1,K}, \mathcal{B}_{P,K}$, and $\mathcal{B}_{L(K),K}$ follow from Lemma B.7. Then, we present the following lemmas.

**Lemma C.3** *For $K, K' \in \mathcal{K}$, the following inequalities hold*

$$\|P_{K,L(K)}\|_F \le \mathcal{G}(K,L(K))/\phi, \quad \text{(C.13)}$$

$$\|\widetilde{P}_{K,L(K)}\|_F \le c_{1,K}, \quad \text{(C.14)}$$

$$\|F_{K,L(K)}\|_F \le c_{2,K}, \quad \text{(C.15)}$$

$$\|\nabla_K \mathcal{G}(K,L(K))\|_F \le 2\|\Sigma_{K,L(K)}\| c_{2,K}, \quad \text{(C.16)}$$

$$\|K\|_F \le c_{3,K}, \quad \text{(C.17)}$$

$$\|L(K)\|_F \le c_{4,K}. \quad \text{(C.18)}$$

**Proof** To start with, we note that for any $K \in \mathcal{K}$, the solution to the Riccati equation (3.8) satisfies $P_{K,L(K)} \ge 0$. Thus, (C.13) can be proved by $\mathcal{G}(K,L(K)) = \mathrm{Tr}(P_{K,L(K)}\Sigma_0) \ge \phi\|P_{K,L(K)}\|_F$, where the inequality follows from $\phi = \lambda_{\min}(\Sigma_0) > 0$ and $\mathrm{Tr}(M) \ge \|M\|_F$ for any matrix $M \ge 0$. Then, for any $K \in \mathcal{K}$, we have by (C.13) and the definition of $\widetilde{P}_{K,L(K)}$ in (3.7) that

$$\left\|\widetilde{P}_{K,L(K)}\right\|_F \le \left\|P_{K,L(K)}\right\|_F + \left\|P_{K,L(K)}DH_{K,L(K)}^{-1}D^\top P_{K,L(K)}\right\|_F \le \frac{\mathcal{G}(K,L(K))}{\phi} + \frac{\|D\|_F^2 \mathcal{G}(K,L(K))^2}{\phi^2 \cdot \lambda_{\min}(H_{K,L(K)})} = c_{1,K}.$$

This completes the proof of (C.14). Subsequently, for any $K, K' \in \mathcal{K}$, we invoke Lemma 11 of [11] to get

$$\mathcal{G}(K,L(K)) - \mathcal{G}(K^*,L^*) \ge \mathcal{G}(K,L(K)) - \mathcal{G}(K',L(K')) \ge \frac{\phi}{\|G_{K,L(K)}\|}\mathrm{Tr}(F_{K,L(K)}^\top F_{K,L(K)}).$$

As a result, we have $\|F_{K,L(K)}\|_F^2 = \mathrm{Tr}(F_{K,L(K)}^\top F_{K,L(K)}) \le \frac{\|G_{K,L(K)}\|}{\phi}\big(\mathcal{G}(K,L(K)) - \mathcal{G}(K^*,L^*)\big)$, and taking a square root of both sides yields (C.15). Moreover, we can prove (C.16) such that

$$\|\nabla_K \mathcal{G}(K,L(K))\|_F^2 \le 4\|\Sigma_{K,L(K)}\|^2 \mathrm{Tr}(F_{K,L(K)}^\top F_{K,L(K)}) \le \frac{4\|\Sigma_{K,L(K)}\|^2 \|G_{K,L(K)}\|}{\phi}\big(\mathcal{G}(K,L(K)) - \mathcal{G}(K^*,L(K^*))\big).$$

Next, we invoke Lemma 25 of [11] and (C.14) to get

$$\|K\|_F \le \frac{c_{2,K}}{\lambda_{\min}(R^u)} + \frac{\left\|B^\top \widetilde{P}_{K,L(K)}A\right\|_F}{\lambda_{\min}(R^u)} \le \frac{c_{2,K} + \|B\|_F\|A\|_F c_{1,K}}{\lambda_{\min}(R^u)} = c_{3,K},$$

which proves (C.17). Lastly, for any $K \in \mathcal{K}$ and recall the definition of $L(K)$ from (3.10)

$$\|L(K)\|_F = \|H_{K,L(K)}^{-1} D^\top P_{K,L(K)}(A - BK)\|_F \le \frac{\|D\|_F \mathcal{G}(K,L(K))}{\phi \cdot \lambda_{\min}(H_{K,L(K)})}\big(\|A\|_F + \|B\|_F c_{3,K}\big) = c_{4,K},$$

which proves (C.18). ∎

**Lemma C.4** *For $K, K' \in \mathcal{K}$ satisfying $\|K' - K\|_F \le \mathcal{B}_{1,K}$, where $\mathcal{B}_{1,K}$ is as defined in Lemma B.7, it holds that*

$$\|F_{K',L(K')} - F_{K,L(K)}\|_F \le c_{5,K} \cdot \|K' - K\|_F. \quad \text{(C.19)}$$

**Proof** For $K, K' \in \mathcal{K}$ and recalling the definition of $F_{K,L}$ in (3.6), we have

$$\|F_{K',L(K')} - F_{K,L(K)}\|_F$$

$$= 2\left\|(R^u + B^\top P_{K',L(K')}B)K' - B^\top P_{K',L(K')}(A - DL(K')) - (R^u + B^\top P_{K,L(K)}B)K - B^\top P_{K,L(K)}(A - DL(K))\right\|_F$$

$$\le 2\|R^u\|_F\|K' - K\|_F + 2\|B\|_F^2\|K'\|_F\|P_{K',L(K')} - P_{K,L(K)}\|_F + 2\|B\|_F^2\|P_{K,L(K)}\|_F\|K' - K\|_F$$

$$\quad + 2\|B\|_F\|A\|_F\|P_{K',L(K')} - P_{K,L(K)}\|_F + 2\|B\|_F\|D\|_F\|L(K')\|_F\|P_{K',L(K')} - P_{K,L(K)}\|_F$$

$$\quad + 2\|B\|_F\|P_{K,L(K)}\|_F\|D\|_F\|L(K') - L(K)\|_F$$

$$= 2\big(\|R^u\|_F + \|B\|_F^2\|P_{K,L(K)}\|_F\big) \cdot \|K' - K\|_F + 2\big(\|B\|_F\|P_{K,L(K)}\|_F\|D\|_F\big) \cdot \|L(K') - L(K)\|_F$$

$$\quad + 2\Big(\|B\|_F^2\big(\|K\|_F + \|K' - K\|_F\big) + \|B\|_F\|A\|_F + \|B\|_F\|D\|_F\big(\|L(K)\|_F + \|L(K') - L(K)\|_F\big)\Big) \cdot \|P_{K',L(K')} - P_{K,L(K)}\|_F.$$

Applying (C.13), (C.17), (C.18), (B.23), and (B.24) proves (C.19), such that

$$\|F_{K',L(K')} - F_{K,L(K)}\|_F$$

$$\leq 2\Big(\|R^u\|_F + \|B\|_F^2 \mathcal{G}(K,L(K))\Big) \cdot \|K' - K\|_F + 2\Big(\|B\|_F\|D\|_F \mathcal{G}(K,L(K))\Big) \cdot \|L(K') - L(K)\|_F$$

$$+ 2\Big(\|B\|_F^2\big(c_{3,K} + \mathcal{B}_{1,K}\big) + \|B\|_F\|A\|_F + \|B\|_F\|D\|_F(c_{4,K} + \mathcal{B}_{L(K),K})\Big) \cdot \|P_{K',L(K')} - P_{K,L(K)}\|_F$$

$$\leq 2\Big[\|R^u\|_F + \|B\|_F^2 \mathcal{G}(K,L(K)) + \|B\|_F\|D\|_F \mathcal{G}(K,L(K))\mathcal{B}_{L(K),K}$$

$$+ \mathcal{B}_{P,K}\|B\|_F\Big(\|B\|_F\big(c_{3,K} + \mathcal{B}_{1,K}\big) + \|A\|_F + \|D\|_F(c_{4,K} + \mathcal{B}_{L(K),K})\Big)\Big] \cdot \|K' - K\|_F.$$

This completes the proof. ∎

# D Simulation Results

In this section, we provide simulation results to complement our theories. In particular, we present convergence (divergence) results in the following four scenarios where we apply PG updates in Algorithms 1 and 2 to solve a finite-horizon zero-sum LQ dynamic game with time-invariant system parameters: (i) double-loop update scheme with both the inner- and outer-loop solved exactly; (ii) double-loop scheme with the inner-loop solved in the derivative-free scheme by sampling system trajectories while the outer-loop solved exactly; (iii) double-loop scheme with the inner-loop solved exactly while the outer-loop solved in the derivative-free scheme by sampling system trajectories; (iv) descent-multi-step-ascent/AGDA/$\tau$-GDA with exact gradient accesses diverge. Subsequently, we extend our numerical experiments to a setting with *time-varying* system parameters in §D.5, and demonstrate the convergence results of double-loop PG updates.

**Simulation Setup.** All the experiments are executed on a desktop computer equipped with a 3.7 GHz Hexa-Core Intel Core i7-8700K processor with Matlab R2019b. The device also has two 8GB 3000MHz DDR4 memories and a NVIDIA GeForce GTX 1080 8GB GDDR5X graphic card. In all settings except the one used in §D.5, we set the horizon of the problem to $N = 5$ and test a linear time-invariant system with the set of system matrices being $A_t = A$, $B_t = B$, $D_t = D$, $Q_t = Q$, $R_t^u = R^u$, and $R_t^w = R^w$, where $R^w = 5 \cdot I$ and

$$A = \begin{bmatrix} 1 & 0 & -5 \\ -1 & 1 & 0 \\ 0 & 0 & 1 \end{bmatrix}, \quad B = \begin{bmatrix} 1 & -10 & 0 \\ 0 & 3 & 1 \\ -1 & 0 & 2 \end{bmatrix}, \quad D = \begin{bmatrix} 0.5 & 0 & 0 \\ 0 & 0.2 & 0 \\ 0 & 0 & 0.2 \end{bmatrix},$$

$$Q = \begin{bmatrix} 2 & -1 & 0 \\ -1 & 2 & -1 \\ 0 & -1 & 2 \end{bmatrix}, \quad R^u = \begin{bmatrix} 4 & -1 & 0 \\ -1 & 4 & -2 \\ 0 & -2 & 3 \end{bmatrix}. \tag{D.1}$$

## D.1 Exact Double-Loop Updates

We set $\Sigma_0 = I$. Then, we present the convergence result following two different initializations, denoted as $K_0^1$ and $K_0^2$, where $K_0^2$ is closer to the boundary of $\mathcal{K}$. Specifically, $K_0^i = \begin{bmatrix} diag((K_0^i)^5) & \mathbf{0}_{15\times3} \end{bmatrix}$, for $i \in \{1, 2\}$, where $K_0^1 = \begin{bmatrix} -0.12 & -0.01 & 0.62 \\ -0.21 & 0.14 & 0.15 \\ -0.06 & 0.05 & 0.42 \end{bmatrix}$ and $K_0^2 = \begin{bmatrix} -0.14 & -0.04 & 0.62 \\ -0.21 & 0.14 & 0.15 \\ -0.06 & 0.05 & 0.42 \end{bmatrix}$. Then, we can verify that $\lambda_{\min}(H_{K_0^1, L(K_0^1)}) = 0.5041$ and $\lambda_{\min}(H_{K_0^2, L(K_0^2)}) = 0.0199$. Moreover, we initialize $L = \mathbf{0}$ in both cases.

**Case 1:** We demonstrate in Figure 4 the convergence of three update combinations, namely PG-NPG, NPG-NPG, and GN-GN, for the inner and outer loop, respectively. The stepsizes are chosen to be $(\eta, \alpha) = (1 \times 10^{-4}, 3 \times 10^{-6})$ for PG-NPG, $(\eta, \alpha) = (0.0635, 3 \times 10^{-6})$ for NPG-NPG, and $(\eta, \alpha) = (0.5, 5 \times 10^{-4})$ for GN-GN. Also, we require the approximate inner-loop solution to have the accuracy of $\epsilon_1 = 0.001$. As shown in the top of Figure 4, the double-loop algorithm with all

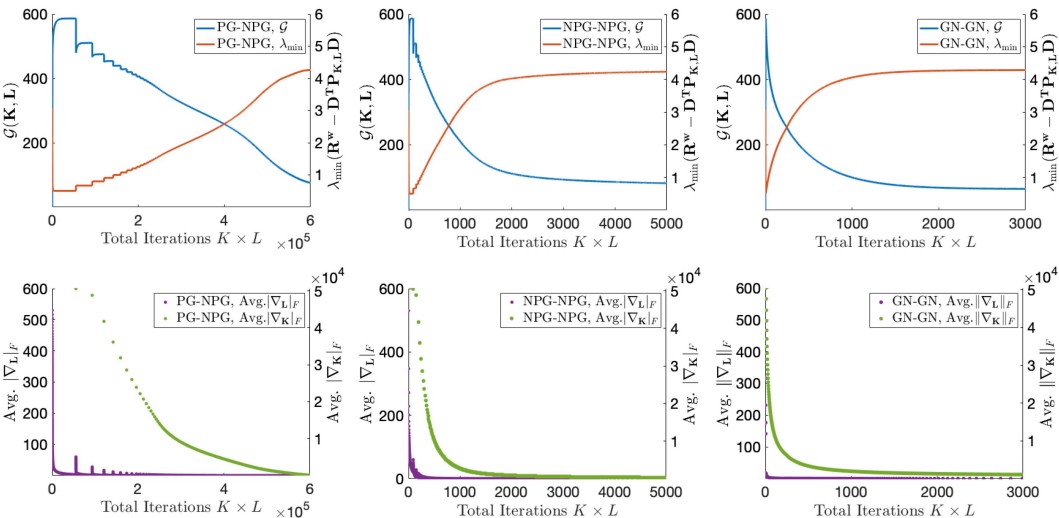

Figure 4: Three exact double-loop PG updates with initial gain matrix being $K_0^1$. *Top:* Convergence of $\mathcal{G}(K, L)$ and $\lambda_{\min}(H_{K,L})$. *Bottom:* Convergence of $\{l^{-1} \sum_{\iota=0}^{l-1} \|\nabla_L(K, L_\iota)\|_F\}_{l \geq 1}$ and $\{k^{-1} \sum_{\kappa=0}^{k-1} \|\nabla_K \mathcal{G}(K_\kappa, \overline{L}(K_\kappa))\|_F\}_{k \geq 1}$. $\overline{L}(K)$ is the approximate inner-loop solution with $\epsilon_1 = 10^{-3}$.

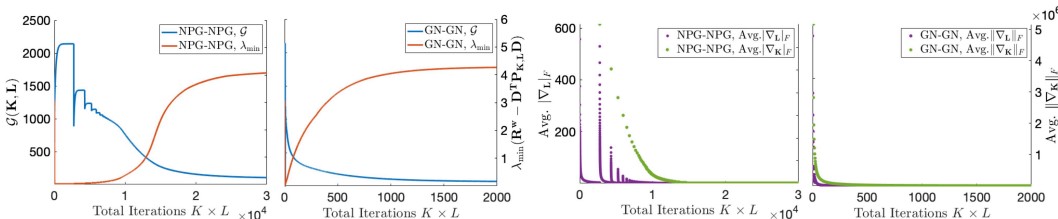

Figure 5: Two exact double-loop PG updates with initial gain matrix being $K_0^2$. *Left:* Convergence of of $\mathcal{G}(K, L)$ and $\lambda_{\min}(H_{K,L(K)})$. *Right:* Convergence of $\{l^{-1} \sum_{\iota=0}^{l-1} \|\nabla_L(K, L_\iota)\|_F\}_{l \geq 1}$ and $\{k^{-1} \sum_{\kappa=0}^{k-1} \|\nabla_K \mathcal{G}(K_\kappa, \overline{L}(K_\kappa))\|_F\}_{k \geq 1}$. $\overline{L}(K)$ is the approximate inner-loop solution with $\epsilon_1 = 10^{-3}$.

three update combinations successfully converges to the unique Nash equilibrium of the game. Also, the sequence $\{\lambda_{\min}(H_{K_k,L(K_k)})\}_{k \geq 0}$ is monotonically non-decreasing, which matches the implicit regularization property we introduced in Theorem 3.7 for the outer-loop NPG and GN updates. As $K_0^1 \in \mathcal{K}$, we can guarantee that all future iterates will stay inside $\mathcal{K}$. The convergences of the average gradient norms with respect to both $L$ and $K$ are also presented in the bottom of Figure 4.

**Case 2:** When the initial gain matrix $K_0^2$ is closer to the boundary of $\mathcal{K}$, one non-judicious update could easily drive the gain matrix outside of $\mathcal{K}$. In this case, we present two combinations of updates, namely NPG-NPG and GN-GN, for the inner and outer loop, respectively. We use the stepsizes $(\eta, \alpha) = (0.0635, 2.48 \times 10^{-7})$ for NPG-NPG and $(\eta, \alpha) = (0.5, 2.5 \times 10^{-4})$ for GN-GN. Similarly, we set $\epsilon_1 = 0.001$. The convergence patterns presented in Figure 5 are similar to those of **Case 1**.

### D.2 Derivative-Free Inner-Loop Oracles

We provide two sets of experiments to validate Theorems 4.1 and B.4. We consider the same example as the one in §D.1, but using derivative-free updates (4.1)-(4.2) by sampling system trajectories to approximately solve the inner-loop subproblem, as proposed in §4.1. Note that the inner-loop subproblem is essentially an indefinite LQR, as introduced in §3. The outer-loop problem is solved using the exact NPG update (3.17). Note that according to Theorem 4.3, one can also use the

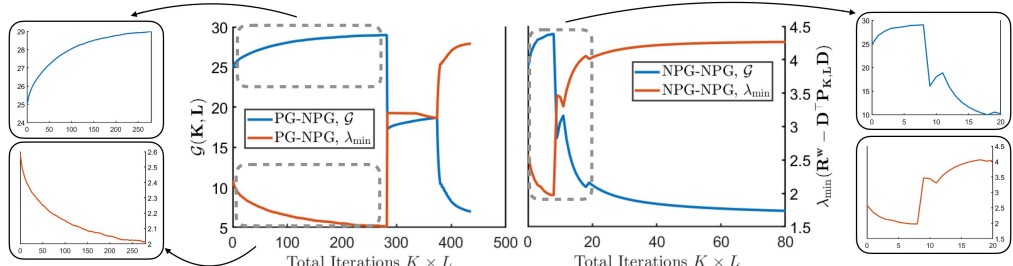

Figure 6: Convergence of $\mathcal{G}(K, L)$ and $\lambda_{\min}(H_{K,L(K)})$ with the initial gain matrix being $K_0^3$. *Left:* The inner loop follows the ZO-PG update (4.1) and the outer loop follows the exact NPG update (3.17). *Right:* The inner loop follows the ZO-NPG update (4.2) and the outer loop follows the exact NPG update (3.17).

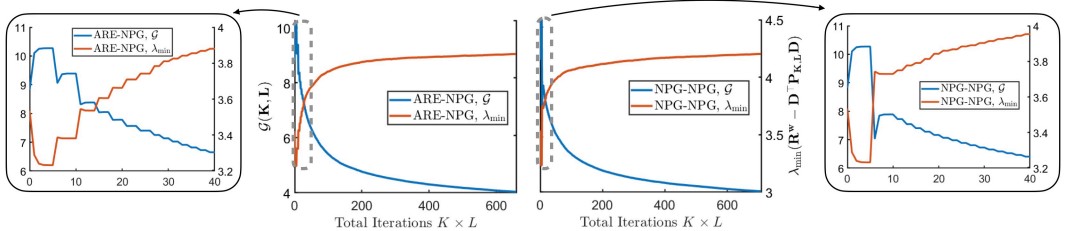

Figure 7: Convergence of $\mathcal{G}(K, L)$ and $\lambda_{\min}(H_{K,L})$ when the outer loop follows the ZO-NPG update (4.3). *Left:* The inner-loop solutions are computed by (3.10). *Right:* The inner-loop solutions are approximated by the exact NPG update (3.14) with an accuracy level of $\epsilon_1 = 10^{-4}$.

outer-loop ZO-NPG (4.3) together with the inner-loop derivative-free updates to achieve the same convergence pattern. However, we use an exact outer-loop update, as it suffices to illustrate our idea here and also has better computational efficiency. In contrast to the example in §D.1, we set

$$\Sigma_0 = 0.1 \cdot I \text{ and initialize } K_0^3 = \begin{bmatrix} diag((K_0^3)^5) & \mathbf{0}_{15\times 3} \end{bmatrix} \text{ with } K_0^3 = \begin{bmatrix} -0.04 & -0.01 & 0.61 \\ -0.21 & 0.15 & 0.15 \\ -0.06 & 0.05 & 0.42 \end{bmatrix}, \text{ where one}$$

can verify that $\lambda_{\min}(H_{K_0^3, L(K_0^3)}) = 1.8673 > 0$. That is, $K_0^3 \in \mathcal{K}$.

**Inner-Loop ZO-PG:** When we use the inner-loop ZO-PG update (4.1) in Algorithm 1, we choose $M_1 = 10^6$ and $r_1 = 1$. Moreover, we set the stepsizes of the inner-loop ZO-PG and the outer-loop exact NPG to be $(\eta, \alpha) = (8 \times 10^{-3}, 4.5756 \times 10^{-4})$, respectively. The desired accuracy level of the double-loop algorithm is picked to be $(\epsilon_1, \epsilon_2) = (0.8, 0.5)$. As shown in the left of Figure 6, the inner-loop ZO-PG (4.2) successfully converges for every fixed outer-loop update, validating our results in Theorem 4.1. Also, the double-loop algorithm converges to the unique Nash equilibrium of the game.

**Inner-Loop ZO-NPG:** When the inner-loop subproblem is solved via the ZO-NPG update (4.2), we choose the same $M_1, r_1, \alpha, \epsilon_1, \epsilon_2$ as the ones used in the inner-loop ZO-PG update but in contrast we set $\eta = 5 \times 10^{-2}$. Similar convergence patterns can be observed in the right of Figure 6.

### D.3 Derivative-Free Outer-Loop NPG

We set $\Sigma_0 = 0.05 \cdot I$. We present convergence results following the initial gain matrix $K_0 = \begin{bmatrix} diag(K^5) & \mathbf{0}_{15\times 3} \end{bmatrix}$, where $K = \begin{bmatrix} -0.08 & 0.35 & 0.62 \\ -0.21 & 0.19 & 0.32 \\ -0.06 & 0.10 & 0.41 \end{bmatrix}$ and $\lambda_{\min}(H_{K_0, L(K_0)}) = 3.2325$. The convergence of Algorithm 2 is implemented for two cases, with the inner-loop oracle being implemented using (i) the exact solution as computed by (3.10); and (ii) the approximate inner-loop solution following

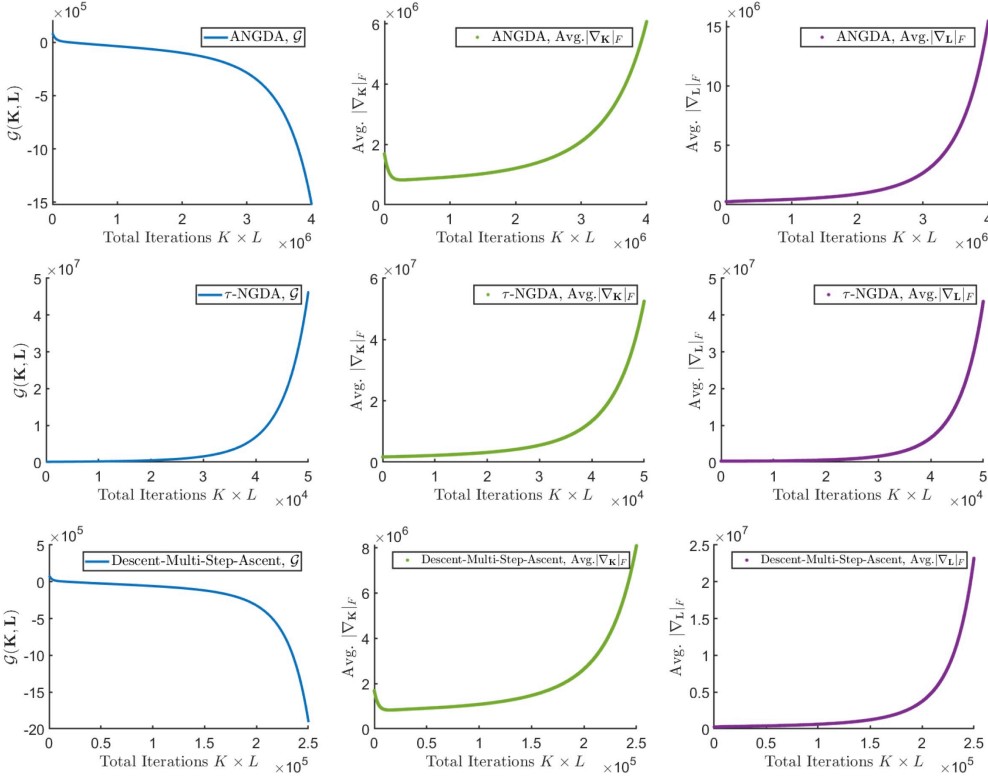

Figure 8: Divergence of $\mathcal{G}(K, L)$, $\{l^{-1}\sum_{l=0}^{l-1}\|\nabla_L(K, L_l)\|_F\}_{l\geq 1}$, and $\{k^{-1}\sum_{\kappa=0}^{k-1}\|\nabla_K\mathcal{G}(K_\kappa, \overline{L}(K_\kappa))\|_F\}_{k\geq 1}$. *Top:* ANGDA with stepsizes $\eta = \alpha = 1.7319\times 10^{-11}$. *Middle:* $\tau$-NGDA with stepsizes $\alpha = 1.7319\times 10^{-11}$ and $\eta = 10^3 \cdot \alpha$. *Bottom:* Descent-multi-step-ascent with update rules (3.14) and (3.17) and stepsizes $\eta = \alpha = 1.7319\times 10^{-9}$. For each iteration of the outer-loop update (3.17), we run 10 iterations of the inner-loop update (3.14).

the exact NPG update (3.14). As proved by Theorems 4.1 and B.4, the inner loop can also be solved approximately using either ZO-PG or ZO-NPG updates, which has been verified via the simulation results presented in §D.2. The parameters of Algorithm 2 in both cases are set to $M_2 = 5\times 10^5$, $r_2 = 0.08$, $(\eta, \alpha) = (0.1, 4.67\times 10^{-5})$, and $(\epsilon_1, \epsilon_2) = (10^{-4}, 0.8)$. Figure 7 illustrates the behaviors of the objective function and the smallest eigenvalue of $H_{K,L}$ following the double-loop update scheme. It is shown that iterates of the outer-loop gain matrix $\{K_k\}_{k\geq 0}$ stay within the feasible set $\mathcal{K}$ along with the double-loop update, which preserves a certain disturbance attenuation level in the view of Remark A.8. Moreover, in both cases, we have Algorithm 2 converging sublinearly to the unique Nash equilibrium.

## D.4 Divergent Cases

As noted in Remark A.7, it is unclear yet if descent-multi-step-ascent, AGDA, or $\tau$-GDA, where $\tau = \eta/\alpha$, can converge globally to the Nash equilibrium in our setting. In this section, we present some scenarios where descent-multi-step-ascent, AGDA, and $\tau$-GDA diverges even with infinitesimal stepsizes. We use the same example as §D.1 and initialize $K_0^4, L_0^4$ to be time-invariant such that

$$K_{0,t}^4 = K_0^4 = \begin{bmatrix} -0.1362 & 0.0934 & 0.6458 \\ -0.2717 & -0.1134 & -0.4534 \\ -0.6961 & -0.9279 & -0.6620 \end{bmatrix} \text{ and } L_{0,t}^4 = L_0^4 = \begin{bmatrix} 0.2887 & -0.2286 & 0.4588 \\ -0.7849 & -0.1089 & -0.3755 \\ -0.2935 & 0.9541 & 0.7895 \end{bmatrix} \text{ for all } t.$$

Note that in descent-multi-step-ascent/AGDA/$\tau$-GDA, the maximizing problem with respect to $L$ for a fixed $K$ is no longer solved to a high accuracy for each iteration of the updates on $K$. Thus, we can relax the constraint $K \in \mathcal{K}$ since the maximizing player will not drive the cost to $\infty$ in such iterates under descent-multi-step-ascent/AGDA/$\tau$-GDA. Figure 8 illustrates the behaviors of

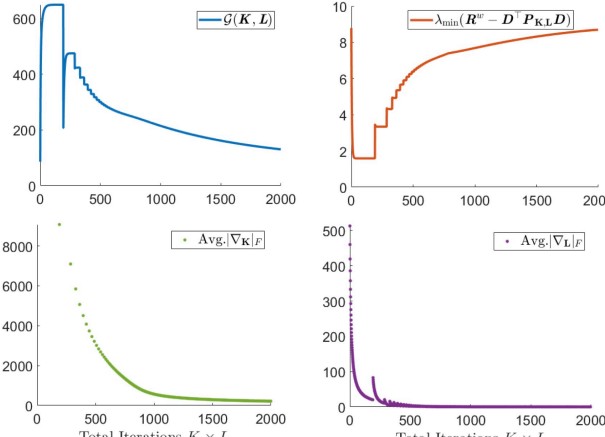

Figure 9: Convergence of the exact double-loop NPG updates with initial gain matrix being $K_0^5$. *Top:* Convergence of $\mathcal{G}(K,L)$ and $\lambda_{\min}(H_{K,L})$. *Bottom:* Convergence of $\{k^{-1}\sum_{\kappa=0}^{k-1}\|\nabla_K\mathcal{G}(K_\kappa,\overline{L}(K_\kappa))\|_F\}_{k\geq1}$ and $\{l^{-1}\sum_{l=0}^{l-1}\|\nabla_L(K,L_l)\|_F\}_{l\geq1}$. The stepsizes of the inner-loop and the outer-loop NPG updates are chosen to be $0.0097$ and $3.0372\times10^{-5}$, respectively. For each iteration of the outer-loop NPG update, we solve the inner-loop subproblem to the accuracy of $\epsilon_1 = 0.001$.

the objective and the gradient norms with respect to $K$ and $L$ when applying alternating natural GDA (ANGDA), $\tau$-natural GDA ($\tau$-NGDA), and descent-multi-step-ascent with updates (3.14) and (3.17) to our problem. The stepsizes for the ANGDA are chosen to be infinitesimal such that $\eta = \alpha = 1.7319\times10^{-11}$. However, we can still observe the diverging patterns from the top row of Figure 8, even with such tiny stepsizes. Further, we test the same example with the same initialization but use $\tau$-NGDA with $\tau = 10^3$. That is, $\eta = 10^3\cdot\alpha$. In such case, it is shown in the middle row of Figure 8 that diverging behaviors still appear. Whether there exists a finite timescale separation $\tau^*$, similar to the one proved in [74], such that $\tau$-GDA type algorithms with $\tau \in (\tau^*,\infty)$ provably converge to the unique Nash equilibrium in our setting requires further investigation, and is left as our future work. Lastly, we present a case where descent-multi-step-ascent with updates following (3.14) and (3.17) diverges. The stepsizes of the inner-loop and the outer-loop updates are chosen to be $\eta = \alpha = 1.7319\times10^{-9}$ and we run 10 iterations of (3.14) for each iteration of (3.17). The bottom row of Figure 8 demonstrates the diverging behaviors of this update scheme.

### D.5 Time-Varying Systems

Instead of setting $A_t = A$, $B_t = B$, and $D_t = D$ for all $t$, we now choose

$$A_t = A + \frac{(-1)^t tA}{10}, \quad B_t = B + \frac{(-1)^t tB}{10}, \quad D_t = D + \frac{(-1)^t tD}{10},$$

and set $R^w = 10\cdot I$. Rest of the parameters are set to the same as in (D.1). The horizon of the problem is again set to $N = 5$ and we initialize our algorithm using $K_{0,t}^5 = K_0^5 = \begin{bmatrix} -0.0984 & -0.7158 & -0.1460 \\ -0.1405 & 0.0039 & 0.4544 \\ -0.1559 & -0.7595 & 0.7403 \end{bmatrix}$.

We demonstrate in Figure 9 the convergence of exact double-loop NPG updates. The stepsizes of the inner-loop and the outer-loop updates are chosen to be $(\eta,\alpha) = (0.0097, 3.0372\times10^{-5})$. Also, we require the approximate inner-loop solution to have the accuracy of $\epsilon_1 = 0.001$. As shown at the top of Figure 9, the double-loop NPG updates successfully converge to the unique Nash equilibrium of the game. Also, the minimum eigenvalue of $R^w - D^\top P_{K,L}D$ is monotonically non-decreasing along the iterations of the outer-loop NPG update, which matches the implicit regularization property we introduced in Theorem 3.7. This guarantees that all future iterates of $K$ will stay in the interior of $\mathcal{K}$ given that the initial $K$ does. The convergences of the average gradient norms with respect to both $L$ and $K$ are also presented at the bottom of Figure 9.