# OpenReview forum: "Derivative-Free Policy Optimization for Linear Risk-Sensitive and Robust Control Design: Implicit Regularization and Sample Complexity"
_NeurIPS.cc/2021/Conference — NeurIPS 2021 Poster_

### Official Review · Reviewer_DsD5 · 2021-07-04

**Rating:** 7
**Confidence:** 3

**Summary:**

The paper analyzes policy gradient methods in the context of the Linear Exponential Quadratic Gaussian and the Linear Quadratic Disturbance Attenuation frameworks. It provides a reduction from these frameworks to a 2 player LQR game, and shows that given a polynomial number of sampled trajectories, one can solve this game and hence the original frameworks.

**Limitations And Societal Impact:**

Yes they have. Thank you for doing this.

**Main Review:**

I should say that I am coming at this as someone who is very familiar with optimization, especially with zero order/bandit optimization. While I am also familiar with the conventional LQR setup in the context of such methods, I generally am not very familiar with control theory.

My answers to the "originality, quality, clarity, and significance" depend on some questions that I have for the authors. So I will directly state my questions.

1. For the reduction described in Lemma 2.2, it seems that one requires at least some knowledge of the system. For instance, $D_t$ is a system matrix, and it seems that this is assumed to be known in order to make the reduction? A similar comment applies for $C_t$? Also, why is it reasonable that the noise covariance $W$ is known, which surely should also appear in the reduction? I agree that it is more reasonable for $R_t$ and $Q_t$ to be known, since these are typically designed by a user. But just to be clear, knowledge of these is also required?

2. It seems that the inner loop solution step is more or less completely standard. We are given access to a (locally) smooth function that satisfies the PL inequality, and there is plenty of prior work which can handle this (Fazel et al., Malik et al. and others). Note that the local smoothness causes no real issues, because the Lipschitz constants should decay as the cost decreases (i.e. as the descent algorithm progresses). Am i accurate in this?

3. The outer loop solution is quite interesting to me, and seems like the meat of this paper. As I said above, in prior work for convention LQR, the typical story is that if the cost function decreases, then the iterates’ stability/feasibility is preserved. This is no longer true here. The authors say that their proof shows that the $P_{K, L(K)}$ matrix itself is decreasing (in the PSD sense), and this is a consequence of the implicit regularization of natural gradient. Can the authors provide me with more explanation of why the decrease of $P_{K, L(K)}$ matrix is important? It seems like a more fine grained characterization of the decrease in cost.  Also, can the authors please provide more intuition (perhaps even using a few equations) for why the natural gradient has this implicit regularization property. I know that they tried to provide this in their paper, but honestly I struggled to understand it.

4. As for the conversion of the population level algorithm to a sample based derivative-free algorithm, the authors discussed a few challenges, but this confused me. Note that the authors are only offering a polynomial sample efficiency guarantee, as opposed to a fine grained characterization of the sample complexity dependency on different problem parameters. Given this, it seems to me that there shouldn’t any issues? For instance, the authors state in their Proof Sketch of Theorem 4.2 that using estimated gradients can lead to deviations in the algorithm’s iterates from the algorithm’s iterates at the population level, and that such statistical errors may blow up over time. But can’t one simply use a very large (but of course still polynomial) number of samples to obtain a very accurate approximation of the population level gradients, and then simply mimic the population level analysis? Indeed, this is what is done in Fazel et al. in the conventional LQR setting. Or is this essentially what the proof sketch is arguing?

Overall, to the best of my knowledge, I liked the paper and I think it makes a good contribution. My questions are more for my own understanding. As long as the answers to these questions are reasonable and don't raise any major red flags, I will keep my score. But I reserve the right to change my score, in the case that a different reviewer finds an issue that I agree with. So please answer the reviewers' questions in depth, since I will read the other reviews.

I listed my confidence as a 3, but it is more like 3.5.

**Time Spent Reviewing:**

5

---

> ### Author Response · Authors · 2021-08-09
> **Response to Reviewer DsD5**
>
> Thank you for your detailed and careful review. We will respond to your comments below, one by one.
>
> **For the reduction described in Lemma 2.2, it seems that one requires at least some knowledge of the system...**
>
> For the disturbance attenuation problem, we do not require explicit knowledge of $D_t$ and $C_t$. Suppose that one would like to solve the LQ disturbance attenuation (LQDA) problem by solving the equivalent zero-sum game; then, the only assumption needed is oracle-level accesses to the LQDA model (cf. line 108). In particular, for a fixed sequence of gains $\{K_t\}$ and any sequence of disturbances $\{w_t\}$, we assume that this LQDA oracle can return a $\|z\|$ as defined in line 114. Then, the LQDA oracle with an additionally injected sequence of independent Gaussian noise (with any positive definite covariance matrix) suffices to serve as the oracle for our double-loop derivative-free PG algorithms (for the stochastic game). Therefore, no explicit knowledge on the system parameters ($A_t$, $B_t$, $D_t$, $C_t$, $E_t$, $\gamma$) is needed.
>
> However, if one would like to solve the LEQG problem by solving an equivalent zero-sum game, one will need knowledge of $W$ (to build-up the black-box sampling oracle/simulator, but still, the exact value of $W$ is not revealed to the learning agent) in addition to oracle-level accesses to the original linear model (cf. line 94). Also, explicit knowledge of the parameters ($A_t$, $B_t$, $Q_t$, $R_t$, $\beta$) is not required. We would like to note that the assumption on the knowledge (and/or the availability of the estimate) of $W$ is reasonable for a large family of control applications where system dynamics and disturbance can be studied separately. For example, consider the risk sensitive control of a wind turbine. The turbine dynamics and the wind information can be gathered separately. One can identify $W$ by looking at the past wind data. Similar situations hold for many aerospace applications where the properties of process noise (e.g. wind gust) can be estimated beforehand. We will add a remark to discuss the subtlety here in our updated version. Thanks for pointing it out.
>
> **It seems that the inner loop solution step is more or less completely standard...**
>
> Your assessments are fairly accurate. Our inner-loop solution is indeed adapted from the standard descent analysis for proving global convergence and sample complexity in infinite-horizon deterministic LQR. But still, we have made extra efforts to address the difficulties brought by the “stochastics” setting and the “indefinite” state-weighting matrix, which are unique to our game-theoretic setting. Also, interestingly, we have made non-trivial contribution to identify the importance of the “non-degeneracy” of the state covariance matrix to the “landscape” of this finite-horizon LQR problem, which coincides with the independent work of [41], whose overall contribution is exactly to study our “inner-loop” setting, but with a standard “positive-definite” state-weighting matrix (see more discussion in the response to Reviewer 1dUQ). Hence, we do not see the inner-loop result as a “straightforward” one.
>
> **The outer loop solution is quite interesting to me, and seems like the meat of this paper...**
>
> Thanks for your interest in our implicit regularization results. As you have identified in your comment, the main technical challenge that we have managed to tackle in this work is the lack of coercivity of the objective function (cf. Lemma 3.5). In LQR, the objective function itself serves as a barrier function guaranteeing for the PG iterates to stay in the feasible set. In our robust control setting, the objective value can remain finite while approaching the boundary of the robustness constraint set $\mathcal{K}$, and a descent direction of the objective value may still drive the iterates out of $\mathcal{K}$, which will lead to unbounded/undefined value.
>
> Next, notice that the set $\mathcal{K}$ in (3.9) is characterized by $\bf{P_{K, L(K)}}$ and thus $\bf{P_{K, L(K)}}$ itself can serve as a barrier function, instead of the objective value, to ensure that the PG iterates will stay within $\mathcal{K}$. The remaining question is to find a decent direction of $\bf{P_{K, L(K)}}$. And yes, as you correctly pointed out, this is indeed a more “fine grained” characterization of the decrease in cost. Fortunately, the directions of natural gradient (NPG) and Gauss-Newton (GN) descents suffice. In brief, the NPG update is related to gradient over a Riemannian manifold, and the GN update is one type of quasi-Newton update, see ref. [13] in our paper for further justifications of these two updates in the context of LQR. Similar to LQR, NPG and GN updates in our setting cancels out the "Fisher information" matrix $\Sigma_{\textbf{K}, \textbf{L}}$ in the gradient formulas (cf. Lemma 3.2), which guarantees that $\bf{P_{K, L(K)}}$ monotonically non-increases in the p.s.d. sense under suitable stepsize choices.
>
> Finally, it is worth mentioning that the implicit regularization argument in [15] (numbered in our paper) can only address the outer loop property when the model is known.  The arguments in [15] rely on a specialized “perturbation technique” which may potentially generate arbitrarily small “margins". Such an argument cannot be directly applied to our model-free setting where a uniform margin is required for provable tolerance of statistical errors. Thus, we have developed a set of new techniques to establish the sample complexity results in our paper.
>
>
> **As for the conversion of the population level algorithm to a sample based derivative-free algorithm...**
>
> Your thoughts are correct. To tolerate the statistical errors over time, we have to use large number of samples (O(1/epsilon^5) in epsilon and polynomial in other parameters) to obtain a very accurate approximation of the population level gradients, similar to what has been done in [Fazel et al.] for the standard LQR. This is exactly what the sketch is arguing, but with more delicate choices of the algorithm parameters, due to the fundamentally more challenging landscape here compared to LQR. We will update our proof sketch in our updated version to better clarify our methodologies.
>
> We hope that the above explanations resolve your concerns, and thanks again for your careful reading and your efforts in the review process.

---

### Official Review · Reviewer_cU8A · 2021-07-14

**Rating:** 7
**Confidence:** 3

**Summary:**

While the machine learning community has given much attention to the LQR problem, much less attention has been given to more sophisticated control problems. This work takes a step in this direction by studying the sample complexity of derivative-free methods applied to the Linear Exponential Quadratic Gaussian (LEQG) and LQ Disturbance Attenuation problems. To solve these problems, they show a reduction of each to the LQ game setting, then propose and analyze the convergence rates and sample complexity of a derivative-free optimization (DFO) algorithm.

**Limitations And Societal Impact:**

Yes.

**Main Review:**

This work studies the application of policy gradient-type algorithms to the LEQG and LQ disturbance attenuation problems, both fundamental problems in robust control. Rather than analyzing these problems directly, it shows that there is a reduction from each of these problems to the LQ game problem, and then analyzes the LQ game problem, obtaining guarantees for the LQ game problem which then apply to the LEQG and LQ disturbance attenuation problems.

In addition to analyzing standard policy gradient methods, it also analyzes derivate-free methods that rely on estimation of the gradient from rollouts of the system’s trajectories. It provides finite-time convergence guarantees for both and shows that both types of algorithms converge to the Nash equilibrium controller. A core technical challenge in obtaining these results is that, unlike LQR where the cost blows up when the controller leaves the feasible set, here the cost may not blow up when the controller leaves the feasible set, and further regularization is needed. To handle this, the authors show that the proposed algorithm has an implicit regularization property, and ensures feasibility of the controller throughout learning.

The primary novel contribution of interest in this work is their analysis of the sample complexity of derivative-free methods to the LQ game problem and control problems more complex than LQR. To my knowledge this is the first work that gives explicit sample complexities on the number of trajectories needed to guarantee converge of derivative free methods on these problems, and in addition shows that assuming sufficiently many samples are collected, the controllers produced when running the algorithm always stabilize the system.


Pros:
- The question of sample complexity of derivate-free methods for LQR has been studied in several works now, and several works look at the convergence of pure policy gradient methods in robust control settings, but no works have yet considered the sample complexity of derivative free methods for robust control. As such, this work addresses a relevant problem that is of interest to the community.
- While the rates obtained are likely suboptimal, this work makes a non-trivial contribution in that it obtains polynomial rates, and deepens our understanding of robust control.


Cons:
- In my opinion the main novel aspect of interest in this paper are the results in Section 4. However, the sample complexity results given appear to be very suboptimal (they achieve a rate of O(1/epsilon^5) while it is known that for LQR a rate of O(1/epsilon^2) can be achieved [3]). At minimum some discussion of the optimality is required, while none is given. Furthermore, the dimensionality dependance and dependence on curvature parameters is hidden in the main text, but should be made clear (I would expect something comparable to the statements in [3]).
- In addition, I could not find a clear statement of what ’N’, the rollout length, must be. While the focus of this work is slightly different than works such as [4] that study regret for LQR in terms of the total samples taken, it is still of interest in this setting to state precisely how many total samples are needed for the optimization procedure.
- I believe the paper could be reorganized slightly in order to make clear what the primary novel contributions are. In particular, it seems that most of the results in Section 3 are closely related to existing results (e.g. [1] studies policy gradient methods for the LEQG problem and [2] studies policy gradient methods for the LQ game problem), and the main novel results of interest are in Section 4. I would suggest expanding Section 4 and shrinking Section 3 to highlight these results. In doing this, it would be useful to put the algorithms from Section 4 (e.g. Algorithm 1 and Algorithm 2 in the appendix) in the main text. Currently, when reading section 4, it is rather unclear what the actual query model is, which makes it difficult to parse the theorem statements (for example, a ‘minibatch’ is never defined in the main text, though this is an essential parameter in the algorithm and important for understanding the true sample complexity).
- Similarly, more discussion should be given on comparing to existing works and highlighting the novel aspects of this work.
- Assumption 2.1 is stated as an assumption on the two-player game problem, but due to the equivalence between the various settings also imposes a constraint on the allowable LEQG and LQ disturbance attenuation problems. How restrictive is this assumption in these settings? It should be discussed in more detail.

In summary, I believe this work does make a contribution to a problem of interest in its analysis of DFO methods for robust control, yet the suboptimality of the sample complexity and lack of clarity in the presentation of the results makes it difficult for me to recommend an accept.

------------------------------------

Update after rebuttal: I believe the authors' response to my concerns, and in particular their explanation of the O(1/epsilon^5) dependence, seems reasonable and I have raised my score accordingly.


[1] Zhang, Kaiqing, Bin Hu, and Tamer Basar. "Policy Optimization for $\mathcal {H} _2 $ Linear Control with $\mathcal {H} _\infty $ Robustness Guarantee: Implicit Regularization and Global Convergence." Learning for Dynamics and Control. PMLR, 2020.
[2] Bu, Jingjing, Lillian J. Ratliff, and Mehran Mesbahi. "Global convergence of policy gradient for sequential zero-sum linear quadratic dynamic games." arXiv preprint arXiv:1911.04672 (2019).
[3] Malik, Dhruv, et al. "Derivative-free methods for policy optimization: Guarantees for linear quadratic systems." The 22nd International Conference on Artificial Intelligence and Statistics. PMLR, 2019.
[4] Simchowitz, Max, and Dylan Foster. "Naive exploration is optimal for online lqr." International Conference on Machine Learning. PMLR, 2020.

**Time Spent Reviewing:**

3

---

> ### Author Response · Authors · 2021-08-09
> **Response to Reviewer cU8A**
>
> We really appreciate the reviewer for your detailed review, mostly positive comments, and constructive suggestions. We respond to your concerns as below, one by one.
>
> **The sample complexity results given appear to be very suboptimal...**
>
> We agree that more discussions and comparisons are needed regarding our sample complexity results. However, we believe that it might not be very fair to directly compare the order of our sample complexity with that for LQR. LQR is known to have a nice optimization landscape, where the objective function is coercive (leading to the “compactness” of sub-level sets and the “smoothness” over the sub-level sets, and enabling the use of “any” descent direction of the objective to ensure feasibility) and satisfies the gradient-domination (PL-) property. None of these properties hold true in our outer-loop problem, which leads to a much richer optimization landscape (cf. Lemma 3.5). In particular, the objective value may remain finite while approaching the boundary of the robustness constraint set $\mathcal{K}$, and a descent direction of the objective value may still drive the iterates out of $\mathcal{K}$, which will lead to unbounded/undefined value. We also note that addressing this “landscape” challenge (different from those considered in [1,2], see detailed comparison in the response below), which was established in Sec. 3, also requires novel contributions (not just Sec. 4 as the reviewer mentioned).
>
> To guarantee that the PG iterates will stay within $\mathcal{K}$ (which is necessary for the iterates to be well-defined), we need specific update rules (e.g., NPG) that follow descent directions of the matrix $\bf{P_{K, L(K)}}$ (in the p.s.d. sense), since the set $\mathcal{K}$ in (3.9) is characterized by $\bf{P_{K, L(K)}}$. In the derivative-free setting, this means that we need many samples to obtain a very accurate approximation of the exact NPG update, ensuring that $\bf{P_{K, L(K)}}$ is monotonically non-increasing in the p.s.d. sense with a high probability along iterations. For LQR, any descent direction of the objective value, which is a scalar, suffices to guarantee that the PG iterates will stay in the feasible set. Therefore, for LQR, the O(1/epsilon^2) rate is expected for the smooth (since the iterates will not leave the feasible set, which is compact) and gradient-dominated objective functions; while for our robust control setting, due to the more stringent requirements on the estimation accuracy and the non-PL objective function, the O(1/epsilon^5) rate is not that unreasonable. In fact, to our knowledge, there is no sample complexity lower-bound applicable to the outer-loop problem we consider here (stochastic nonconvex optimization with no global smoothness nor coercivity nor PL condition, with an inner-loop oracle). Thus, it seems not clear if (and how) our results are that “suboptimal”.
>
> As a first step toward studying the global convergence and sample complexity of derivative-free PG methods for risk-sensitive/robust control problems, we have focused on proving polynomial complexity bounds and did not try to “optimize” for the best dependence on the parameters, similar to those in the seminal works [11, 16, 41] in our paper. Besides, due to the technical challenges mentioned above, we have developed new proof techniques that are significantly more complicated than those presented in [3]. We believe that developing accelerated algorithms with sharper sample complexity bounds (with respect to epsilon), as well as obtaining better parameter dependences and studying the lower-bound of the setting to justify the “optimality” of the sample complexity, could all be highly non-trivial while very interesting future directions. We will add detailed comments in the updated version of the paper to discuss and compare our sample complexity results.
>
> **I could not find a clear statement of what ’N’, the rollout length, must be...**
>
> Our work studies the finite-horizon setting with length $N$, where the rollout length $N$ is a design choice, like the $Q$ and $R$ matrices. This contrasts with the existing works which consider settings with infinite-horizon (e.g., ref. [11] in our paper), where the rollout length “must be” large enough to approximate the infinite-horizon cost accurately. We will also add ref. [4] in our final version.
>
> **I believe the paper could be reorganized slightly...**
>
> We agree with you that we should slightly adjust the organization of our paper and expand the novel results in Section 4 to make them self-contained. We will bring our Algorithms back to the main text and introduce all parameters in detail in the final version, when more space is available. However, we would like to comment that our results in Section 3 also differentiate themselves from the existing works [1, 2], and are novel contributions of our work. Compared to [1, 2], which considers settings with deterministic dynamics and infinite time-horizon, Section 3 is the first result that studies the global convergence and implicit regularization of PG methods for finite-horizon stochastic risk-sensitive and robust control problems, which is less understood in the literature, and requires different landscape characterizations (see the illustration of the idea in our Figure 2, which is quite different from the techniques in [1,2]). In fact, studying PG methods for the “finite-horizon” setting (even for LQR) is still an active area and requires non-trivial research efforts, see e.g., ref. [41] in our paper, not to mention that we have considered a more challenging robust control setting.
>
> **Similarly, more discussion should be given on comparing to existing works and highlighting the novel aspects of this work.**
>
> Thanks for your suggestions. We have already highlighted our technically novel aspects in lines 32-58, in the Contribution paragraph, in the discussions after Lemma 3.5 and Theorem 4.2, and in Figure 1. We will add more detailed comparisons to differentiate our technical contributions in the final version when more space is available.
>
> **Assumption 2.1 is stated as an assumption on the two-player game problem...**
>
> By refs. [24, 25] in our paper, Assumption 2.1 is a standard assumption which suffices to guarantee the existence of a feedback Nash equilibrium of the two-player zero-sum LQ game. Besides sufficiency, Assumption 2.1 is also “almost necessary", and “quite tight" as noted in Remark 6.8 of [25]. In particular, if the sequence of matrices $R_{t}^{w} - D_t^{\top}P_{t+1}^{*}D_{t}$ for all $t \in \{0, \cdots, N-1\}$ in Assumption 2.1 admits any negative eigenvalues, then the upper value of the game becomes unbounded.
>
> Now, we answer your question of whether Assumption 2.1 is restrictive in the context of LEQG and LQ disturbance attenuation (LQDA) problems. Suppose we set the system parameters of LEQG, LQDA, and zero-sum LQ game (game) according to Lemma 2.2, and in particular set $\beta^{-1}\textbf{I}$ in LEQG, $\gamma^{2}\textbf{I}$ in LQDA, and $\textbf{R}^{\textbf{w}}$ in the game to be the same. Then these three problems are equivalent as far as their optimum solutions go (which is what we seek). Due to this equivalence, Assumption 2.1 is a sufficient and “almost necessary" condition for the existence of a solution to the equivalent LEQG/LQDA. Specifically, if the sequence of matrices $R_{t}^{w} - D_t^{\top}P_{t+1}^{*}D_{t}$ for all $t \in \{0, \cdots, N-1\}$ in Assumption 2.1 admits any negative eigenvalues, then no state-feedback controller can achieve a $\gamma$-level of disturbance attenuation in the equivalent LQDA, and no state-feedback controller can achieve a $\beta$-degree of risk-sensitivity in the equivalent LEQG. Therefore, Assumption 2.1 is not restrictive and is in fact quite tight.
>
> Again, we sincerely thank the reviewer for the detailed comments. We hope we have fully addressed your concerns.

---

> ### Comment · Reviewer_cU8A · 2021-08-17
> **Update review**
>
> I believe the authors' response is reasonable and addresses my concerns. I have raised my score accordingly.

---

### Official Review · Reviewer_1dUQ · 2021-07-14

**Rating:** 7
**Confidence:** 4

**Summary:**

In this paper, the authors established the first sample complexity result of the policy gradient method for a class of risk-sensitive linear-quadratic control problems. This class of control problems can also be viewed as a class of zero-sum two-player games.


**Main Review:**

Pros:

Compared to the seminal work (Fazel et. al 2018) which considers an LQR problem with deterministic dynamics and infinite time-horizon, the set-up in this paper (with stochastic dynamics and finite time-horizon) has more applications but is less understood in the literature.

This paper provides the first convergence and sample complexity results for this important control/game framework. I believe it will be a nice contribution to the literature of RL theory.

Cons:

Compared to the existing works ([1], [2], and [3]), the technical contributions should be further elaborated. Are there any additional technical difficulties when integrating partial ideas from [1] and [2]? To handle the stochastic dynamics and the finite time-horizon, are there any technical similarities or differences compared to [3]?

From a game perspective, the inner-loop updates for L (under a fixed K) seem to be time-consuming and not practical for certain applications. Is it possible to extend the result to an iterative framework where L is updated once which followed by one update of K?

[1] Kaiqing Zhang, Zhuoran Yang, and Tamer Basar. Policy optimization provably converges to Nash equilibria in zero-sum linear quadratic games. In Advances in Neural Information Processing Systems, pages 11602–11614, 2019.
[2] Kaiqing Zhang, Bin Hu, and Tamer Basar. Policy optimization for H2 linear control with H1 robustness guarantee: Implicit regularization and global convergence. arXiv preprint arXiv:1910.09496, 2019.
[3] Ben M Hambly, Renyuan Xu, and Huining Yang. Policy gradient methods for the noisy linear quadratic regulator over a finite horizon. arXiv preprint arXiv:2011.10300, 2020.

========after authors response=======
 Thanks for the response from the authors. I read the comments from other reviewers and the authors' responses to all reviewers. Risk-sensitive RL with robust design is a hard problem and I think this paper has a solid contribution to the theoretical front of RL. Also, I am happy with the response from the authors.

**Time Spent Reviewing:**

5

---

> ### Author Response · Authors · 2021-08-09
> **Response to Reviewer 1dUQ**
>
> We thank you for acknowledging our contributions. Detailed responses to your comments are provided as follows:
>
> **Compared to the existing works ([1], [2], and [3]), the technical contributions should be further elaborated.**
>
> Thanks for your constructive comment. We will revise our paper accordingly. In the meantime, some of the differences between our work and existing works [1][2][3] are partially provided in the following answers.
>
> **Are there any additional technical difficulties when integrating partial ideas from [1] and [2]?**
>
> Yes, there are significant technical difficulties here, and our paper has made novel contributions in developing new proof techniques. We first note that the arguments in [1] cannot be applied to our setting since the algorithms in [1] used an extra projection step which requires model information. The use of the projection step in [1] also leads to stronger assumptions which are non-standard for robust control tasks. In contrast, the “implicit regularization” argument in [2] allows one to remove the projection step in [1] and is more relevant to our problem. However, the arguments in [2] only apply to the setting with a “known” model and rely on a specialized “perturbation technique” which may potentially generate arbitrarily small “margins". Such an argument cannot be directly applied to our “model-free” setting where a uniform margin is required for provable tolerance of statistical errors. Thus, a set of new techniques are required to establish the sample complexity results in our paper. We have commented on this in lines 54-58 of our main text and will further elaborate on this point in the revised version, when more space is available.
>
>
>
>
>
> **To handle the stochastic dynamics and the finite time-horizon, are there any technical similarities or differences compared to [3]?**
>
> Yes, for a fixed outer loop gain matrix $\textbf{K} \in \mathcal{K}$, the resulting inner loop maximization problem is a finite-horizon stochastic LQR problem studied in [3] (which is an independent work that appears almost concurrently to our work), but we allow the state-weighting matrix to be “indefinite”. This generalizes the setting in [3], where the $Q$-matrix is assumed to be positive-definite. Nonetheless, our inner-loop indefinite LQR problem indeed shares some technical similarities with [3], as both works need to guarantee the “non-degeneracy” of the state covariance matrix at any time, by requiring the additive random noise in the system dynamics to have a positive-definite covariance matrix.  Our outer loop problem is subtler and more challenging than the problem considered in [3] due to the lack of coercivity of the optimization landscape. Our implicit regularization argument is mainly developed to address the feasibility of the outer loop iteration.
>
> **From a game perspective, the inner-loop updates for L (under a fixed K) seem to be time-consuming and not practical for certain applications. Is it possible to extend the result to an iterative framework where L is updated once which followed by one update of K?**
>
> As noted in Remark A.7 (lines 892-907) and Section D.4 (lines 1736-1771) in our appendix, it is not hard to construct cases where alternating gradient descent ascent (GDA) (which is the algorithm suggested by the reviewer) diverges even with infinitesimal stepsizes. This is due to the lack of “global smoothness” of the objective function in our continuous control setting, with unbounded decision spaces and objective functions. How to design more efficient policy gradient methods for LQ games remains open, and is in fact one of our ongoing research directions.

---

### Official Review · Reviewer_16Qr · 2021-07-16

**Rating:** 7
**Confidence:** 2

**Summary:**

This article studies the convergence of double-loop policy gradient methods for finite-horizon two-layer zero-sum LQ game arising from the risk-sensitive control problems. It shows that, for suitable choices of the initial guess of the outer iteration, both the natural gradient method and the Gauss-Newton method converge to the unique Nash equilibrium. The authors then extend the algorithm to derivative-free methods based on observed state trajectories. To the best of my knowledge, this is the first theoretical result on convergence of gradient-based methods for competitive multi-agent RL problems.

**Limitations And Societal Impact:**

Yes,

**Main Review:**

This article contains novel convergence results of policy gradient methods for multi-agent problems, which are important contributions to the literature. Even though the details have not been carefully checked, the theoretical results seem to be correct. I don't have any further suggestion to improve the paper.

**Time Spent Reviewing:**

1 hour

---

> ### Author Response · Authors · 2021-08-09
> **Response to Reviewer 16Qr**
>
> We thank you for your appreciation of our work! We are more than happy to address any additional questions during the discussion period.

---

### Author Response · Authors · 2021-08-09
**General Response by the Authors of Paper 1493**

We thank the reviewers for the constructive feedback. We will respond to each review separately. Here we provide a common response clarifying the unique novelty of our paper over existing works. We mainly compare our paper with the following references:

* [F18] Fazel et al. "Global convergence of policy gradient methods for the linear quadratic regulator."
* [M19] Malik et al. "Derivative-free methods for policy optimization: Guarantees for linear quadratic systems."
* [Z19a] Zhang et al.  “Policy optimization provably converges to Nash equilibria in zero-sum linear quadratic games.”
* [Z19b] Zhang et al. "Policy Optimization for $H_{2}$ Linear Control with $H_{\infty}$ Robustness Guarantee: Implicit Regularization and Global Convergence."
*[B19] Bu et al. "Global convergence of policy gradient for sequential zero-sum linear quadratic dynamic games."
*[H20] Hambly et al. “Policy gradient methods for the noisy linear quadratic regulator over a finite horizon.”

Seminal works [F18, M19] presented sample complexity results for the infinite-horizon deterministic LQR, and [H20] studied the finite-horizon stochastic LQR. The main technical challenge between our robust control setting and [F18, M19, H20] is the lack of coercivity of the objective function (cf. Lemma 3.5). In particular, the objective value may remain finite while approaching the boundary of the robustness constraint set $\mathcal{K}$, and a descent direction of the objective value may still drive the iterates out of $\mathcal{K}$, which will lead to an unbounded/undefined value. To address this “landscape” challenge in the model-based setting, [Z19a] used an extra projection step, which is restrictive since it requires model information as well as non-standard assumptions. This projection step has been removed in [B19] and [Z19b], again in the case when the model is known. Specifically, the Riccati-based arguments in [B19] are tailored for the zero-sum game setting and require assumptions that are non-standard for robust control. The implicit regularization arguments in [Z19b] align better with the robust control literature, by leveraging standard tools such as the KYP lemma. However, the arguments in [Z19b] only apply to the setting with a known model, as they rely on a specialized “perturbation technique” which may potentially generate arbitrarily small “margins" between the iterates and the boundary of the robustness feasible set. Such an argument cannot be directly applied to our model-free setting where a uniform margin is required for provable tolerance of statistical errors, when “samples” are used to estimate the policy gradients. Our work has overcome the technical challenges mentioned above, by developing the first implicit regularization results when applying model-free PG methods for robust controller designs.

Our new arguments are summarized as follows. Notice that the feasible set $\mathcal{K}$ in (3.9) is characterized by $\bf{P_{K, L(K)}}$ and thus $\bf{P_{K, L(K)}}$ itself can serve as a barrier function, instead of the objective value, to ensure that the PG iterates will stay within $\mathcal{K}$. The remaining question is to find a decent direction of $\bf{P_{K, L(K)}}$. Fortunately, the directions of natural gradient (NPG) and Gauss-Newton (GN) descents suffice. The NPG and GN updates in our setting cancels out the "Fisher information" matrix $\Sigma_{\textbf{K}, \textbf{L}}$ in the gradient formulas (cf. Lemma 3.2), which guarantees that $\bf{P_{K, L(K)}}$ monotonically non-increases in the p.s.d. sense under suitable stepsize choices.

**Comparison of our results with [M19]:**

We believe that it would not be very fair to directly compare the order of our sample complexity with that for LQR. To guarantee that the PG iterates will stay within $\mathcal{K}$ (which is necessary for the iterates to be well-defined), we need specific update rules (e.g., NPG) that follow descent directions of the matrix $\textbf{P}_{\textbf{K}, \textbf{L(K)}}$ (in the p.s.d. sense. In the derivative-free setting, this means that we need many samples to obtain a very accurate approximation of the exact NPG update. For LQR, “any” descent direction of the objective value, which is a scalar, suffices to guarantee that the PG iterates will stay in the feasible set. Therefore, for LQR, the O(1/epsilon^2) rate is expected for the smooth (since the iterates will not leave the feasible set, which is compact) and gradient-dominated objective functions; while for our robust control setting, due to the more stringent requirements on the estimation accuracy and the non-gradient-dominated objective function, the O(1/epsilon^5) rate is not that unreasonable. In fact, to our knowledge, there is no sample complexity lower-bound applicable to the outer-loop problem we consider here (stochastic nonconvex optimization with no global smoothness nor coercivity nor PL condition, with an inner-loop oracle). Thus, it seems not clear if (and how) our results are that “suboptimal”.

**Comparison of our inner-loop results with [H20]:**

For a fixed outer-loop gain matrix $\textbf{K} \in \mathcal{K}$, the resulting inner-loop maximization problem is a finite-horizon stochastic LQR, a setting studied in [H20] (which is an independent work that appears concurrently to our work). However, we allow the state-weighting matrix to be indefinite, while [H20] requires $Q$ to be positive-definite. Nonetheless, our inner-loop indefinite LQR problem indeed shares some technical similarities with [H20], as both works need to guarantee the non-degeneracy of the state covariance matrix at any time, by requiring the additive random noise in the system dynamics to have a positive-definite covariance matrix. We found this independent discovery interesting.

---

### Decision · Program_Chairs · 2021-09-27

**Decision:**

Accept (Poster)

**Comment:**

The paper analyzes policy gradient methods in the context of a two-player LQR game (which can be shown to correspond to certain robust control problems) and establishes polynomial sample complexity. The authors show that the optimization landscape is more difficult, as unlike standard LQR, a decrease in the objective does not ensure feasibility of the iterates. They propose a double-loop algorithm that alternates between optimizing controllers for the two players. For the problematic outer loop, they show that NPG and Gauss-Newton with certain step size preserve feasibility. They extend the results to the case where gradients are estimated from samples, and show a polynomial sample complexity.
The established sample complexity seems suboptimal, and the paper could be organized better, especially when it comes to terms of positioning contributions in the context of existing work. However, overall the contributions are strong enough to warrant acceptance.